EMBO
Molecular Medicine

# Chaperone-mediated autophagy regulates the metastatic state of mesenchymal tumors

Xun Zhou [1,8], Eva Berenger[1,8], Yong Shi [1,8], Vera Shirokova [2], Elena Kochetkova [1], Tina Becirovic[1], Boxi Zhang [1], Vitaliy O Kaminskyy [1], Yashar Esmaeilian [1], Kayoko Hosaka[3], Cecilia Lindskog [4], Per Hydbring[5], Simon Ekman[5,6], Yihai Cao[3], Maria Genander [2], Marcin Iwanicki [7], Erik Norberg [1✉] & Helin Vakifahmetoglu-Norberg [1✉]

## Abstract

Tumors often recapitulate programs to acquire invasive and dissemination abilities, during which pro-metastatic proteins are distinctively stabilized in cancer cells to drive further progression. Whether failed protein degradation affects the metastatic programs of cancer remains unknown. Here, we show that the human cancer cell-specific knockout (KO) of LAMP-2A, a limiting protein for chaperone-mediated autophagy (CMA), promotes the aggressiveness of mesenchymal tumors. Deficient CMA resulted in widespread tumor cell dissemination, invasion into the vasculature and cancer metastasis. In clinical samples, metastatic lesions showed suppressed LAMP-2A expression compared to primary tumors from the same cancer patients. Mechanistically, while stimulating TGFβ signaling dampens LAMP-2A levels, genetic suppression of CMA aggravated TGFβ signaling in cancer cells and tumors. Conversely, pharmacological inhibition of TGFβ signaling repressed the growth of LAMP-2A KO-driven tumors. Furthermore, we found that multiple EMT-driving proteins, such as TGFβR2, are degraded by CMA. Our study demonstrates that the tumor suppressive function of CMA involves negative regulation of TGFβ-driven EMT and uncovers a mechanistic link between CMA and a major feature of metastatic invasiveness.

**Keywords** Cancer; Chaperone-mediated Autophagy; EMT; Metastasis; TGFβ
**Subject Categories** Autophagy & Cell Death; Cancer

## Introduction

Metastatic progression can be associated with the acquisition of genetic, epigenomic, and metabolic characteristics that foster a tumor's ability to break out of the primary site. One of the key processes through which cancer cells invade and disseminate is the epithelial–mesenchymal transition (EMT) (Dongre and Weinberg, 2019), which has been shown to be fundamental to malignant progression (Thiery, 2002). EMT governs the dynamic process which facilitates molecular and functional changes in cancer cells that differentiate them from a relatively immobile epithelial phenotype to a more migratory mesenchymal one. Transforming growth factor-β (TGFβ) signaling is an eminent driver of EMT. Upon TGFβ ligand binding to a cell surface complex of TGFβ receptors, downstream canonical Smad or non-canonical TGFβ signaling is stimulated, which in turn activates the EMT process (Hanahan and Weinberg, 2011).

While EMT plays a key role in cancer progression by regulating the initial steps of the invasion-metastasis cascade, targeting drivers of this event to develop approaches with anti-metastatic effects has remained one of the main challenges in cancer research. Proteins that promote EMT and dissemination are often upregulated and distinctively stabilized in advanced tumor cells not only due to acquired mutations, but also because of failure in accurate clearance or dysfunctional protein control systems (Scheel et al, 2011). Yet, the most explored aspect of this reprogramming event has been the genetic and transcriptional regulation. Whether failed or inefficient protein turnover can affect pro-metastatic protein accumulation in cancer cells and control subsequent metastatic potential and progression of cancer, remains largely unexplored.

During recent years, chaperone-mediated autophagy (CMA), one of the major autophagy pathways, has been increasingly recognized to play a crucial role in various physiological and pathological conditions (Bourdenx et al, 2021; Caballero et al, 2021; Di Blasi et al, 2008; Dikic and Elazar, 2018; Nixon, 2006). CMA is the only protein-specific form of mammalian autophagy. Its specificity relies on the presence of a pentapeptide motif (KFERQ-like) prerequisite on target proteins. Upon the recognition of this motif via the cytosolic heat shock protein 70 family member 8 (HSPA8), CMA enables the translocation of individual proteins to the lysosome-associated membrane protein type 2a, LAMP-2A, which facilitates substrate uptake by the lysosome. The LAMP-2A

[1]Department of Physiology and Pharmacology, Karolinska Institutet, 171 65 Stockholm, Sweden. [2]Department of Cell and Molecular Biology, Karolinska Institutet, 171 65 Stockholm, Sweden. [3]Department of Microbiology, Tumor and Cell Biology, Karolinska Institutet, 171 65 Stockholm, Sweden. [4]Department of Immunology, Genetics and Pathology, Rudbeck Laboratory, Uppsala University, Uppsala, Sweden. [5]Department of Oncology and Pathology, Karolinska Institutet, 171 64 Stockholm, Sweden. [6]Thoracic Oncology Center, Theme Cancer, Karolinska University Hospital, 17176 Stockholm, Sweden. [7]Department of Chemistry and Chemical Biology, Stevens Institute of Technology, Hoboken, NJ, USA. [8]These authors contributed equally: Xun Zhou, Eva Berenger, Yong Shi. ✉E-mail: Erik.norberg@ki.se; Helin.norberg@ki.se

protein is a critical molecular component of CMA as the only known rate-limiting regulator of this process to date, and its levels directly correlate with CMA activity (Kaushik and Cuervo, 2018).

Beyond its primary function as a protein quality control mechanism, emerging evidence suggests a role for CMA in multiple pathological conditions. Numerous studies have described the relationship between CMA and cancer since its first reported involvement (Kon et al, 2011). There has been evidence for both a pro-oncogenic capability of CMA in supporting cancer growth (Arias and Cuervo, 2020; Kaushik and Cuervo, 2018) and a tumor suppressive function, in line with a higher risk of malignant transformation due to declined CMA activity with age (Arias and Cuervo, 2020; Hao et al, 2019; Kacal et al, 2021; Kon et al, 2011; Schneider et al, 2015; Vakifahmetoglu-Norberg et al, 2013; Xia et al, 2015). While the anti-cancer activity of CMA is mostly attributed to its selective targeting of multiple notorious oncoproteins, including Eps-8, mutant p53, HK2, Chk1, Mdm2, and c-Myc (Gomes et al, 2017; Park et al, 2015; Rios et al, 2021), little is still known on how and to what extent CMA may contribute to cancer progression and aggressiveness. Further, while CMA has been reported to differ between primary cancer and healthy tissues (Ding et al, 2016; Han et al, 2017; Kon et al, 2011; Zhou et al, 2016), isoform specific *LAMP-2A* knockout (KO) human cancer cells and xenograft tumor models have been lacking, limiting the exploration of CMA's potential mechanism to affect the process of cancer progression. In this regard, the impact of complete CMA loss-of-function on carcinogenesis and CMA's role regarding biological aspects of tumor type, stage, and metastatic potential of cancer cells remains to be thoroughly defined in humans.

We recently created the first human specific *LAMP-2A* KO in cancer cells as CMA loss-of-function models (Yu et al, 2022). In the present study, we demonstrate that loss of LAMP-2A drives the aggressiveness of tumors and cancer cell dissemination and metastasis with mesenchymal traits through its impact on TGFβ signaling, while LAMP-2A levels are markedly suppressed in metastatic tumors. We found that CMA functionality is dampened by TGFβ-induced EMT leading to increased stability of pro-metastatic proteins promoting cancer progression. Our data reveal a reciprocal regulation of CMA and TGFβ-induced EMT, a major feature of metastatic invasiveness. Overall, our study provides mechanistic insight into the tumor-suppressive nature of CMA by connecting two important oncogenic pathways, TGFβ signaling and the pentose phosphate pathway (PPP), to LAMP-2A loss-of-function. Furthermore, it suggests that CMA activity can be an important determinant of metastatic potential in mesenchymal cancer. Thus, a CMA-based approach is likely to exhaust the metastatic potential of cancer cells through modulation of TGFβ signaling.

# Results

## Broad implication of CMA targetability on multiple steps of the epithelial–mesenchymal transition

It remains largely unknown whether CMA can display degradative selectivity towards regulators of metastasis and thereby affect the metastatic potential and further progression of cancer cells. To experimentally analyze this, we undertook a multiplexed

quantitative mass spectrometry (MS)-based proteomics approach of the ES2 cell line, derived from a poorly differentiated ovarian clear cell carcinoma, which displays high genetic similarity to ovarian tumors with mesenchymal traits (Gong et al, 2021; Guo et al, 2017). Cells grown under normal culture conditions were compared to cells treated with the small molecule receptor tyrosine kinase inhibitor AC220 (quizartinib), known to selectively reduce cellular glucose uptake. Cells were simultaneously treated with the specific and potent macroautophagy (MA) inhibitor, Spautin1 (Liu et al, 2011), to eliminate the contribution from the MA and to activate CMA, as previously described (Hao et al, 2019; Xia et al, 2015). Moreover, AC220+Spautin1 treated ES2 cells were compared to cells genetically depleted of LAMP-2A, enabling the detection of the lysosomal proteome impacted by CMA. Lysosomal fractions were then isolated from cells under these treatment conditions by differential large-scale multi-layered density gradient centrifugations and analyzed in four replicates by 8-plex tandem mass tags (TMTs) isobaric-based labeling approach, allowing simultaneous comparison of dynamic changes in protein abundance in multiple samples (Fig. EV1A,B). The sample groups were compared by principal component analysis (PCA), which visualized a good overlap within the biological replicates for all sample sets and a clear separation of the sample groups (Fig. EV1C).

The proteomic analysis yielded quantifications of proteins with a significant lysosomal enrichment (adj. *P* value < 0.01) upon treatment compared to control samples. To provide biological context for this proteome, we performed Gene Ontology (GO) analyses (Fig. EV1D). The dot plot enrichment map revealed the top 10 significantly overrepresented pathways (*P* < 0.05), uncovering proteins annotated to metabolic, endoplasmic reticulum (ER), and multiple integrated stress response-related processes as the most enriched pathways following treatment. This coincides with the fact that glucose is required for glycosylation, a post-translational modification that occurs primarily in the ER and with the fact that AC220 treatment-induced glucose starvation leads to accumulation of unfolded or unprocessed proteins during ER stress.

Next, proteins with a significant fold enrichment upon treatment over the control and that also displayed at least 30% reduced lysosomal levels due to LAMP-2A depletion were considered as LAMP-2A dependent. These combined criteria allowed a more stringent approach that increased the probability to identify proteins that are reliant on CMA. In total, 15% of the lysosomal enriched proteins were significantly (adj. *P* value < 0.01) reduced, indicating their dependency on LAMP-2A. Out of the total LAMP-2A-dependent proteome, 59% displayed canonical, while 34% contained putative KFERQ-like CMA motif(s) (Fig. EV1E). To test whether the degradation dynamics of these proteins had any association with specific cellular processes, we performed GO term enrichment analyses comparing the LAMP-2A dependent proteome (AC220+Spautin1+si*LAMP-2A*) to those enriched upon AC220+Spautin1 treatment. This uncovered significant enrichment of organismal and connective tissue processes, RNA processing, and lipid metabolic pathways as the top significantly enriched pathways for the rescuable proteins with canonical CMA motifs. Notably, the regulation of TGFβ1 signaling was among the top 10 enriched pathways (Fig. EV1F). Further characterization of the LAMP-2A-dependent proteome revealed

multiple proteins involved in cellular processes covering cell migration and adhesion events (Fig. 1A). Mesenchymal markers and positive regulators of EMT, such as TGFβ1 and Vimentin, were in particularly enriched in lysosomes upon CMA activation in a LAMP-2A-dependent manner (Fig. 1B).

To provide further evidence, we examined the prevalence of CMA motifs within the entire human pro-metastatic proteome, based on the rationale that CMA selectively targets proteins with a KFERQ-like motif. We began by compiling a comprehensive list of all genes in the human genome annotated with the GO term for EMT, encompassing all pathways implicated in EMT. After microRNAs were excluded, only protein-coding genes covered by both GO terms and Wiki pathways were combined for further selection. Proteins involved in the positive regulation of EMT and signaling processes related to epithelial cell proliferation and migration, referred to as pro-metastatic, were subjected to in silico CMA motif search (Fig. 1C). We uncovered the presence of CMA motifs in about 48% of all EMT-promoting proteins across all phases of the EMT process, including 93 EMT-inducing proteins, 4 EMT-driving transcription factors (TF)s, and 5 classical mesenchymal markers, such as TGFβ, TWIST, Vimentin, Fibronectin, and N-cadherin (Fig. 1D,E, Table 1), featuring either one or multiple canonical motifs in their amino acid sequences. In the remaining proteome, 31% were predicted with putative CMA motifs generated by post-translational modifications. This reveals a broad and significant implication of CMA activation on multiple aspects of EMT and provides an explanation as to why an enhanced lysosomal accumulation of EMT-related proteins was observed in our proteomic study.

Combined, these findings show that CMA displays a broad targetability on multiple steps of the EMT process and indicate a substantial role of CMA in regulating the stability of pro-metastatic and EMT-driving proteins by promoting their lysosomal degradation.

## LAMP-2A expression diminishes in metastatic tumors with mesenchymal traits

To investigate the clinical importance of our findings, we undertook two approaches using patient material. First, we collected samples from patients with lung cancer, representing a cancer type with high metastatic potential. We compared surgically removed clinical samples covering primary non-small cell lung cancer (NSCLC) tumors matched to the cognate brain metastases from the same patient (Fig. 2A; Appendix Table S1). Gene expression analysis revealed that metastatic tumors in 5 out of 6 patients were indeed characterized by changes related to the EMT-phenotype with an enrichment of a pro-mesenchymal gene signature (Fig. 2B). In these patients, the transcript level of LAMP-2A was more abundant in the primary tumors when compared to the matched metastatic tumors (Fig. 2C), while no significant decrease in LAMP-2A expression was detected between the primary and metastatic tumors from the patient with no clear acquisition of mesenchymal traits (Fig. 2D). Reduced LAMP-2A levels, indicative of low CMA activity, in metastatic lesions of human tumors suggested a correlation between diminished CMA functionality and the epithelial–mesenchymal state of these tumors.

Next, we performed a large-scale LAMP-2A immunohistochemical (IHC) analysis on tissue arrays from metastatic lesions of

cancers with different origins and stages, encompassing 184 patients (from ovary, liver, colon, kidney, prostate, bladder, pancreas, larynx, stomach, small intestine, rectum, esophagus, uterus, cervix, breast, skin, skin malignant melanoma, and thyroid). We employed the anti-LAMP-2A antibody ab240018, which is KO-validated (Zhou et al, 2023), for staining of histological sections at the Human Protein Atlas. Co-assessment of Vimentin and LAMP-2A in 184 samples showed an inverse staining pattern of these two proteins in 115 (63%) of the samples (Fig. 2E). The inverse LAMP-2A/Vimentin correlation was more frequently observed (69%) in high-grade, less differentiated metastatic lesions (grade II–III) compared to low-grade (grade I) tumors (Fig. 2F). These observations showed that tumors of higher-grade display lower CMA activity, indicated by less LAMP-2A staining, while expressing higher Vimentin levels. Since Vimentin is a classical mesenchymal marker and an oncoprotein (Strouhalova et al, 2020), which confers the ability to invade and metastasize, our data provide strong evidence for an association of CMA deficiency with increased mesenchymal cancer cell states.

## LAMP-2A knockout promotes cell proliferation and growth of primary mesenchymal tumors

To understand the functional importance of CMA loss on tumor growth and the metastatic potential of cancer cells, we created isoform-specific LAMP-2A KO in epithelial-derived A549 adenocarcinoma and HT1080 fibrosarcoma cells, derived from a malignant cancer of mesenchymal origin (Fig. EV2A) by CRISPR-Cas9 gene editing (Yu et al, 2022). LAMP1, LAMP-2B, and LAMP-2C transcript and protein levels were unaffected (Fig. EV2B–D). While AC220+Spautin1 treatment significantly elevated the number of CMA-indicative puncta per cell in wild-type (WT) cancer cells (measured by the PAmCherry-KFERQ reporter), no significant change was detected in the LAMP-2A KO cells providing further evidence of CMA loss (Fig. EV2E). These newly generated cells represent specific LAMP-2A KOs, providing a valuable tool for experimentally evaluating CMA loss-of-function in human cancer models.

In vitro characterization of these cells revealed that the LAMP-2A KO exhibited approximately double the growth rate and increased clonogenicity compared to the WT counterparts of HT1080, but not in A549 cells (Fig. 3A,B). To assess the impact of CMA loss on tumorigenicity in vivo, WT and LAMP-2A KO cells were both xenografted into BALB/c nude mice subcutaneously, on each flank of the same animal (Fig. 3C). As expected, LAMP-2A KO tumors displayed a complete lack of LAMP-2A protein expression, while LAMP1 and LAMP-2B protein levels were unaffected in the tumors, also observed from immunofluorescence (IF) and IHC staining (Figs. 3D and EV3A,B). Consistent with the in vitro results, KO of LAMP-2A resulted in a marked increase in volume and weight of HT1080 tumors, compared to the WT tumors in the same mice (Fig. 3E–G). Mice with WT HT1080 tumors reached the same tumor burden endpoint as mice with LAMP-2A KO tumors after twice as much time (Fig. EV3C). However, no significant difference was observed in the growth of WT and LAMP-2A KO tumors from A549 cells (Fig. 3E–G). Accordingly, LAMP-2A KO HT1080 tumors showed a higher proliferative capacity than the WT, as evident by a higher grade of 5-Ethynyl-2′-deoxyuridine (EdU) positivity (Fig. EV3D). This suggested that, unlike the epithelial

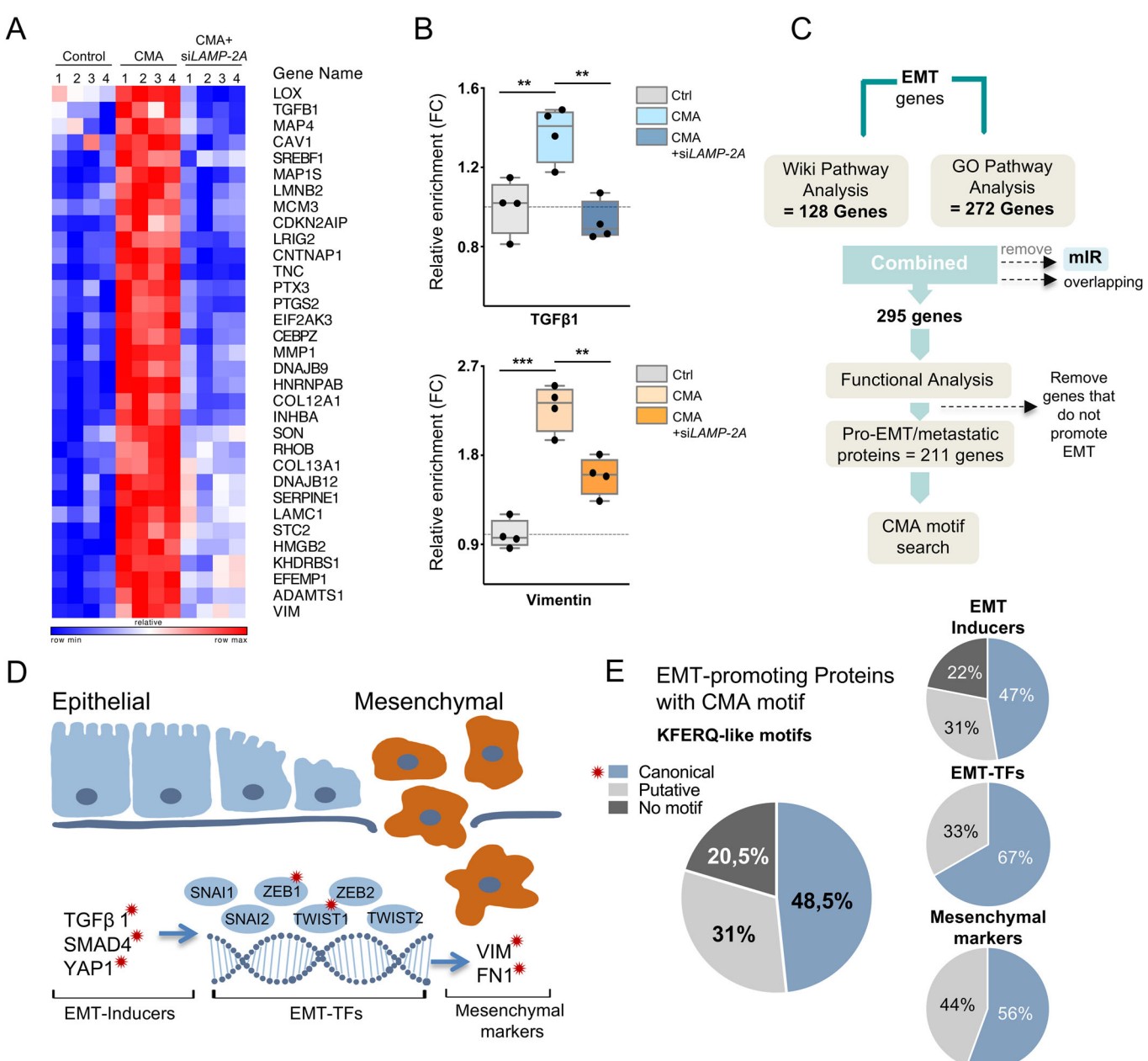

**Figure 1. Chaperone-mediated autophagy targets pro-metastatic proteins.**

(**A**) Heatmap presentation of EMT-related quantitative proteomics data obtained from multiplexed (TMT) mass spectrometry analysis detected in isolated lysosome fractions from ES2 cells untreated (control), 16 h CMA (AC220+Spautin1) or 16 h CMA following 48 h siRNA-mediated *LAMP-2A* knockdown conditions ($n_{exp} = 4$). The heatmap represents the relative expressions for each gene on each row. The row max and row min are shown for the overall heatmap. (**B**) Box plots depicting the fold TMT intensity of the lysosomal enrichment of TGFβ1 and Vimentin proteins in isolated lysosomes from ES2 cells treated as in A. Data presented as ($n_{exp} = 4$) indicated by the data points. Bars represent mean ± sd. *P* values refer to **$P < 0.01$, ***$P < 0.001$ (TGFβ1: $P_{Ctrl\ vs\ CMA} = 0.0055$, $P_{CMA\ vs\ CMA+siLAMP-2A} = 0.0017$; Vimentin: $P_{Ctrl\ vs\ CMA} < 0.0001$, $P_{CMA\ vs\ CMA+siLAMP-2A} = 0.0015$; ANOVA). The boxes extend from the 25th to 75th percentile, the middle line shows the median, whiskers extend to the most extreme data. (**C**) Schematics depicting selection criteria of CMA motif search on the total human EMT-promoting genes. Illustration (**D**) and prevalence/percentages (**E**) of EMT-inducer, EMT-transcription factors (TFs), and mesenchymal marker proteins containing canonical CMA (KFERQ)-motifs marked with star (red). Putative CMA motifs can be generated by post-translational modifications that change the biophysical properties of amino acids. Source data are available online for this figure.

**Table 1. EMT-related proteins with canonical CMA motif(s).**

| No. | Proteins | CMA motif | Motif(s) |
|---|---|---|---|
| **1. EMT transcription factors (EMT-TFs)** | | | |
| 1 | TWIST1 | Single | VRERQ |
| 2 | TWIST2 | Single | VRERQ |
| 3 | ZEB1 | Single | DKIFQ |
| 4 | ZEB2 | Single | QEFVK |
| **2. Mesenchymal markers** | | | |
| 5 | FN1 | Single | QVLRD |
| 6 | ITGB3 | Single | EVKKQ |
| 7 | MMP1 | Multiple | QVEKR, EKLKQ |
| 8 | N-cadherin | Multiple | QELVR, REKVQ |
| 9 | Vimentin | Multiple | KVELQ, RFLEQ, ELRRQ, REKLQ, RDVRQ |
| **3. EMT inducing proteins** | | | |
| 10 | ADAM17 | Multiple | QRLEK, KRDLQ, EKRVQ, KLDKQ |
| 11 | ANXA1 | Single | EFLKQ |
| 12 | BMP2 | Multiple | QVFRE, VFREQ |
| 13 | BMP4 | Single | LFREQ |
| 14 | CARD10 | Multiple | QRLRD, RLRDQ, QRLRE, LRELQ, RDRIQ, QEILR, RLDFQ, RELVQ, ELLRQ |
| 15 | CCND1 | Single | QKEVL |
| 16 | CDC42 | Single | QIDLR |
| 17 | CEACAM1 | Single | EVKKQ |
| 18 | CEBPB | Multiple | QKKVE, KKVEQ |
| 19 | CXCL13 | Single | IDRIQ |
| 20 | DAB2 | Multiple | KDLFQ, FKDFQ |
| 21 | DDX5 | Multiple | QIRDL, KIVDQ, KEVRQ |
| 22 | EFNB2 | Single | QDIKF |
| 23 | ELL3 | Multiple | QRFIE, DKIIQ |
| 24 | EP300 | Multiple | DIFKQ, QDRFV, QFRDI |
| 25 | EPB41L5 | Single | EKLKQ |
| 26 | EPHA2 | Multiple | QRVDF, EKVVQ |
| 27 | ERBB2 | Single | LRELQ |
| 28 | ERN1 | Multiple | LEKIQ, QLLRE, ERLFQ |
| 29 | EZH2 | Multiple | QKILE, QLKKD |
| 30 | FERMT1 | Single | KLVEQ |
| 31 | FGF9 | Single | VFREQ |
| 32 | FGFR2 | Single | KEFKQ |
| 33 | FLT1 | Multiple | RRIDQ, QKKEI |
| 34 | FOXM1 | Single | QVKVE |
| 35 | FZD2 | Single | QERVV |
| 36 | GLI1 | Single | LRLDQ |
| 37 | GPC3 | Multiple | RELIQ, QIIDK |
| 38 | GPI | Single | RKELQ |
| 39 | HBEGF | Single | VRDLQ |
| 40 | HDAC1 | Multiple | EKIKQ, QRLFE |
| 41 | HDAC2 | Multiple | EKIKQ, QRLFE |
| 42 | HNRNPAB | Single | KVLDQ |
| 43 | HRAS | Single | REIRQ |
| 44 | HRG | Single | QDLRV |
| 45 | HSPB1 | Single | RLFDQ |
| 46 | ID1 | Single | IRDLQ |
| 47 | IGF1 | Single | QRREI |
| 48 | IL4 | Single | QEIIK |
| 49 | IL6 | Multiple | RIDKQ, KEFLQ |
| 50 | IRF6 | Multiple | DIKFQ, LIERQ |

**Table 1.** (continued)

| No. | Proteins | CMA motif | Motif(s) |
|---|---|---|---|
| 51 | ITCH | Multiple | QDLRR, QFVKE, QLKEK |
| 52 | ITGA2 | Multiple | EKFVQ, LDVRQ, LKREQ |
| 53 | ITGA3 | Single | QKLEL |
| 54 | ITGB4 | Multiple | QKEVR, IIKEQ |
| 55 | JAG1 | Single | EVRVQ |
| 56 | JUN | Single | QERIK |
| 57 | JUNB | Multiple | QERIK, LLREQ |
| 58 | KDM5B | Single | IRLEQ |
| 59 | KRAS | Single | REIRQ |
| 60 | LEF1 | Multiple | QKEKI, REKLQ |
| 61 | LGR5 | Single | QKIDL |
| 62 | LOXL3 | Multiple | RVEIQ, RVEVQ |
| 63 | LRG1 | Single | QLERL |
| 64 | MAD2L2 | Single | QVIKD |
| 65 | MAP3K3 | Single | ERIVQ |
| 66 | MAP3K7 | Single | VELRQ |
| 67 | MAP4K4 | Multiple | FIRDQ, QKRIE, KRIEQ, VEKKQ |
| 68 | MTOR | Multiple | QLKKD, KKDIQ, RIKEQ, IFRDQ, EIIRQ, QRIVE, QKVEV |
| 69 | MYC | Multiple | VEKRQ, VLERQ |
| 70 | NODAL | Multiple | ELRLQ, LERFQ |
| 71 | NOTCH4 | Single | QLRDF |
| 72 | NR2C2 | Multiple | DRIKQ, QIEKF, IEKFQ |
| 73 | NR2F2 | Single | QVEKL |
| 74 | NRP2 | Multiple | QVDLR, QVVRE |
| 75 | PAK1 | Single | KELLQ |
| 76 | PAX2 | Multiple | DVVRQ, QRIVE |
| 77 | PIK3CA | Multiple | QDFRR, DILKQ, FERFQ |
| 78 | PIK3R3 | Multiple | QLVKE, KIRDQ |
| 79 | PLXND1 | Multiple | IDVRQ, QVKEK, QKFLD |
| 80 | PRKCA | Single | QKFEK |
| 81 | PROX1 | Multiple | QFIDR, LIREQ |
| 82 | PTK2 | Multiple | QKIVD, QRERI, EKFLQ |
| 83 | PTPN11 | Multiple | QLKEK, DKVKQ, QRRIE |
| 84 | ROCK1 | Multiple | VDLKQ, QVKEL, QDLRK |
| 85 | ROCK2 | Multiple | QRKLE, LEKRQ, ELLKQ, QKDVL, LKIEQ, QVREL, LKDVQ, QIRIE, RIELQ |
| 86 | SERPINF1 | Single | RLDLQ |
| 87 | SHC1 | Single | EVRKQ |
| 88 | SKP1 | Multiple | QEFLK, LKVDQ, QVRKE |
| 89 | SMAD4 | Single | FDLRQ |
| 90 | SRC | Single | EKLVQ |
| 91 | STAT1 | Multiple | KFLEQ, VVERQ, LLKDQ |
| 92 | STAT3 | Multiple | QDVRK, VVERQ |
| 93 | STAT5A | Single | IIEKQ |
| 94 | TGFß1 | Multiple | DKFKQ, LKVEQ |
| 95 | TGFß2 | Multiple | RDLLQ, QRIEL |
| 96 | TGFß3 | Multiple | QRIEL, IDFRQ |
| 98 | TIAM1 | Single | KKLEQ |
| 99 | TERT | Single | QLREL |
| 100 | TNC | Multiple | QIEVK, QRKLE |
| 101 | TRIM28 | Multiple | QVDVK, RLERQ, QRFFE |
| 102 | XBP1 | Single | QLLRE |
| 103 | YAP1 | Multiple | QELLR, ELLRQ |

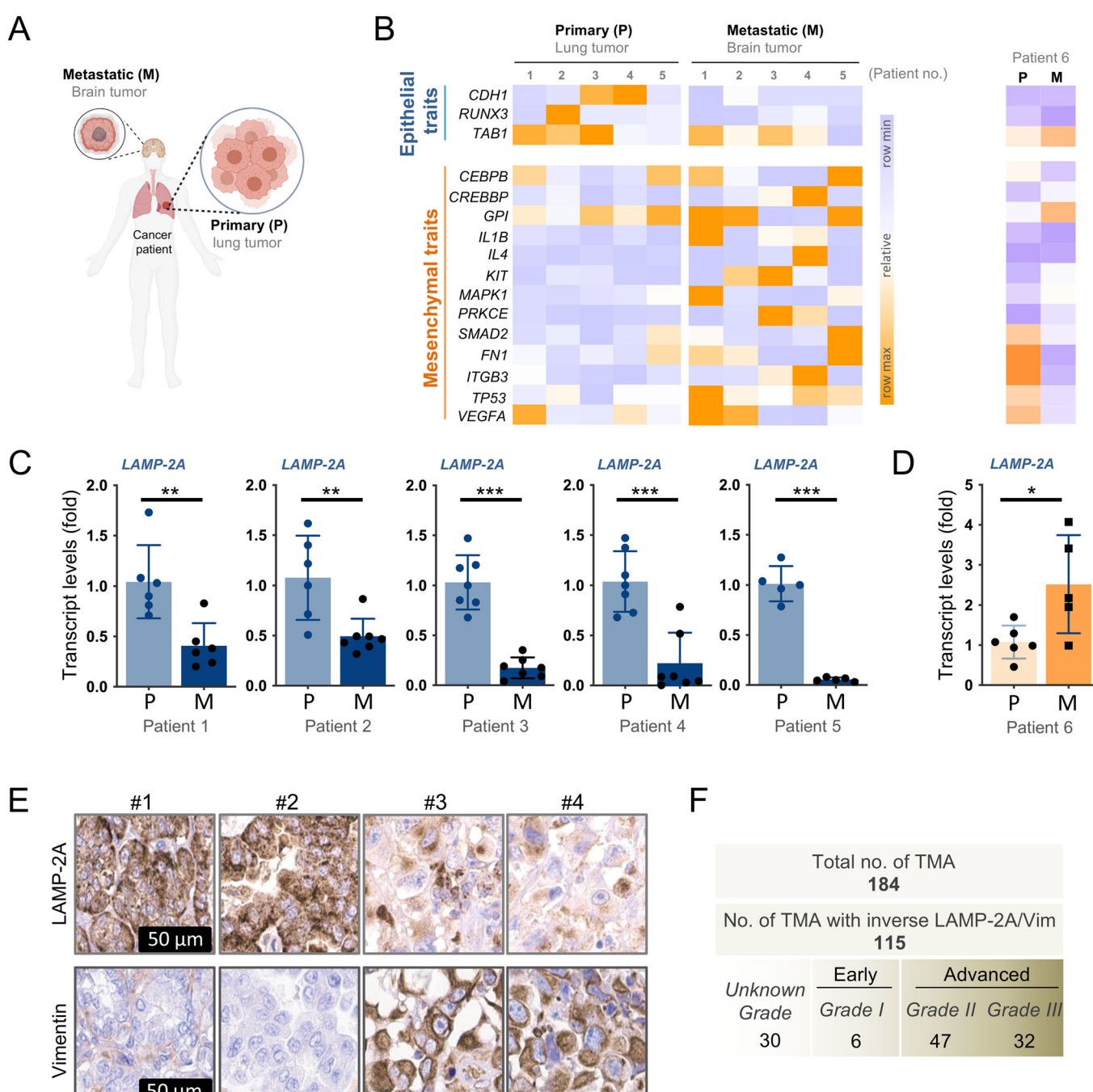

Figure 2. Suppressed expression of *LAMP-2A* in human metastatic lesions compared to matched primary tumors.

(A) Schematic illustration of matched NSCLC primary tumor and brain metastasis in cancer patient. (B) Comparative RNA expression analysis of clinical human NSCLC primary (P) tumors to matched brain metastases (M) (retrospective FFPE samples) ($n_{patient} = 6$), depicting indicated EMT-related genes extracted from a specific nanostring gene set. The heatmap represents the relative expressions for each gene on each row. The row max and row min are shown for the overall heatmap. (C, D) RT–qPCR detection of *LAMP-2A* expression in matched human metastatic (M) and primary (P) tumors from these patients, ($n_{technical\ repeates} > 5$), as indicated by the data points. Bars represent mean ± sd. P values refer to *P < 0.05, **P < 0.01, ***P < 0.001 ($P_{patient1} = 0.0045$, $P_{patient2} = 0.0063$, $P_{patient3} < 0.0001$, $P_{patient4} = 0.0003$, $P_{patient5} < 0.0001$, $P_{patient6} = 0.023$; Student's t-test). (E) Immunohistochemical (IHC) analysis of multiple organ metastatic tissue microarrays (TMAs) stained with anti-LAMP-2A or anti-Vimentin (Vim) antibody in 19 organs/184 cases. Two pairs of representative images are shown to illustrate the inverse LAMP-2A/Vim staining pattern. Scale bar: 50 μm. (F) Table depicting the number of TMAs with inverse LAMP-2A/Vim expression and its correlation with tumor grade, representing well-differentiated (low grade I), moderately differentiated (intermediate grade II), and poorly differentiated (high grade III). Source data are available online for this figure.

A549 cancer cell model, CMA displays a significant role in the tumorigenicity of HT1080 cancer cells, in which LAMP-2A loss aggravates the aggressiveness of this sarcoma model.

## LAMP-2A knockout promotes metastasis

To analyze the gene expression changes affected by LAMP-2A KO, we performed RNA-seq on the HT1080 xenografts and compared their expression profiles with those in WT tumors. GO analysis of the RNA-seq data revealed that the top 10 most enriched pathways (adjusted $P$ value < 0.01), included several related to cell migration, cell adhesion, and blood circulation (Fig. 4A). We further found that several angiogenesis and metastasis-related genes were significantly upregulated (adjusted $P$ value < 0.05, fold >2) in the HT1080 LAMP-2A KO tumors (Fig. 4B). Moreover, the staining intensities of the ID1 protein were significantly higher in LAMP-2A KO HT1080 tumors as compared to the WT (Fig. EV3E). Given that ID1 contributes to cell invasion and migration, and a correlation between ID1 protein and tumor angiogenesis has been determined (Hu et al, 2017), this further supports our observation of the driving impact of LAMP-2A loss on malignant progression.

To experimentally assess whether LAMP-2A KO impacts the metastatic colonization to distant organs, we injected WT and LAMP-2A KO A549 or HT1080 cancer cells into the tail veins of mice and analyzed potential metastasis after 21 days (Fig. 4C). Mice bearing HT1080 LAMP-2A KO cells showed a heavier tumor burden in lung tissue with a significantly increased number of metastatic foci on the lung surface compared with mice bearing WT HT1080 cells (Fig. 4D–F). Histological analysis of hematoxylin and eosin (H&E)-stained lung sections further confirmed an elevation of spontaneous tumor metastasis in mice injected with LAMP-2A KO cells compared to WT HT1080 (Fig. 4F). A slight increase in lung metastasis count was observed in mice bearing A549 LAMP-2A KO cells compared to mice injected with WT cells, albeit to a much lesser extent than the HT1080 KO cells. However, compared to A549 cells, mice bearing HT1080 LAMP-2A KO cells further developed metastatic lesions in organs beyond the lung, including the adrenal gland and subcutaneous metastases, while no tumors were detected in any other tissue in mice with A549 LAMP-2A KO cells (Fig. 4G).

To further assess the effects of CMA loss on cancer spread, we implanted WT or LAMP-2A KO HT1080 cells in zebrafish embryos, which resulted in widespread tumor cell dissemination and invasion into the vasculature within two days, the earliest step of cancer metastasis (Fig. 4H). In contrast, most WT HT1080 cancer cells remained at the primary sites. These data show that KO of LAMP-2A stimulated invasion and increased tissue colonization of mesenchymal cancer cells, suggesting that CMA plays a regulatory role in the metastasis progression.

## CMA deficiency rewires metabolism towards reliance on the pentose phosphate pathway

To explore the cause behind the observed growth advantage of CMA-deficient tumors and to test whether tumorigenicity of LAMP-2A KO tumors was associated with distinct metabolic patterns, we compared the metabolome of WT and LAMP-2A KO xenograft tumors by targeted quantitative mass spectrometry-based metabolomics, monitoring >110 key metabolites of major metabolic

pathways. This analysis confirmed that the LAMP-2A KO tumors have acquired a significantly distinct metabolic profile compared to WT tumors (Fig. 5A).

Multiple metabolic pathways could contribute to the generation of biomass required for rapidly dividing cells (Fig. 5B). Among these, amino acids are involved in key pathways that feed cancer cells to provide building blocks required for cancer cell growth. Aspartate, glutamine, serine, and glycine are among the key amino acids linked to nucleotide biosynthesis that is essential for cancer cell proliferation and tumor growth (Cantor and Sabatini, 2012; Faubert et al, 2020; Stanley et al, 2014; Sullivan et al, 2016; Yang and Vousden, 2016). We found no significant changes in either the individual amino acid levels across the tumor sample groups and by measuring the total amino acid levels (Fig. 5C bottom panel). Instead, we identified the metabolite levels of the PPP to be significantly more abundant in the LAMP-2A KO tumors (Fig. 5C, top panel). Given that nucleotide metabolism is enhanced to ensure a steady supply of building constituents required for RNA and DNA biosynthesis (Chandel, 2021), higher PPP is often correlated with the proliferation rate (Lane and Fan, 2015). Correspondingly, our analysis revealed the biosynthetic output of the PPP, namely nucleotides (IMP, AMP, and GMP), to be enriched in the LAMP-2A KO tumors (Fig. 5C). This predicts a carbon flux from glucose into the PPP. To test this, we measured the glycolysis metabolites (Fig. 5D) and performed 1,2-$^{13}$C$_2$-glucose isotopomer analysis comparing glucose flux from glucose-6-phosphate to fructose-6-phosphate (glycolysis intermediate, M + 2) or ribulose-5-phosphate (a metabolite unique to the PPP, M + 1) (Fig. 5E). While no difference in fructose-6-phosphate labeling was observed, a significant elevation of flux into the PPP was detected, as evident by enrichment of the M + 1 isotopologue of ribulose-5-phosphate (Fig. 5F). These data raised the possibility that the PPP may be more active in the KO tumor compared to the WT. Indeed, measurement of the rate-limiting enzyme activity of glucose-6-phosphate dehydrogenase (G6PDH), showed a ~2-fold increase in the LAMP-2A KO cells (Fig. 5G), indicating that higher G6PDH activity is likely to account for the boosted PPP. Considering that metastasizing cancers have been known to depend on G6PDH (Aurora et al, 2022), our findings demonstrate that LAMP-2A-deficient tumor cells undergo a metabolic switch to become more reliant on G6PDH, and thus PPP function.

Furthermore, we observed that the levels of glycolytic intermediates differed in an inconsistent fashion between WT and LAMP-2A KO tumors (Fig. 5C middle panel, 5D). Yet, lactate (a product of glycolysis), was significantly lower along with a deceased lactate to pyruvate ratio. This suggests that mitochondrial metabolism may be elevated along with lowered glycolysis in the LAMP-2A KO tumors. To assess the effect of glycolysis or mitochondrial metabolism on cell growth, we supplemented the cell culture medium with galactose over 5 days, which suppresses glycolysis. This resulted in a growth reduction of both WT and LAMP-2A KO by ~60% (Fig. 5H) compared to their growth in normal medium (Fig. 3A). This indicated that glycolysis may not be the metabolic feature that provides the specific growth advantages in LAMP-2A KO tumors. Instead, mitochondrial metabolism may be relevant in this context, consistent with the increased Acetyl-CoA levels observed in the LAMP-2A KO cells (Fig. 5C, middle panel). Measurement of the mitochondrial oxygen consumption rates using an Extracellular Flux Analyzer showed that mitochondrial activity was higher in the LAMP-2A KO

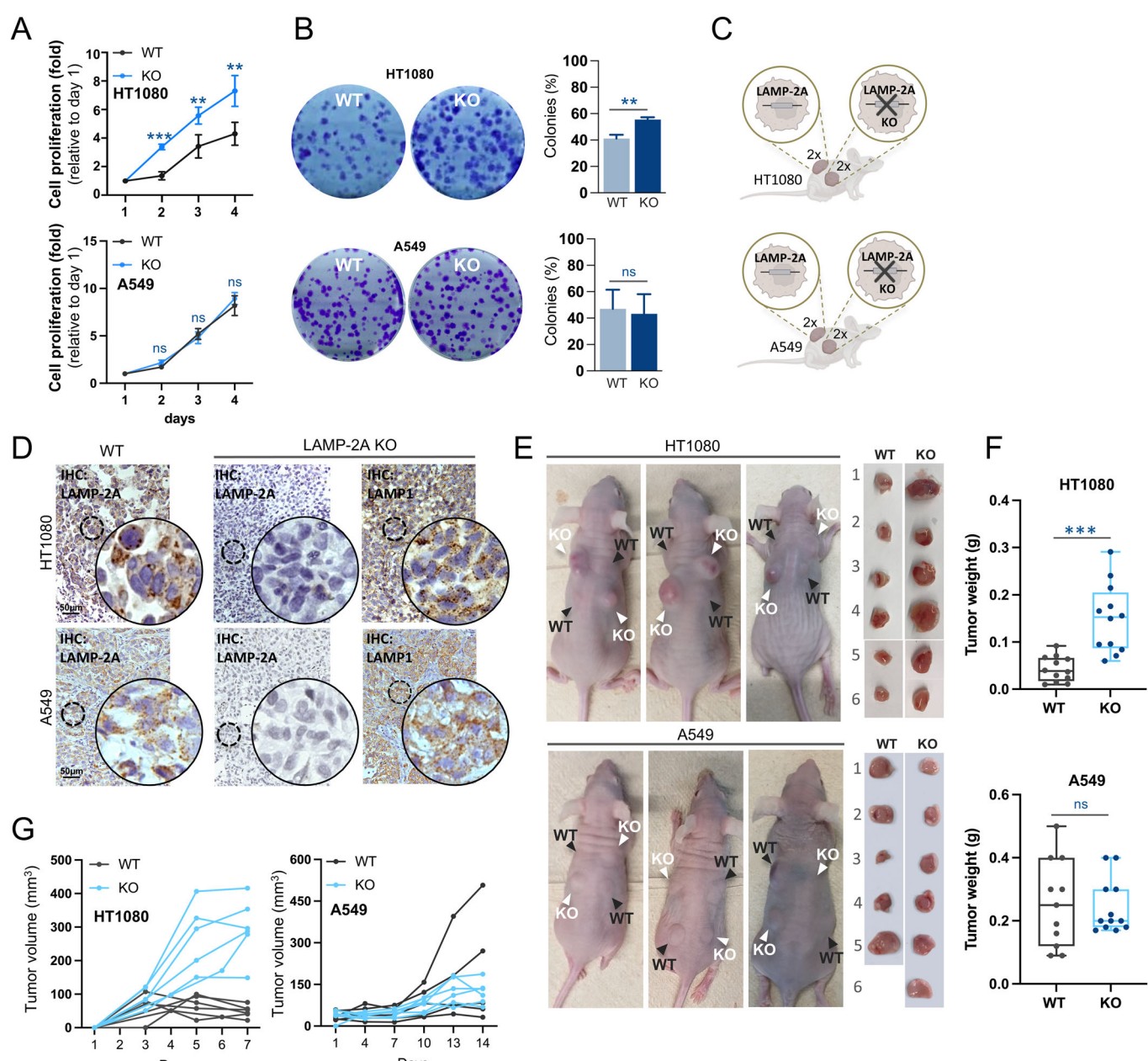

**Figure 3. Knockout of *LAMP-2A* drives proliferation and tumor growth of mesenchymal tumor type.**

(A) Cell proliferation (fold) depicting the effect of wild type (WT) and *LAMP-2A* knockout (KO) in HT1080 and A549 cancer cells, and (B) on clonogenicity (percentage) compared to WT cells ($n_{exp} = 3$). Bars represent mean ± sd. *P* values refer to **$P < 0.01$, ***$P < 0.001$; ns=non-significant (HT1080: $P_{day2} < 0.0001$, $P_{day3} = 0.0051$, $P_{day4} = 0.0043$; A549: $P_{day2} = 0.26$, $P_{day3} = 0.65$, $P_{day4} = 0.56$; clonogenicity: $P_{HT1080} = 0.0018$, $P_{A549} = 0.78$; Student's *t*-test). (C) Illustration of the xenograft model with WT and *LAMP-2A* KO cancer cells transplantation on both flanks of each animal, four tumors (two WT and two KO) per mouse ($n_{mice} = 6$/group). (D) Representative images of immunohistochemical (IHC) staining using anti-LAMP-2A or anti-LAMP1 antibody on WT and *LAMP-2A* KO HT1080 and A549 tumor sections, highlighted with circular magnification. The scale: 50 μm. (E) Representative images (left panel) from three out of six xenograft mice transplanted with WT and *LAMP-2A* KO HT1080 or A549 cancer cells, and (right panel) harvested tumors from these mice ($n_{tumor/condition} = 5$–6). One transplanted mouse from the A549 group had no growth in one WT tumor. (F) Ex vivo weights of all harvested WT and *LAMP-2A* KO HT1080 or A549 tumors ($n_{mice} = 6$/group, $n_{tumor/condition} = 11$–12) as indicated by the data points. Bars represent mean ± sd. *P* values refer to ***$P < 0.001$; ns = non-significant ($P_{HT1080} < 0.0001$, $P_{A549} = 0.82$; Student's *t*-test). The boxes extend from the 25th to 75th percentile, the middle line shows the median, whiskers extend to the most extreme data. (G) Individual growth curves of WT and *LAMP-2A* KO HT1080 and A549 tumors. Source data are available online for this figure.

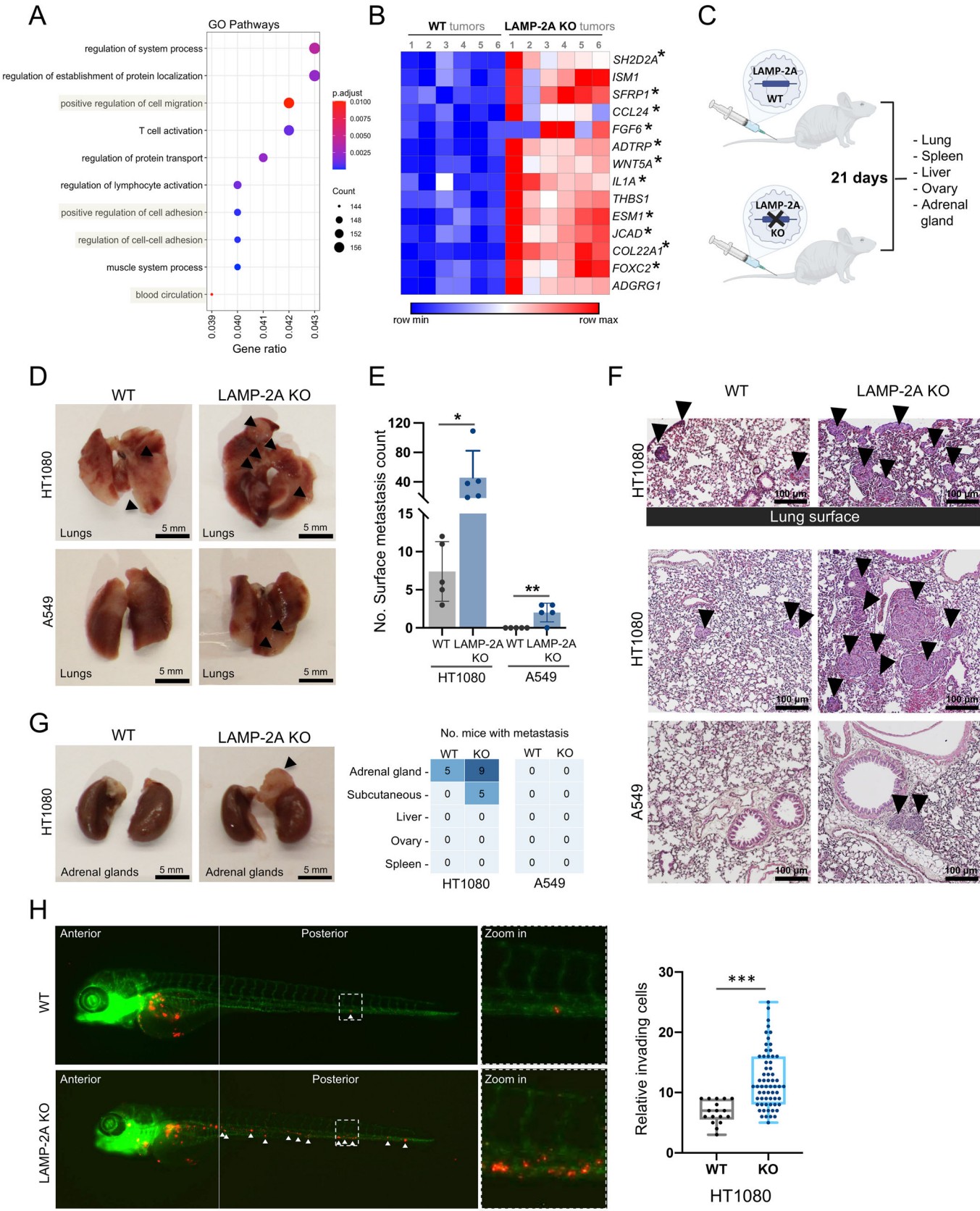

**Figure 4. LAMP-2A knockout promotes invasion and metastasis of mesenchymal cancer cells.**

(A) Pathway enrichment analysis by Gene ontology (GO) comparing RNA-seq data obtained on WT and *LAMP-2A* KO HT1080 mice xenograft tumors. Dot plot of top 10 enriched pathways indicated as the ratio of the differentially expressed gene number to the total gene number in a certain annotation. The size and color of the dots represent the gene ratio and the range of adjusted *P* values, respectively (Over Representation Analysis). (B) Heatmap presentation of the significantly upregulated genes related to angiogenesis or metastasis in *LAMP-2A* KO mouse xenografts compared to control tumors ($n_{tumor}$ = 6). Pro-angiogenic genes are indicated with a star*. The heatmap represents the relative expressions for each gene on each row. The row max and row min are shown for the overall heatmap. (C) Illustration of the experimental metastasis assay by tail vein injection of WT or *LAMP-2A* KO cancer cells in mice up to 21 days ($n_{mice}$ = 5/group) and collected organs. (D) Representative photographs of lungs from mice injected with either WT or *LAMP-2A* KO A549 or HT1080 cancer cells; arrows point on surface metastases. (E) Surface lung metastasis assessed by quantifying the surface foci in lungs of mice ($n_{mice}$ = 5/group), as indicated by the data points. Bars represent mean ± sem. *P* values refer to *$P$ < 0.05, **$P$ < 0.01 ($P_{HT1080}$ = 0.0497, $P_{A549}$ = 0.0065; Student's *t*-test). (F) Representative images of H&E-stained lung sections. Upper two images include representative lung surface from WT or *LAMP-2A* KO HT1080 cell bearing mice lung. The middle and bottom images show lung section from WT or *LAMP-2A* KO HT1080 or A549 cancer cell bearing mice; arrows point on metastatic lesions. (G) (Left panel) Representative photographs of adrenal glands from mice bearing either WT or *LAMP-2A* KO A549 or HT1080 cancer cells. (Right panel). Heatmap presentation of total number of metastases per condition in collected organs. (H) Representative images (left panel) and quantification (right panel) of zebrafish embryos with GFP-labeled vasculature (green) injected with WT or *LAMP-2A* KO HT1080 cells (red) in the yolk sac. Marked migration of cancer cells (white arrows) into the vasculature to reach the posterior tail region and their extravasated in the embryo's caudal region is magnified (zoom) two days after injection. ($n_{embryo}$ = 48). Bars represent mean ± sd. *P* values refer to ***$P$ < 0.001 ($P_{HT1080}$ < 0.0001; Student's *t*-test). The boxes extend from the 25th to 75th percentile, the middle line shows the median, whiskers extend to the most extreme data. Source data are available online for this figure.

compared to the WT (Fig. 5I), without boosting the production of reactive oxygen species (Fig. 5J). Consequently, suppression of mitochondrial metabolism by a low non-toxic concentration of rotenone, a complex I inhibitor of the electron transport chain, eliminated the growth advantage of *LAMP-2A* KO cells over the WT (Fig. 5K). Combined, these data reveal that *LAMP-2A* KO cancer cells rewire their metabolism to meet the demands of rapid proliferation, with the PPP and mitochondrial metabolism as the major contributing metabolic features distinguishing them from WT tumors.

## Reciprocal interaction of CMA and TGFβ signaling

LAMP-2A loss had a more pronounced effect on the sarcoma cells, a cancer of mesenchymal type. To investigate whether this effect depends on the tumor cell-of-origin or the mesenchymal signature of cancer cells, we focused on carcinoma cells of epithelial origin, where the EMT process is essential to acquiring mesenchymal traits necessary for metastatic dissemination. Using a panel of ovarian carcinoma cells spanning from epithelial (E) to mesenchymal (M) states, we found that LAMP-2A expression varied significantly with the EM phenotype of the cells. *LAMP-2A* was transcriptionally less expressed in cancer cells with increased M traits compared to those with E characteristics, mirroring our observations in human clinical samples (Fig. 6A,B). Notably, one of the primary TGFβ receptors, the type II (TGFβR2) and its downstream mediator protein Co-Smad (Smad4) were inversely expressed as LAMP-2A level decreased along with mesenchymal traits increased in these cells (Fig. 6A).

TGFβ signaling is a major EMT trigger known to play a crucial role in the invasiveness of carcinomas (Colak and Ten Dijke, 2017; Derynck et al, 2021), in which Smad4 is required for TGFβ-induced EMT and metastasis (Cheng et al, 2016). Therefore, we focused on evaluating the involvement of CMA in the TGFβ signaling pathway. First, to explore the effect of EMT stimulation on CMA, we tested whether TGFβ-induced EMT signaling impacts *LAMP-2A* transcript levels. Treatment of epithelial OVPA8 cells with TGFβ ligand for 48 h successfully induced EMT, as evidenced by reduced E-cadherin (*CDH1*) and increased Vimentin (*VIM*) levels, resulting in a marked suppression of basal *LAMP-2A* expression (Fig. 6C). TGFβ-induced EMT also suppressed *LAMP-2A* levels under CMA activated conditions (Fig. 6D). Conversely, pharmacological inhibition of TGFβ signaling in the mesenchymal FUOV1 cell line with a clinically used

compound, Tranilast, caused an elevation of the LAMP-2A transcript and protein levels (Fig. 6E). In fact, silencing of *TGFβR2* by siRNA-mediated knockdown resulted in increased LAMP-2A mRNA and protein levels in FUOV1 and ES2 cancer cells (Fig. 6F,G). Taken together, these data suggest that CMA function may be suppressed during tumor progression by transcriptional repression of *LAMP-2A* with increased metastatic potential of cancer cells.

Moreover, since Tranilast treatment increased *LAMP-2A* transcript to levels comparable to CMA activation in mesenchymal cells (Fig. 6E), we hypothesized that inhibition of TGFβ signaling may trigger CMA. Correspondingly, we found that the number of puncta per cell, as measured by the CMA reporter PAmCherry-KFERQ, was significantly increased in cancer cells treated with Tranilast (Fig. 7A). Tranilast significantly reduced TGFβR2 and Smad4 protein levels, accompanied by the degradation of classical CMA substrate, mutant p53, in a similar manner to CMA activation (Fig. 7B). We further found that CMA activation by AC220+Spautin1 had a more pronounced effect on TGFβR2 protein level than on Smad4. To test whether the TGFβR2 could be controlled by CMA-mediated lysosomal degradation, we treated cells with the lysosomal inhibitor, chloroquine (CQ), which indeed blunted both the Tranilast and CMA-induced TGFβR2 protein degradation (Fig. 7C). In fact, CQ treatment alone led to increased TGFβR2 levels in cancer cells. Moreover, quantitative mass spectrometry data on isolated lysosomes confirmed reduced enrichment levels of TGFβR2 upon *LAMP-2A* silencing (Fig. 7D). In addition, we found that CMA activation significantly reduced the expression levels of the EMT-associated TFs Twist and Snail, two key regulators of EMT under TGFβ signaling, in the mesenchymal cell lines ES2 and FUOV1, respectively (Fig. 7E). In line with these data, TGFβ ligand-induced upregulation of Snail in epithelial cancer cells, normally expressed at low levels, was significantly blocked upon CMA activation (Fig. 7F), demonstrating that CMA activation leads to compromised EMT-induction by preventing the accumulation of mesenchymal proteins.

The presence of KFERQ-like targeting motifs in TGFβR2 further suggests that CMA controls TGFβR2 abundance and LAMP-2A loss results in its aberrant accumulation. In strong alignment with this, both knockdown (Fig. 7G) and KO (Fig. 7H) of *LAMP-2A* led to a marked elevation of TGFβR2 and Smad proteins levels in the mesenchymal cancer cell line HT1080, but not in the epithelial

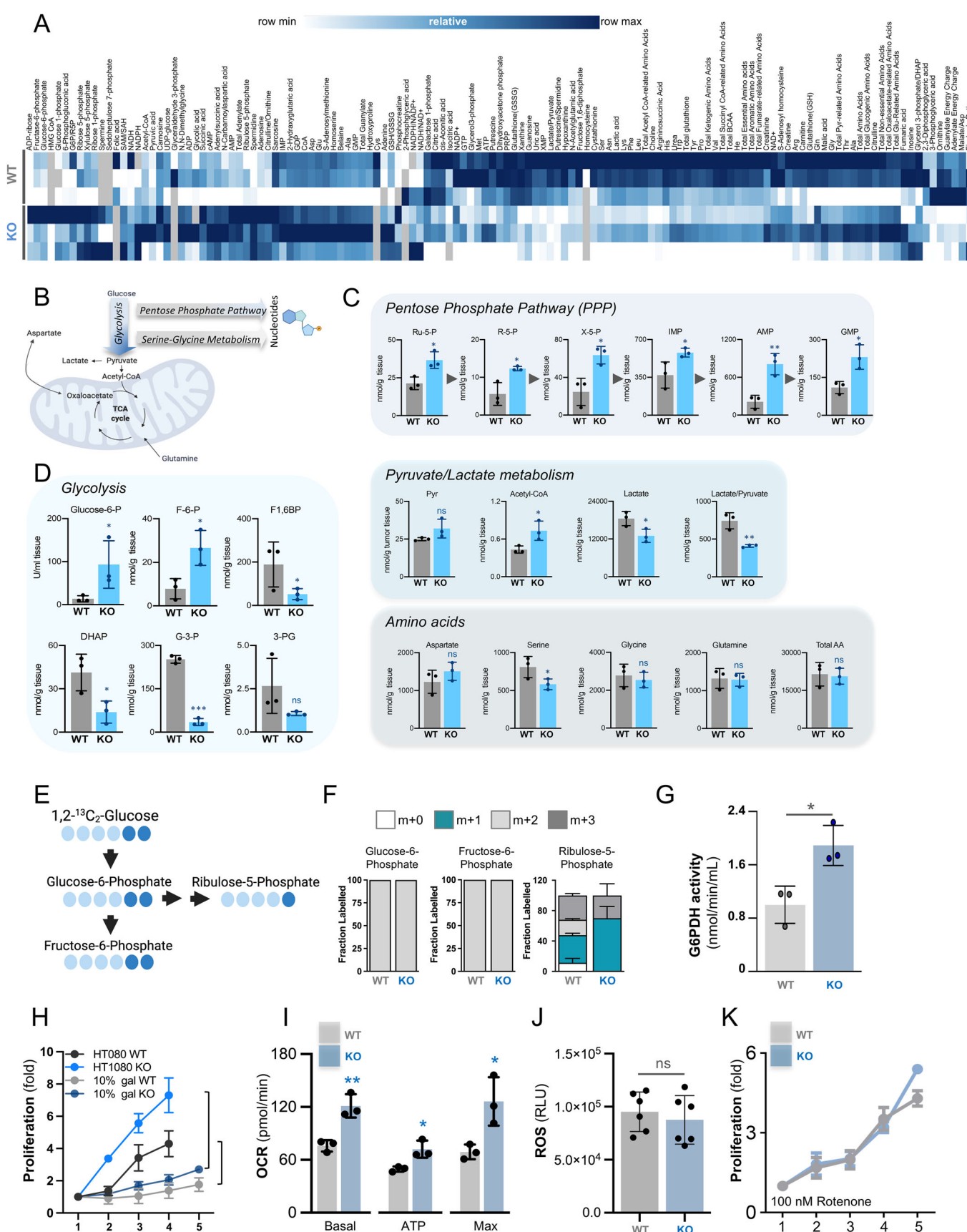

**Figure 5. CMA deficiency reshapes the tumor metabolome to support proliferation.**

(A) Metabolite levels in WT and *LAMP-2A* KO HT1080 tumors derived from mouse xenografts. Each row represents an individual tumor, each column a metabolite. (B) Schematic picture of metabolic pathways important for nucleotide production. Individual metabolite levels of central metabolic pathways including: (C) pentose phosphate pathway (PPP; $P_{Ru-5-P} = 0.018$, $P_{R-5P} = 0.019$, $P_{X5-P} = 0.017$, $P_{IMP} = 0.048$, $P_{AMP} = 0.0064$, $P_{GMP} = 0.018$; Student's *t*-test), pyruvate/lactate metabolism ($P_{Pyr} = 0.13$, $P_{Acetyl-CoA} = 0.033$, $P_{Lactate} = 0.038$, $P_{Lactate/Pyruvate} = 0.0061$; Student's *t*-test), and amino acids ($P_{Aspartate} = 0.28$, $P_{Serine} = 0.066$, $P_{Glycine} = 0.60$, $P_{Glutamine} = 0.87$, $P_{Total AA} = 0.82$; Student's *t*-test), and (D) glycolysis ($P_{Glucose-6P} = 0.069$, $P_{F6-P} = 0.025$, $P_{F1.6BP} = 0.091$, $P_{DHAP} = 0.033$, $P_{G-3-P} < 0.0001$, $P_{3-PG} = 0.16$; Student's *t*-test) in WT and *LAMP-2A* KO HT1080 tumors. Bars represent mean ± sd. (E) Schematic picture highlighting carbon flux from 1,2-$^{13}$C$_2$-glucose isotopomer into the PPP pathway (ribulose-5-phosphate) or glycolysis (fructose-6-phosphate). Dark blue dots indicate the potential carbon transfer from the isotopomer used. (F) Mass distribution vector (MDV) of the carbon flux for each of the measured metabolites (glucose-6-phoshate, fructose-6-phosphate, and ribulose-5-phosphate), and (G) enzymatic activity of G6PDH as nmol/min/mL in WT and *LAMP-2A* KO HT1080 tumors ($P = 0.020$; Student's *t*-test). Bars represent mean ± sem (MDV) or sd (G6PDH activity). (H) Cell proliferation (fold) depicting WT and *LAMP-2A* KO HT1080 cells cultured in media containing 10% galactose, compared to cell proliferation of WT and *LAMP-2A* KO HT1080 cells grown in normal cell culture media as presented in Fig. 3A. Decreased growth is shown. (I) Oxygen consumption rate (OCR) depicting basal-, ATP-linked respiration and maximal (Max) respiratory capacity, and (J) Intracellular ROS production in WT and *LAMP-2A* KO HT1080 cells ($P_{basal} = 0.0061$, $P_{ATP\ linked} = 0.018$, $P_{Max} = 0.026$, $P_{ROS} = 0.55$; Student's *t*-test). Bars represent mean ± sd. (K) Cell proliferation of WT and *LAMP-2A* KO HT1080 cells after treatment with Rotenone (inhibitor of complex I of the respiratory chain). Bars represent mean ± sem. All graphs are from metabolomics tumor data ($n = 3$) as indicated by the data points. *P* values refer to *$P < 0.05$, **$P < 0.01$, ***$P < 0.001$, ns=non-significant. Source data are available online for this figure.

A549 cell line. A similar increase in TGFβR2 was also observed in HT1080 xenograft tumors (Fig. 7I). Consistently, re-expression of *LAMP-2A* in the HT1080 KO cells restored TGFβR2 protein expression levels back to WT levels (Fig. 7J).

To probe whether the effect of CMA loss is dependent on the mesenchymal status of cancer cells rather than the tumor cell-of-origin, we created a *LAMP-2A* KO in NCI-H2087 cells, which is an adenocarcinoma hyper diploid cell line from lung metastatic site, as an additional lung cancer cell line, beyond A549 (Fig. EV4A). In line with the data on the fibrosarcoma model HT1080, CMA loss in NCI-H2087 lung cancer cells led to significant increase in TGFβR2, Snail, and Slug levels, concomitant with a marked decrease in E-Cad expression. Moreover, we created a *LAMP-2A* KO in ES2 cells (Fig. EV4B), that normally already display a mesenchymal phenotype in its WT state (Fig. 6A). In this model, KO of *LAMP-2A* instead led to further expression of TGFβR2 and expression of an additional mesenchymal marker, Snail, that is not expressed in the WT ES2 cells. These data argue that CMA loss is more relevant for the maintenance of the mesenchymal phenotype, rather than for actual induction of EMT.

Collectively, these data show a reciprocal interaction of CMA and TGFβ signaling. CMA activity, regulated through transcription of *LAMP-2A*, is modulated by TGFβ signaling. Conversely, degradation of TGFβR2 upon CMA activation decreases mesenchymal traits of cancer cells, thereby attenuating their EMT. This suggests that the increased TGFβ sensitivity, driven by elevated TGFβR2 expression in CMA-deficient cells results in enhanced EMT and the further acquisition of pro-metastatic traits.

### Inhibition of TGFβ signaling demotes the metabolic growth advantages of *LAMP-2A* KO tumors

While TGFβ-induced metabolic reprogramming, such as enhanced glycolysis and oxidative phosphorylation during EMT, has been reported in carcinomas (Hua et al, 2020; Liu et al, 2020), a direct link between TGFβ signaling and the PPP has not been described yet (Liu and Chen, 2022). To this end, we investigated whether there is a mechanistic link between *LAMP-2A* KO-mediated effects on TGFβ signaling and altered cancer metabolism. We found that inhibition of TGFβ signaling in *LAMP-2A* KO cells by Tranilast could markedly restore the KO-mediated metabolic features, including the enhanced G6PDH activity (Fig. 8A), increased mitochondrial respiration (Fig. 8B), and cell proliferation (Fig. 8C) to WT levels. This suggests that

augmented TGFβ signaling may be at least partly responsible for the metabolic switch observed in the *LAMP-2A* KO tumors. We further found that *LAMP-2A* KO-driven cancer cell proliferation was reduced by direct inhibition of G6PDH activity using 6-aminonicotinamide (6-AN) treatment (Fig. 8D). Importantly, administration of Tranilast or 6-AN to mice implanted with HT1080 WT or *LAMP-2A* KO cancer cells reversed KO-driven tumor growth and size in vivo (Figs. 8E,F and EV4C,D) and suppressed the proliferative capacity of the KO tumors (Fig. 8G), demonstrating that targeting TGFβ signaling or the rate-limiting enzyme activity of PPP can eliminate the metabolic growth advantages of *LAMP-2A* KO tumors.

## Discussion

CMA deficiency has mainly been analyzed in mice models with either whole body or tissue-specific knockout (KO) of *LAMP-2A* (Dong et al, 2020; Kaushik and Cuervo, 2018; Rothaug et al, 2015). Cells and tissues from *LAMP-2A* KO human samples have not been available earlier. Most, if not all, investigations on CMA suppression in human cancer cells, thus so far, rely on *LAMP-2A* knockdown (KD) studies (Ali et al, 2011; Arias and Cuervo, 2020; Kon et al, 2011). Here, by using CRISPR-Cas9 gene editing to generate isoform specific *LAMP-2A* KO, we performed the first investigation on the impact of complete CMA loss-of-function on malignant progression in human cancer models, to understand the extent to which CMA may be contributing to the aggressiveness of tumors.

Our study unveils the tumor suppressor function of the CMA process especially in the progression of mesenchymal cancer cell state in vitro and in vivo, while no difference was observed between WT and KO *LAMP-2A* xenografts from epithelial cancer cells within the same mice. These results provoke an idea that the mesenchymal state of the cancer cells is a critical determinant for the role of CMA in cancer, regardless of the tumor cell origin. The lack of significant changes in TGFβ signaling in *LAMP-2A* KO A549 cancer cells, which require EMT to become more aggressive, suggests that the loss of LAMP-2A alone is not sufficient to induce EMT. However, activation of CMA simultaneously with stimulation by TGFβ ligand counteract EMT induction and the expression of mesenchymal markers in epithelial carcinoma cells. Our data further show that the CMA pathway can target key components of

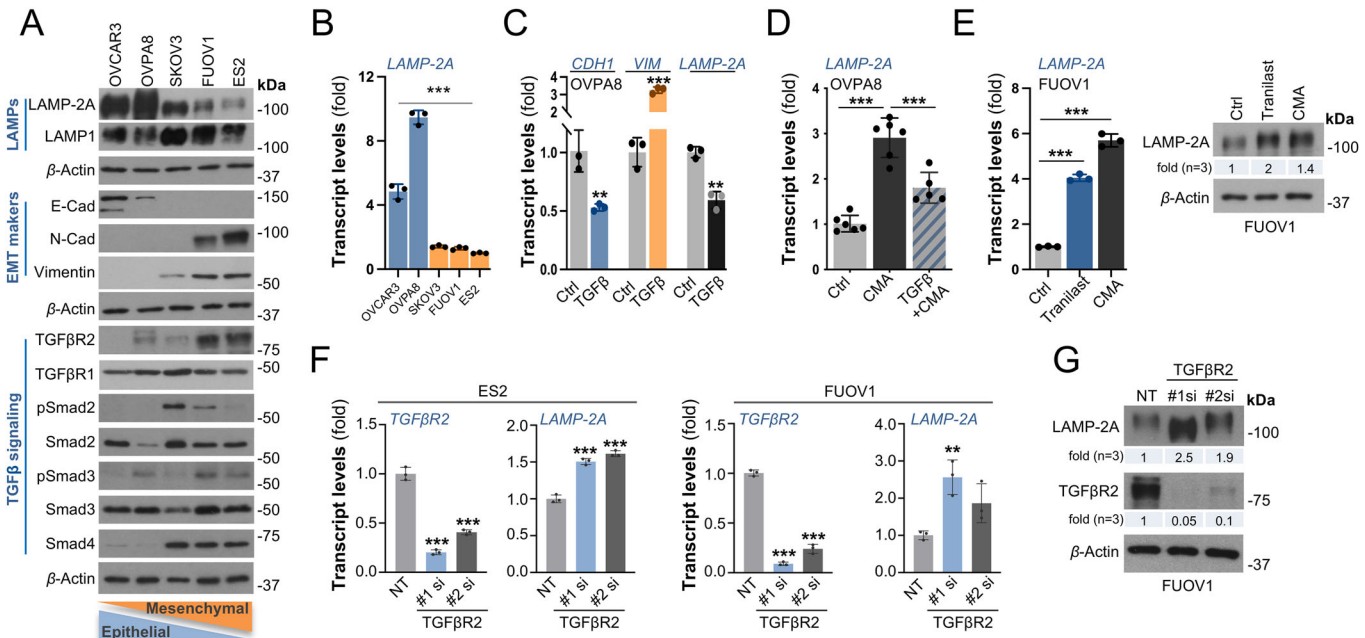

**Figure 6.  Reciprocal interaction between CMA and TGFβ signaling.**

(A) Immunoblot detection of indicated LAMP, EMT markers, and TGFβ signaling proteins in an ovarian carcinoma cancer cell line panel spanning from epithelial to mesenchymal states. β-actin: loading control. (B) RT-qPCR detection of *LAMP-2A* expression in the indicated cell line panel as in A ($P < 0.0001$; ANOVA). (C) RT-qPCR detection of *VIM* (Vimentin), *CDH1* (E-cad), and *LAMP-2A* expression in untreated and TGFβ ligand treated OVPA8 cells for 48 h to induce EMT ($P_{VIM} < 0.0001$, $P_{CDH1} = 0.0099$, $P_{LAMP-2A} = 0.0012$; Student's *t*-test). (D) RT-qPCR detection of *LAMP-2A* expression following CMA activation (AC220+Spautin1 16 h) or TGFβ ligand treatment for 48 h in OVPA8 cells ($P_{Ctrl \, vs \, CMA} < 0.0001$, $P_{CMA \, vs \, CMA+TGFβ} = 0.0002$; ANOVA). (E) RT-qPCR detection of *LAMP-2A* expression (left panel; $P_{Ctrl \, vs \, Tranilast} < 0.0001$; $P_{Ctrl \, vs \, CMA} < 0.0001$; ANOVA) and immunoblot detection (right panel) of LAMP-2A protein level in FUOV1 cells treated with the TGFβ inhibitor (Tranilast) or following CMA activation (AC220+Spautin1) for 16 h. β-actin: loading control. (F) RT-qPCR detection of *TGFβR2* and *LAMP-2A* expression following siRNA-mediated *TGFβR2* knockdown (KD) in ES2 (TGFβR2: $P_{si\#1} < 0.0001$, $P_{si\#2} < 0.0001$; LAMP-2A: $P_{si\#1} < 0.0001$, $P_{si\#2} < 0.0001$; ANOVA) and FUOV1 (TGFβR2: $P_{si\#1} < 0.0001$, $P_{si\#2} < 0.0001$; LAMP-2A: $P_{si\#1} = 0.0069$, $P_{si\#2} = 0.083$; ANOVA) cells. The KD efficiency is indicated by *TGFβR2* expression. All RT-qPCR graphs ($n_{exp} = 3$) as indicated by the data points. Bars represent mean ± sd. *P* values refer to *$P < 0.05$, **$P < 0.01$, ***$P < 0.001$. (G) Immunoblot detection of LAMP-2A protein levels following siRNA-mediated *TGFβR2* KD in FUOV1 cells. β-actin: loading control. All immunoblot data are presented with their quantification ($n_{exp} = 3$) indicated by the average signal intensity for each protein band. Source data are available online for this figure.

the EMT cascade and that pharmacological activation of CMA lessens EMT traits by degrading mesenchymal marker proteins. This further raises the possibility that an underlying cause of pro-metastatic protein accumulation may be due to functional dysregulation of CMA in tumor cells. The KO of *LAMP-2A*, thus CMA loss, indeed leads to pro-mesenchymal protein accumulation, impacting the mesenchymal state and subsequent cancer progression by driving pro-angiogenic gene expression in vivo. Moreover, in cancer cells that already exhibit a mesenchymal phenotype in their WT state, the loss of LAMP-2A results in expression of additional mesenchymal markers, suggesting that CMA loss plays a greater role in maintaining the mesenchymal phenotype rather than inducing EMT. In line with this, we discovered that *LAMP-2A* is less expressed in tumor cells that have undergone EMT in metastatic tissues of human cancers, and its expression inversely correlates with advanced tumor grading.

The role of CMA in EMT is an emerging area of research and is not as extensively studied as other aspects of CMA's involvement in cancer. Recently, impaired degradation of two proteins, the transcriptional co-activator YAP1 and IL6ST (interleukin 6 cytokine family signal transducer), by CMA has been shown to promote both the proliferation and migration of hepatocellular carcinoma cells (Desideri et al, 2023). CMA silencing has also been

shown to reduce the degradation of NF-κB p65, leading to increased levels of Twist and Snail, thereby influencing the level of E-cadherin and EMT. While these studies indicate a potential link between the two processes, our study demonstrates a much broader and more substantial role of CMA at multiple aspects of EMT, bringing insights into a function of CMA on regulating the invasiveness of cancer cells. Beyond selective targeting and degradation of multiple proteins associated with EMT, we show that CMA affects the EMT-inducing signaling pathways by directly regulating the level of TGFβR2, which is a well-known factor of EMT.

Since TGFβ signals exert tumor-promoting effects and metastasis in advanced cancer, our study indicates that the ability of CMA to inhibit TGFβ signaling may be a key to its downstream effects in cancer cells. TGFβ signaling is involved in several cell responses, and cell type and context-dependent factors contribute to the regulation of tumor metastasis. TGFβR2 is a specific receptor for TGFβ ligands. Increased TGFβ1 and receptor protein expression by tumor cells correlates with progression of multiple cancer types. However, it is also known that TGFβR2 and Smad4 could be inactivated through mutation and loss of heterozygosity in several types of carcinomas (Levy and Hill, 2006), which may in part explain the more pronounced effect of CMA displayed on cancers

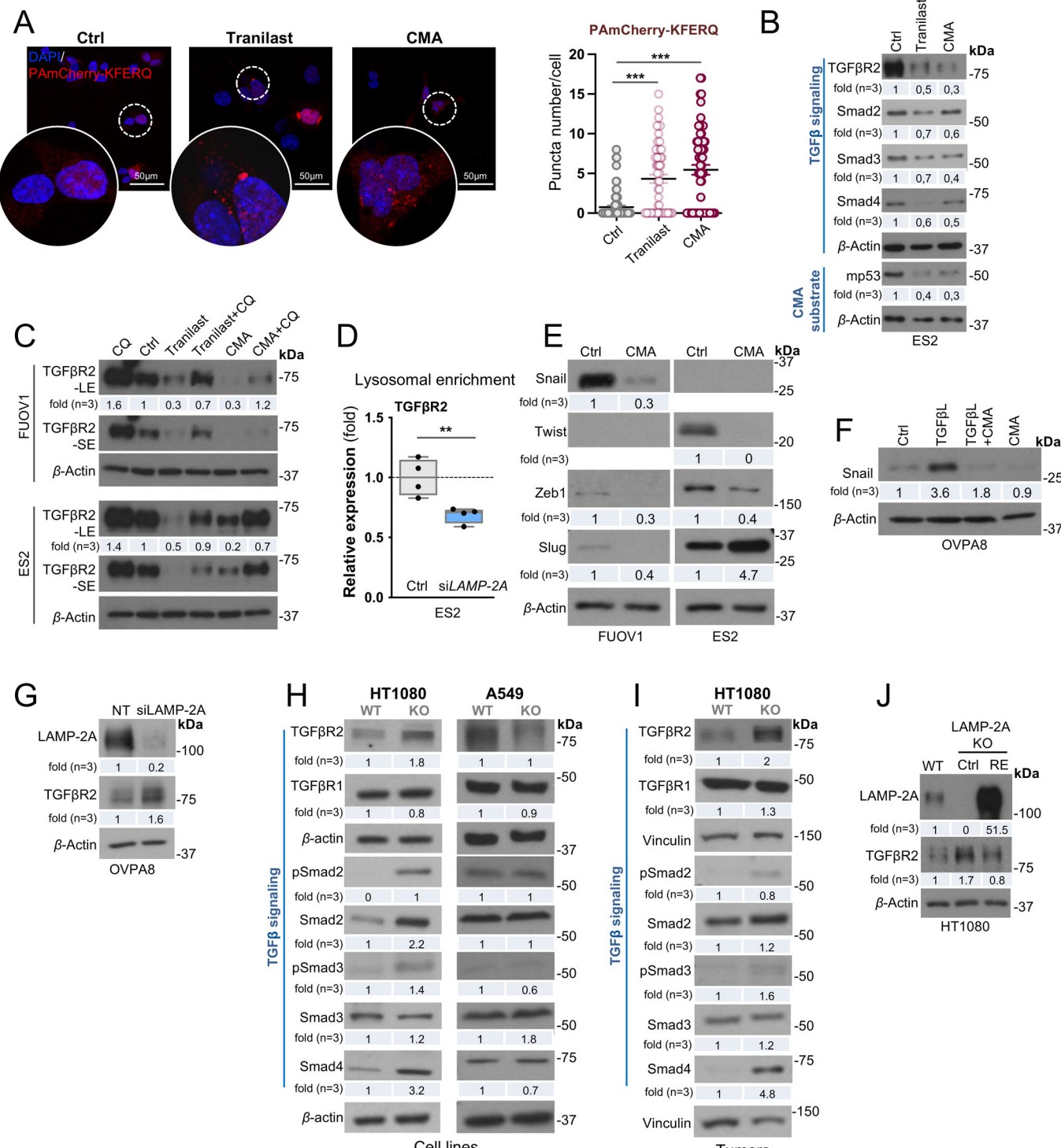

with mesenchymal phenotype. In fact, we did not find any difference in the production of TGFβ ligand, which is frequently secreted in high amounts by cancer and surrounding cells of the tumor microenvironment, in the tested WT and *LAMP-2A* KO tumors in this study.

Furthermore, we propose that CMA's ability to control TGFβ signaling might be key to its multifaceted role in cancer. TGFβ

signaling can exert different, even opposite, effects depending on the cell type and tissue-specific contexts through the regulation of cell proliferation, survival, and differentiation (Massagué, 2012). While it drives tumorigenesis at later stages, it is known to inhibit tumor development at early stages (Massagué, 2008; Massagué et al, 2000). Considering the dual role of CMA in different cellular contexts with observed enhanced CMA activity in certain primary

◀ **Figure 7.  CMA targets TGFβR2 and impacts the TGFβ signaling.**

(A) CMA activity measured in FUOV1 cells expressing PAmCherry-KFERQ CMA reporter. Cells were treated with the TGFβ inhibitor (Tranilast) or AC220+Spautin1 to activate CMA for 16 h. (Left panel) Immunofluorescence confocal images of merged red (PAmCherry-KFERQ) and blue (nuclei, DAPI) highlighted with (circular insets) magnification. The scale: 50 μm. (Right panel) Quantification of number of puncta per cell ($n_{cell} > 50$) represented by mean ± sd. $P$ values refer to $P$ *** < 0.001 ($P_{Ctrl \, vs \, Tranilast} < 0.0001$, $P_{Ctrl \, vs \, CMA} < 0.0001$; ANOVA). (B) Immunoblot detection of indicated TGFβ signaling and CMA substrate protein level in ES2 cells treated with Tranilast or CMA activation (AC220+Spautin1) for 16 h. β-actin: loading control. (C) Immunoblot detection of TGFβR2 protein level in FUOV1 and ES2 cell lines following treatment with chloroquine (CQ) alone, Tranilast, Tranilast+CQ, CMA (AC220+Spautin1), or CMA + CQ for 16 h. β-actin: loading control. LE: long exposure, SE: short exposure. (D) Fold TMT intensity enrichment of TGFβR2 level in isolated lysosomes from control and siRNA-mediated *LAMP-2A* KD ES2 cells analyzed by mass spectrometry. Data presented ($n_{exp} = 4$) as indicated by the data points $P$ values refer to **$P < 0.01$ ($P = 0.0025$; Student's $t$-test). The boxes extend from the 25th to 75th percentile, the middle line shows the median, whiskers extend to the most extreme data. (E) Immunoblot detection of EMT-TF proteins in control and CMA activated (AC220+Spautin1) FUOV1 and ES2 cells. β-actin: loading control. (F) Immunoblot detection of Snail protein in OVPA8 cells treated with the TGFβ ligand for 48 h and/or with CMA activation (AC220+Spautin1). β-actin: loading control. (G) Immunoblot detection of TGFβR2 following siRNA-mediated *LAMP-2A* KD in OVPA8 cells. β-actin: loading control. Immunoblot detection of TGFβ signaling proteins in WT or *LAMP-2A* KO HT1080 and A549 (H) cell lines, and (I) tumor lysates. β-actin or vinculin: loading control. (J) Immunoblot detection of LAMP-2A and TGFβR2 following re-expression (RE) of LAMP-2A in the *LAMP-2A* KO HT1080 cells. β-actin: loading control. All immunoblot data are presented with their quantification ($n_{exp} = 3$) indicated by the average signal intensity for each protein band. The statistical significance is presented as Appendix Table S4. Source data are available online for this figure.

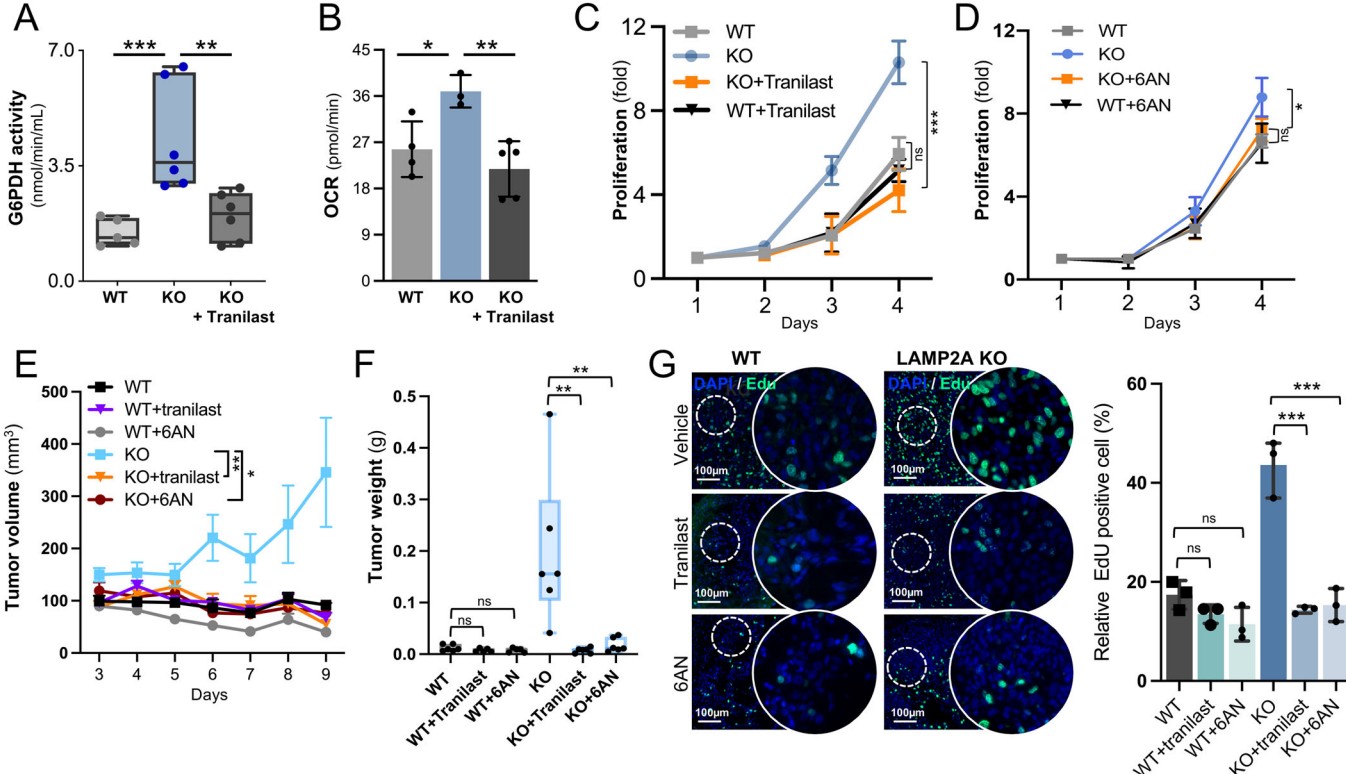

**Figure 8.  Regression of *LAMP-2A* knockout tumors in vivo.**

(A) G6PDH enzymatic activity and (B) mitochondrial respiration (OCR) in WT or *LAMP-2A* KO HT1080 cells alone or treated with Tranilast (50 μM) for 6 h (G6PDH activity: $P_{WT \, vs \, KO} = 0.0009$, $P_{KO \, vs \, KO+Tranilast} = 0.0035$; OCR: $P_{WT \, vs \, KO} = 0.032$, $P_{KO \, vs \, KO+Tranilast} = 0.0050$; ANOVA). The boxes extend from the 25th to 75th percentile, the middle line shows the median, whiskers extend to the most extreme data. (C) Proliferation of WT and *LAMP-2A* KO HT1080 cells treatment with Tranilast (50 μM), or (D) 6-AN (100 nM) up to four days (Tranilast: $P_{WT \, vs \, WT+tranilast} = 0.66$, $P_{KO \, vs \, KO+Tranilast} = 0.0003$; 6-AN: $P_{WT \, vs \, WT+6-AN} = 0.82$, $P_{KO \, vs \, KO+6-AN} = 0.027$; Student's $t$-test). (E, F) Average tumor growth and ex vivo weight of WT and *LAMP-2A* KO HT1080 tumors from mice bearing 2 xenografts, administered with Tranilast (100 mg/kg) or 6-AN (10 mg/kg) for 9 days ($n_{mice} = 3$; tumor growth: $P_{KO \, vs \, KO+Tranilast} = 0.0057$, $P_{KO \, vs \, KO+6-AN} = 0.012$; weight: $P_{WT \, vs \, WT+Tranilast} = 0.27$, $P_{WT \, vs \, WT+6-AN} = 0.29$, $P_{KO \, vs \, KO+Tranilast} = 0.0029$, $P_{KO \, vs \, KO+6-AN} = 0.0046$; ANOVA). The boxes extend from the 25th to 75th percentile, the middle line shows the median, whiskers extend to the most extreme data. (G) Representative image (left panel) and quantification (right panel) of EdU positive (green) and DAPI (blue) labeled WT and *LAMP-2A* KO HT1080 tumors sections after treatment with Tranilast and 6-AN ($n_{mice} = 3$). Corn oil was used as vehicle for the control mice group. Scale bar: 100 μm. Bars represent mean ± sd ($n_{exp} = 3$) or ($n_{tumor} = 6$) as indicated by the data points. $P$ values refer to *$P < 0.05$, **$P < 0.01$, ***$P < 0.001$, ns = non-significant ($P_{WT \, vs \, WT+Tranilast} = 0.25$, $P_{WT \, vs \, WT+6-AN} = 0.078$, $P_{KO \, vs \, KO+Tranilast} = 0.0002$, $P_{KO \, vs \, KO+6-AN} = 0.0002$ ANOVA). Source data are available online for this figure.

cancer cell types, and given the targetability of TGFβR2 by CMA, a possible suppression of TGFβ signaling at early tumor stages by CMA may promote malignant transformation. In fact, it is well known that expression of *c-MYC*, a well-known pro-oncogenic transcription factor activating the expression of many proliferative genes to promote cancer initiation, is repressed by the TGFβ pathway. Moreover, during adipose differentiation, MYC and TGFβ were both identified to be affected by CMA (Kaushik et al, 2022). In advanced tumors, activation of CMA and its mediation of TGFβR2 degradation, thus consequent loss of balanced TGFβ signaling would instead block EMT, preventing progression toward dissemination. Thus, depending on when and in which cells CMA is activated, it may exert different, even opposite, effects on cancer.

Moreover, given that mesenchymal proteins accumulate at high levels during EMT along with cancer progression, our study demonstrates that the CMA regulator LAMP-2A is less transcribed in cancer cells with increasing mesenchymal traits compared to epithelial cancer cells, mirroring our observations in human clinical samples. While the molecular details of how stimulation of TGFβ signaling suppresses the LAMP-2A level and whether this depends on canonical Smad-dependent or non-canonical TGFβ signaling needs further investigation, our findings show a significant weakening of CMA function along EMT induction and in metastatic tumors. This indicated that CMA function may be actively suppressed during cancer progression, thus less functional in advanced cancers. We speculate that cancer cells can co-regulate the stimulation of their aggressiveness through EMT signaling while suppressing CMA that counteracts EMT. In this way cancer cells retain metastatic properties to enhance their mobility and invasion, while neutralizing a pathway that prevents it. To our understanding, to date, CMA has not been investigated in matched clinical samples from patients with cancer. Unfortunately, in our series, we did not have access to more matched tumor samples, which would be crucial to further define the CMA's potential role in cancer.

We further found that CMA-deficient sarcoma tumors displayed higher levels of PPP metabolites with boosted activity of glucose-6-phosphate dehydrogenase (G6PDH), the rate-limiting enzyme of the PPP. In line with studies showing that melanoma cells are more dependent on G6PDH during metastasis (Aurora et al, 2022), these findings suggest not only a new metabolic link between the PPP and CMA, but also a new role of PPP in the aggressiveness of certain tumors. Yet, a direct link between TGFβ signaling and the PPP has not been described in cancer (Liu and Chen, 2022). However, in human aortic endothelial cells, G6PDH deficiency has been shown to activate endothelial cell and leukocyte adhesion via TGFβ/NADPH oxidases/ROS signaling (Parsanathan and Jain, 2020), representing a possible link between G6PDH, and thereby the PPP, with TGFβ. Moreover, a recent study observed that loss of G6PDH activity occurs in cancers with activated NRF2, a transcription factor reported to upregulate LAMP-2A levels (Ding et al, 2021). Thus, there seems to be an inverse correlation between *LAMP-2A* expression and CMA with G6PDH activity. Mechanistically, our study identified a direct or indirect crosstalk between TGFβ and the PPP, which contributes to tumor growth, presumably through effects on tumor cells proliferation and stimulation of EMT. Therefore, it is plausible that the anti-tumor effect of CMA can be a consequence of cooperative inhibition of these pathways.

In summary, our study provides a mechanistic explanation of CMA's role in cancer progression and dissemination, and the ability of CMA to target pro-metastatic proteins. It also provides a mechanistic description of CMA's tumor-suppressive nature by connecting two important oncogenic pathways, TGFβ signaling and the PPP, to LAMP-2A loss-of-function. Many sarcoma subtypes exhibit an aggressive clinical phenotype characterized by early metastasis and frequent relapse, which is associated with unfavorable clinical outcomes. As pro-metastatic proteins remain largely undetectable in healthy tissues, yet frequently accumulate in advanced cancers, they allow for a therapeutic window of CMA activation. In conclusion, our study unveils an important role for CMA in EMT and promoting tumor cell growth, which may contribute to exploration of the mechanism of CMA in cancer and propose a novel approach for targeting EMT. By activating this selective degradative process, we present how CMA could exhaust the metastatic potential in mesenchymal cells as an attractive anti-metastatic approach, likely to have implications on a wide range of human cancers that can metastasize.

## Methods

**Reagents and tools table**

| Reagent/Resource | Reference or Source | Identifier or Catalog Number |
|---|---|---|
| **Experimental Models** | | |
| Balb/cAnN-Foxn1nu/nuRj | Janvier (France) | |
| ES2 | ATCC | CRL-1978 |
| OVPA8 | DSMZ | ACC 871 |
| OVCAR3 | ATCC | HTB-161 |
| SKOV3 | ATCC | HTB-77 |
| HT1080 | ATCC | CCL-121 |
| A549 | ATCC | CCL-185 |
| NCI-H2087 | ATCC | CRL-5922 |
| HEK293FT | Thermo Fisher Scientific | R70007 |
| **Recombinant DNA** | | |
| pCMV6-XL5-LAMP2 | Origene | SC118738 |
| pSIN-PAmCherry-KFERQ-NE | Addgene | 102365 |
| **Antibodies** | | |
| Primary antibodies | | Appendix table S3 |
| Secondary antibodies | | Appendix table S3 |
| **Oligonucleotides and other sequence-based reagents** | | |
| sgRNA | (Yu et al, 2022) | |
| Genotyping primer | (Yu et al, 2022) | |
| siRNA non-targeting | Dharmacon | D-001810-01-20 |
| siRNA targeting LAMP-2A | GenePharma (Shanghai) | 1330 |
| siRNA targeting TGFβR2 #1 | Invitrogen | s14077 |

| Reagent/Resource | Reference or Source | Identifier or Catalog Number |
|---|---|---|
| siRNA targeting TGFβR2 #2 | Invitrogen | s14079 |
| PCR primer | This study | Appendix table S2 |
| **Chemicals, Enzymes and other reagents** | | |
| Lysosome Isolation Kit | Sigma-Aldrich | LYSISO1 |
| RNeasy FFPE kit | Qiagen, Hilden, Germany | 73504 |
| Haematoxylin | Thermo Fisher Scientific | 6765009 |
| Eosin | Sigma-Aldrich | HT110116 |
| Pertex | Histolab | 00801 |
| Matrigel | Corning | 356231 |
| Corn oil | MedChemExpress | HY-Y1888 |
| Click-It EdU Alexa Fluor 488 Kit | Thermo Fisher Scientific | A10044 |
| Dil cell-labeling solution | Thermo Fisher Scientific | V-22885 |
| 40 μm cell strainer | SARSTEDT | 83.3945.040 |
| Polyvinylpyrrolidone | Sigma-Aldrich | 81430 |
| Dapi | Thermo Fisher Scientific | D1306 |
| Vectashield | Vector Laboratories | H-1000-10 |
| Vectashield with DAPI | Vector Laboratories | H-1200-10 |
| Citrate buffer | Merck | C9999 |
| Bovine Serum Albumin | Sigma-Aldrich | A4503 |
| Normal donkey serum | Jackson Immunoresearch | 017-000-121 |
| DAB chromogen | Dako | K3468 |
| RPMI | Sigma-Aldrich | R8758 |
| DMEM + F12 | Gibco | 11320033 |
| FBS | Gibco | 10500064 |
| Penicillin/streptomycin | Sigma-Aldrich | P0781 |
| Glutamine | Sigma-Aldrich | G7513 |
| AC220 | Selleckchem | S1526 |
| Spautin-1 | Sigma-Aldrich | SML0440 |
| Chloroquine | Sigma-Aldrich | C6628 |
| Tranilast | Selleckchem | S1439 |
| 6-aminonicotinamide (in vivo) | Sigma-Aldrich | A68203 |
| 6-aminonicotinamide (in vitro) | Abcam | ab18529 |
| TGFβ1 | Thermo Fisher Scientific | 100-21 |
| 1,2-13C2-glucose | Cambridge Isotope Laboratories, Inc. | CLM-504-0.25 |
| Protease inhibitor cocktail | Roche | 04693124001 |
| PhosSTOP | Roche | 04906837001 |
| BCA Protein Assay Kits | Thermo Fisher Scientific | 23225 |
| ViaFect | Promega | E4982 |
| Lipofectamine 2000 | Invitrogen | 11668019 |
| RNAqueous phenol-free total RNA isolation kit | Invitrogen | AM1912 |

| Reagent/Resource | Reference or Source | Identifier or Catalog Number |
|---|---|---|
| IScript cDNA Synthesis Kit | Bio-Rad | 1708890 |
| Maxima qPCR SYBR Green Master Mix | Thermo Fisher Scientific | K0222 |
| PureLink RNA Mini Kit | Invitrogen | 12183025 |
| RNA 6000 Nano Kit | Agilent | 5067-1511 |
| Base medium | Seahorse Bioscience | 102353-100 |
| XF Cell Mito Stress Test Kit | Seahorse Bioscience | 103010-100 |
| G6PDH Assay Kit | Abcam | ab102529 |
| ROS-Glo H2O2 assay | Promega | G8820 |
| **Software** | | |
| R (v 4.0.5) | https://www.r-project.org/ | |
| tximport package (v 1.26.1) | https://bioconductor.org/packages/release/bioc/html/tximport.html | |
| clusterProfiler package (v 4.6.2) | https://bioconductor.org/packages/release/bioc/html/clusterProfiler.html | |
| DESeq2 package (v 1.38.3) | https://bioconductor.org/packages/release/bioc/html/DESeq2.html | |
| Salmon tool | https://combine-lab.github.io/salmon/ | |
| Limma package (v 3.56.2) | https://bioconductor.org/packages/release/bioc/html/limma.html | |
| dbEMT 2.0 | http://dbemt.bioinfo-minzhao.org/index.html | |
| KFERQ finder | https://rshine.einsteinmed.edu/ | |
| uniprot | https://www.uniprot.org/ | |
| ImageJ | https://imagej.net/ij/ | |
| Wave | Agilent Technology | |
| GraphPad Prism (v 9.0) | https://www.graphpad.com/ | |
| Biorender | https://app.biorender.com | |
| **Other** | | |
| 2200 TapeStation system | (Agilent, Santa Clara, CA, USA) | |
| Novaseq PE150 platform | Illumina | |
| Decloaking chamber | Biocare Medical | |
| Autostainer 480 instrument | Thermo Fisher Scientific | |
| Leica Biosystems | Deer Park, IL, USA | |
| Zeiss Axio Imager M2 microscope with Axio Cam HR camera | Zeiss | |
| ZEISS LSM 710 inverted confocal microscope | Zeiss | |
| Eclipse Si inverted microscope with DS-Fi1 camera | Nikon | |
| SuperFrost glass slides | Avantor | 631-0108 |

| Reagent/Resource | Reference or Source | Identifier or Catalog Number |
|---|---|---|
| μ-Plate 96 Well Square | Ibidi | 89621 |
| HP G4010 scanner | HP | |
| Mixer Mill MM 400 | Retsch | |
| 7500 Real-Time PCR system | Applied Biosystems, Foster City, CA, USA | |
| Extracellular Flux Analyzer | Seahorse Bioscience, MA, USA | |
| GloMax Discover Microplate Reader | Promega, Madison, WI, USA | |
| CE-TOF MS | Human Metabolome Technologies, Boston, MA, USA | |

## Ethics statement

Human samples of primary lung tumors and matched brain metastases (FPPE tissue sections) were used, and the study received approval from the regional ethical committee at Karolinska University Hospital and Stockholm Bioabank (permit number: 2016/944-31/1, Bbk 01605) as well as Uppsala University Hospital and Uppsala Biobank (permit number: 2006/325, 2012/352, BbA-827-2018-058). At the time of data analysis and sample collection, all patients were deceased. It was considered by us and approved by the ethical review board that contacting the family members of the deceased would cause more harm than good by risking evoking painful memories/traumas or rekindle grief. In addition, the retrospective and population-based nature of our study without risk of revealing individual patient identities was a further motivation to refrain from the consent process. Therefore, no informed consent was deemed necessary as judged by the previously mentioned authorities. We confirm that the experiments conformed to the principles in the WMA Declaration of Helsinki and the Department of Health and Human Services Belmont Report. All animal experiments were conducted in accordance with Swedish animal welfare regulations and were authorized by the Stockholm Animal Ethics Committee with permit numbers: N116/16 and 1185-2022 for xenograft experiments, 14049-2019 (zebrafish invasion), and 11075-2024 (experimental metastatic mouse model). All experiments using animals followed protocols approved by the Karolinska Institutet and Karolinska University Hospital.

## Lysosomal fractionation and LC-MS/MS

Isolation of lysosomal fractions was performed from ES2 cancer cells under three conditions: control (DMSO), 16 h CMA activation (1.5 μM AC220 + 10 μM Spautin1), and 16 h CMA activation following genetic *LAMP-2A* depletion (siRNA) for 48 h. The procedure was performed according to the manufacturer of the Lysosome Isolation Kit. Gradient fractionation was conducted and low-percent fractions with high lysosome content were subjected to LC-MS/MS for a multiplexed quantitative mass spectrometry (MS)-based proteomics analysis at the EMBL proteomics Core Facility, Heidelberg, Germany, as previously described (Hao et al, 2019). The full raw data were first log2 transformed and then batch

cleaned by applying the limma package in R to fit a linear model (Ritchie et al, 2015). For distribution normalization based on total protein, the quadruplicate data from different sample group sets were normalized using the Quantile method (Ritchie et al, 2015). The inter-experiment normalization was then performed using a selected lysosomal protein, LAMP1, across the sample sets as unchanged signal through the experiment. The quality of normalized data was checked by PCA. The fold change and statistical significance between different groups were calculated over the control using the limma package. Proteins with adjusted *P*-value less than <0.01 and a positive fold change of >1.3 over the control were considered for further analysis. For the *LAMP-2A* siRNA samples analysis, proteins meeting the above criteria (1.3-fold enrichment upon CMA activation with an adjusted *P*-value < 0.01) and additionally displaying at least a 30% reduced lysosomal enrichment upon LAMP-2A depletion with adjusted *P*-value < 0.01 were considered as LAMP-2A dependent.

## CMA motif search

The total of 1184 human EMT genes were obtained from the dbEMT 2.0 database. These genes were selected for their involvement in the regulation of EMT and epithelial cell proliferation/migration selected by 13 Gene Ontology (GO) terms (GO:0001837, GO:0010719, GO:0010717, GO:0010718, GO:0050678, GO:0050673, GO:0090132, GO:0010631, GO:0010632, GO:0050679, GO:0010634, GO:0050680, GO:0010633), and 3 Wiki pathways (WP4239, WP3859, WP366). The overlapping genes among these terms and pathways were selected with all microRNAs removed; thereby 295 genes were subjected to functional analysis in Uniprot. Among these, EMT-promoting genes ($n = 211$) were searched for CMA motif using the database KFERQ finder V0.8 (Kirchner et al, 2019).

## Human clinical samples

The human clinical material was obtained from patients with NSCLC, that underwent surgery of the primary tumor and brain metastases. Total RNA was isolated from formalin-fixed, paraffin-embedded (FFPE) $4 \times 4$ μm tissue sections of primary tumors and matched brain metastases using RNeasy FFPE kit per manufacturers' instructions. RNA quantity and quality were assessed using RNA Screen Tapes on a 2200 TapeStation system. All tissue samples displayed similar RNA integrity number curves. Systematic mRNA expression was measured on an nCounter FLEX™ Analysis System (nanoString, Seattle, WA, USA) using the nCounter® PanCancer Immune Profiling gene expression panel (nanoString Technologies Inc.) and a minimum of 150 ng of total RNA input for each sample.

Human multiorgan metastatic tissue microarrays (TMA) representing metastases from carcinoma across 19 different organs were purchased from Biocat: MT2081, comprising of 104 cases/208 cores, and MT801, comprising of 80 cases/80 cores. Both TMAs contain metastatic carcinoma tissues with confirmed pathological diagnoses. The TMA analysis, immunohistochemistry (IHC) and slide scanning were performed as described before (Kampf et al, 2012; Zhang et al, 2017). The sections were deparaffinized in xylene, hydrated in graded alcohols, and blocked for endogenous peroxidase in 0.3% hydrogen peroxide diluted in 95% ethanol. A Decloaking chamber was used for antigen retrieval.

Immunohistochemical staining of Vimentin and LAMP-2A was performed using an Autostainer 480 instrument, incubating the slides with anti-Vimentin and anti-LAMP-2A antibodies, and developed for 10 min using diaminobenzidine Quanto as chromogen. All incubations were followed by rinsing in wash buffer 2 × 5 min. Slides were counterstained in Mayers hematoxylin and cover slipped using Pertex mounting medium. The stained slides were digitalized using the automated scanning system Leica Biosystems using a ×20 objective.

## Cell culture

The ovarian cancer cell lines OVCAR3, OVPA8, SKOV3, ES2; the fibrosarcoma cell line HT1080, and the lung cancer cell lines A549 and NCI-H2087 were maintained in RPMI medium with 10% (v:v) heat-inactivated fetal bovine serum (FBS), 1% [v/v] penicillin/streptomycin, and 1% (v:v) glutamine. The ovarian cancer cell line FUOV1 was cultured in DMEM + F12 (1:1) medium, supplemented with 10% (v:v) FBS and 1% [v/v] penicillin/streptomycin. All cells obtained from ATCC were authenticated by SRT profile and cultured at 37 °C in a 5% $CO_2$ humidified incubator and tested regularly for mycoplasma contamination. Isoform specific LAMP-2A knockout (KO) A549, HT1080, ES2, and NCI-H2087 cancer cells were generated by CRISPR-Cas9 gene-editing as described previously (Yu et al, 2022). Validation of the KO cells was confirmed through genotyping, RT-qPCR, immunoblotting, IF, and IHC analyses (see Reagent and Tools table). Throughout the experiments, cells were treated with 1.5 μM AC220 and 10 μM Spautin1 for 16 h to induce CMA as previously described (Hao et al, 2019). The following compounds were used to treat cells in the indicated experiments: 50 μM chloroquine (CQ), 50 μM Tranilast, 100 nM 6-Aminonicotinamide (6-AN), and 10 ng/mL TGFβ1.

## In vivo animal experiments

Female Balb/c Nude mice (Balb/cAnN-Foxn1$^{nu/nu}$Rj) were purchased from Janvier (France) and housed under specific pathogen-free conditions in individually ventilated cages. Mice were housed under standard laboratory conditions (22 °C, 14/10 h light/dark cycle) with unrestricted access to drinking water and a standard chow diet. For xenograft tumor growth, 1 × 10$^6$ mycoplasma-negative cells (WT and LAMP-2A KO A549 and HT080) were xenografted subcutaneously into the flank of 7- or 8-week-old BALB/c nude mice. The total volume of one injection was 50 μL in 50% Matrigel, and 4 injections (2 × 2 symmetrical) were made per mouse. For Tranilast and 6-AN treatment, 1 × 10$^6$ WT and LAMP-2A KO HT080 cells were mixed with 50% Matrigel and then xenografted subcutaneously into the flank of BALB/c nude female mice (2 injections per mouse). The 18 mice were randomly divided into 6 groups: WT-Control, WT-Tranilast, WT-6-AN, KO-Control, KO-Tranilast, KO-6-AN. A concentration of 100 mg/kg Tranilast or 10 mg/kg 6-AN was injected intraperitoneally (i.p.) once a day from day 3 post implantation. Untreated mice were injected with corn oil to serve as the control. During the experiments, tumor growth was measured daily using a digital caliper. Mice were sacrificed when either at least one tumor or the total volume of all tumors in one mouse had reached the maximum allowed size (1.5 cm$^3$) or (2 cm$^3$), respectively, as determined by the caliper using the formula (3.14 × tumor length × tumor width$^2$)/6 or if ulceration was observed. Additionally,

Kaplan–Meier survival curve analysis was conducted using GraphPad Prism 9, with survival defined based on tumor size reaching the endpoint criteria. Tumors were separated from the skin and accurately weighed. Fresh tumor tissues were processed for RNA isolation, IHC and immunofluorescence (IF) staining. The remaining tissues were snap-frozen in liquid nitrogen and stored at −80 °C until further use. For EdU labeling, the mice were injected i.p. with 100 μL EdU (1 mg/mL) 2 h before sacrifice. Tumor sections were prepared according to the IHC protocol. EdU was detected by the Click-It EdU Alexa Fluor 488 Kit. All samples were imaged with a Zeiss Axio Imager M2 microscope with an Axio Cam HR camera. ImageJ was used for analysis.

For the experimental metastasis assay, 20 mice were randomly divided into 4 groups (n = 5 per group): HT1080 WT, HT1080 LAMP-2A KO, A549 WT, and A549 LAMP-2A KO. Mycoplasma-negative cells were collected during exponential growth phase and 3 × 10$^6$ cells were injected in 100 μL of sterile PBS (pH 7.4) solution into the lateral tail veins of 5-week-old female (Balb/cAnN-Foxn1$^{nu/nu}$Rj) mice using 27G needles, one injection per mouse. The mice health was regularly monitored during the experiment. 21 days post injection, mice were sacrificed and major organs (lungs, spleen, kidneys with adrenal glands, and liver) were harvested and fixed in 4% paraformaldehyde (PFA) for further analysis. Lungs surface metastases were counted, and organs were photographed for documentation before paraffin embedding. Embedded tissues were cut into 5 μm-thick sections and mounted on SuperFrost glass slides. Sections were deparaffinized and rehydrated by sequential immersion in hydrated in graded ethanol. Tissue-embedded slides were stained with hematoxylin and eosin (H&E). After dehydration by sequential immersion in 95 and 99% ethanol, slides were mounted with Pertex mounting media. H&E images were captured using Eclipse Si Nikon inverted microscope equipped with DS-Fi1 Nikon camera.

No mice were excluded for this study and all established xenografts were included to the analysis. Blinding was not possible due to the mice tagging and identification codes accessible by the investigators.

For the zebrafish embryonic invasion and metastasis assay, WT and LAMP-2A KO HT1080 cells were incubated with DiI cell-labeling solution for 20 min at 37 °C. To remove the unincorporated dye and make a single-cell suspension, cells were rinsed twice with PBS, passed over a 40 μm cell strainer, and resuspended in 1.5% polyvinylpyrrolidone. Zebrafish embryos were staged according to Kimmel et al (Kimmel et al, 1995). For xenotransplantation, zebrafish embryos are raised up to 2 days post fertilization in E3 water containing 30 mg/L phenylthiourea (PTU) before cell injection. Approximately 100–300 labeled cells were injected into the perivitelline cavity of each embryo. After transplantation, embryos were screened for successful transplantation (no cells leaking into the circulation or the yolk sack) and incubated at 33 °C for 48 h. For high-throughput imaging, an imaging 96-well plate was prepared with 250 μL of 1% agarose in E3 medium per well. Then, single embryos were distributed into the wells of the 96-well imaging plate together with 150 μL of exposure medium (160 μg/mL tricaine, 30 μg/mL PTU in E3 medium). Embryos were manually oriented into position and imaged. The images were processed using ImageJ.

## Histology and microscopy

Cells were fixed for 15 min in 4% formaldehyde and permeabilized with digitonin (0.025%, 10 min at RT). Cells were stained overnight at 4 °C using primary antibodies (Appendix Table S3) diluted in 2%

BSA and PBS. The next day, the samples were washed followed by staining with the secondary Alexa Fluor Plus 647 goat anti-rabbit IgG or Alexa Fluor Plus 488 goat anti-mouse IgG for 1 h at RT. Nuclei were counterstained with DAPI (1 mg/mL) for 10 min at RT and slides were mounted using Vectashield mounting medium. The pictures were taken using ZEISS LSM 710 inverted confocal microscope and the 40x/1.2 W Corr objective. WT and *LAMP-2A* KO tumors from BALB/c nude mice were fixed in 4% PFA overnight at 4 °C, embedded in paraffin according to the routine procedure, and 5-μm-thick sections were used for staining. IHC and IF staining were performed according to standard procedures. Briefly, sections were deparaffinized followed by antigen retrieval for 10 min in sodium citrate buffer pH 6.5 in a steamer. Sections were permeabilized and blocked for 60 min in 1% BSA, 5% normal donkey serum followed by incubation with the primary antibodies (Appendix Table S3) diluted in 1% BSA in PBS-Triton 0.1% at 4 °C overnight. Secondary antibodies with the correspondent species specificity were used for the signal detection: donkey anti-rabbit Alexa Fluor Cy3 and 488 conjugated secondary antibodies for IF, and HRP-conjugated anti-rabbit antibody followed by DAB chromogen detection for IHC. All samples were imaged with a Zeiss Axio Imager M2 microscope with an Axio Cam HR camera. ImageJ was used for image analysis.

### Cell proliferation and clonogenicity

Cells were seeded at a density of 3000 cells/well in their respective culture medium in 6-well plates. After seeding, the number of cells was determined every day using a hemocytometer and a light microscope. To metabolically challenge the cells, they were grown in medium with either 10% galactose and 0.1 μM rotenone and cell numbers were counted every day for up to 5 days. Cells were cultured in triplicates in 6-well plates at 200 cells per well and allowed to form colonies over 8 days. Colonies were fixed with 1% PFA and stained with 0.1% Crystal Violet solution. After washing and drying, the plates were imaged by a scanner and individual colonies were manually counted.

### Metabolomics and isotopomer analysis

Targeted quantitative metabolomics was performed on WT and *LAMP-2A* KO HT1080 xenografted tumors (25 mg of tissue) to measure the levels of intracellular metabolites using capillary electrophoresis-time of flight mass spectrometry (CE-TOF MS and CE-QqQMS) in the cation and anion modes relative to internal standards. To measure the levels of intracellular metabolites, extracts were prepared from $5 \times 10^6$ cells per sample in three biological replicates and analyzed using a capillary electrophoresis (CE)-connected ESI-TOF MS system as previously described (Krassikova et al, 2021). To trace a potential glucose-derived carbon flux into the pentose phosphate pathway, cells were washed three times with PBS and incubated at 37 °C in glucose-free RPMI medium supplemented with 10% dialyzed FBS, 4 mM glutamine, and 10 mM 1,2-$^{13}C_2$-glucose for 2 h. Intracellular metabolites were extracted with methanol following Human Metabolome (HMT) metabolite extraction protocol for adherent cells (Kochetkova et al, 2022). Briefly, 5% mannitol was used to wash cells three times and metabolites were extracted in 1.3 mL methanol containing 10 μM Internal Standard Solution (Human Metabolome Technologies). The extracted metabolites were centrifuged at $2300 \times g$ at 4 °C for 5 min. The supernatants were filtered

through a Millipore 5-kDa cutoff filter to remove proteins at $9100 \times g$ at 4 °C for 3 h. The filtrate was lyophilized, suspended in Milli-Q water and analyzed using CE-TOF MS as described previously (Krassikova et al, 2021; Soga et al, 2003; Sugimoto et al, 2010).

### Western blot

Cells were harvested and lysed using the RIPA lysis buffer supplemented with protease and phosphatase inhibitor cocktails. Tumor tissues were homogenized in the same lysis buffer using a tissue lyser and steel beads for 5 min at 25 Hz, and then centrifuged at 10,000 rpm for 10 min. Concentrations in the lysates were determined using the bicinchoninic acid (BCA) assay and equal amounts of protein were mixed with Laemmli loading buffer and boiled for 5 min at 95 °C. Total protein extracts were separated in 10, 12 or 15% acrylamide gels, transferred to nitrocellulose membranes, and the membranes were blocked for 1 h in PBS – 0.1% Tween 20 with 5% skimmed milk and subsequently incubated overnight with the primary antibody (Appendix Table S3) at 4 °C. After washing and incubation with the secondary antibody, the recognized proteins were detected by enhanced chemiluminescence (ECL) substrate on X-ray film.

### Plasmids and siRNAs

The siRNA (see Reagent and Tools Table) transfections were performed using the Lipofectamine 2000 Reagent according to the manufacturer's instructions. Plasmid pCMV6-XL5-LAMP-2 was transfected with ViaFect according to the manufacturer's recommended procedure. The efficiency of siRNA-mediated knockdown and expression level of proteins was monitored at 48–72 h post-transfection by qPCR or Western blotting.

### RNA isolation and qPCR

The total RNA was isolated from cells and tissues using the RNAqueous phenol-free total RNA isolation kit according to the manufacturer's instructions. A total of 1 μg RNA was used for cDNA synthesis with the iScript cDNA Synthesis Kit, according to the manufacturer's instructions. The cDNA samples were analyzed by quantitative PCR using Maxima qPCR SYBR Green Master Mix and amplified using the 7500 Real-Time PCR system. The ΔΔCt method was used to calculate the relative mRNA levels after normalization with the housekeeping gene (ACTB or HPRT). The following primers were used: see Appendix Table S2.

### RNA-seq analysis

RNA was extracted from WT and *LAMP-2A* KO tumors from BALB/c nude mice with PureLink RNA Mini Kit and the RNA quality was determined with the RNA 6000 Nano Kit. For each sample, 200 ng of RNA was submitted to Novogene Corporation Inc. (Cambridge, UK) for cDNA library construction and 150-bp paired-end sequencing using the Novaseq PE150 platform. The mRNA sequencing was performed at Novogene on the NovaSeq PE150 platform following poly A enrichment. The transcript variants for *LAMP2* were identified using the Salmon software tool (Patro et al, 2017). Based on Ensembl's human genome sequence GRCh38 gene annotations, reads were mapped to the human transcriptome at the transcript level. To do gene level analysis, the

tximport package (v 1.26.1) was used to add up the read counts associated with each transcript in each sample. The resulting matrix of read counts was analyzed using R. Differentially expressed genes were identified using the DESeq2 package in R (v 1.38.3), the Gene Ontology categories of the significant differentially expressed genes were determined using the clusterProfiler package (v 4.6.2).

## CMA reporter assay

HEK293FT cells were transfected with pSIN-PAmCherry-KFERQ-NE, pLP1, pLP2, and pLP VSV-G in a 10 cm cell culture dish for lentivirus production. Fresh medium was added to replace the used medium 24 h after transfection. Two days later, medium was collected and filtered through a 0.45 μm filter, then stored at −80 °C for future viral transduction. To generate stable cells expressing PAmCherry-KFERQ-NE, cells were seeded in 6-well plate. On the next day, 1 mL of the lentivirus solution was added to the cells for transduction. At 48 h post transduction, the medium was replaced by fresh medium containing puromycin (1 μg/mL) for 5 days, whereafter cells were used for further analysis. Cells stably expressing PAmCherry-KFERQ-NE were seeded on sterilized, round cover slips in 6-well plates. The next day, the cells were treated with either 50 μM Tranilast or 1.5 μM AC220 and 10 μM Spautin1 to induce CMA for 16 h. Post treatment, the cells were exposed to UV light (365 nm) for 45 min to photoactivate PAmCherry fluorescence. Then, cells were fixed in 4% PFA for 15 min. After washing with PBS, cells were mounted using VECTASHIELD antifade mounting medium with DAPI. The samples were analyzed with a Zeiss LSM700 confocal microscope. For each biological replicate ($n = 3$), at least 4 images were analyzed with ImageJ.

## Mitochondrial oxygen consumption rate

Mitochondrial oxygen consumption rate (OCR) was measured in real-time using the XFp Extracellular Flux Analyzer. Cells were seeded in 6-well plates at density of 80,000 cells/well and treated with 50 μM Tranilast for 6 h. Then, the cells were transferred to an XFp miniplate at a density of 8000 cells/well in their respective growth medium (supplied with the same concentration of Tranilast for Tranilast-treated cells). On the next day, cells were washed twice with 100 μL of XF Base medium, supplied with 1 mM pyruvate, 2 mM glutamine, and 10 mM glucose, and left in 180 μL of medium for 45 min. Cells were subjected to analysis using XF Cell Mito Stress Test Kit, following the manufacturer's instructions. After baseline measurement, three following subsequent injections were made: 1 μM oligomycin (a mitochondrial ATP synthase inhibitor), 0.5 μM FCCP (a mitochondrial uncoupler), and 0.5 μM Rotenone/Antimycin A (Complex I/Complex III inhibitors). Wave and GraphPad Prism 9 software were used to analyze the obtained data.

## Analysis of G6PDH enzymatic activity

Cells were treated with 50 μM Tranilast for 24 h, and $1 \times 10^6$ cells per condition were lysed in ice-cold PBS. After centrifugation at $12,000 \times g$ for 5 min at 4 °C, the supernatant was collected for further analysis. G6PDH enzymatic activity was then determined using the colorimetric G6PDH Assay Kit according to manufacturer's instructions. Absorbance was measured at 450 nm using a microplate reader at 37 °C between 2 to 60 min. The enzymatic activities were normalized to protein concentrations, as determined by BCA assay.

### The paper explained

**Problem**

Despite considerable advances in drug discovery, clinical success has proven challenging in preventing the spread of cancer. Tumors often recapitulate programs to acquire advantageous invasive and dissemination ability, during which pro-metastatic proteins are distinctively stabilized in cancer cells to drive and sustain cancer progression. To date, the most studied aspect of metastatic reprogramming events is genetic and transcriptional regulation. However, whether failure in accurate clearance and inefficient protein degradation can control metastatic potential and drive cancer progression remains largely unexplored.

**Results**

In this study, we determined that CMA, a selective protein degradative process, contributes to the removal of pro-metastatic proteins, thereby decreasing the mesenchymal traits of cancer cells to exhaust their migratory potential. The lack of CMA promotes the growth and dissemination of mesenchymal tumors by TGFβ-driven EMT.

**Impact**

This study proposes that CMA can exhaust the metastatic potential in mesenchymal cancer cells as an anti-cancer pathway, likely to have implications on human cancers that can metastasize.

## Analysis of ROS level

The ROS level was determined by the ROS-Glo $H_2O_2$ assay according to the manufacturer's instructions. The HT1080 WT and *LAMP-2A* KO cells were plated at a density of $1 \times 10^4$ cells/100 μL into each well of a 96-well plate. After 12 h, cells were incubated with $H_2O_2$ substrate solution for 6 h, whereafter the ROS-Glo detection solution was added. The luminescence was measured by a GloMax Discover Microplate Reader.

## Statistics

Statistical analysis was performed using GraphPad Prism (v 9.0). The data were presented as Error bars, means ± SD (standard deviation) or otherwise stated, from a minimum of three independent experiments. For the comparison of two sets of data, if they conform to a normal distribution, a two-tailed unpaired Student's *t* test was employed. For comparisons involving more than two groups, a one-way ANOVA was performed. A $P$ value (or adjusted $P$ value) was considered statistically significant indicated in the corresponding figures as \*$P < 0.05$, \*\*$P < 0.01$, \*\*\*$P < 0.001$ and with exact $P$ values stated in the figure legends. Over-representation analysis (ORA) was performed using the cluster-Profiler R package (v4.14.4) to conduct the GO analysis.

# Data availability

Proteomics data were deposited in ProteomeXchange via the PRIDE database platform with accession number PXD058467. UTR for the MS data: https://www.ebi.ac.uk/pride/archive/projects/PXD058467. RNA sequencing data were deposited in the ArrayExpress platform with accession number E-MTAB-14665.

UTR for the RNAseq data: https://www.ebi.ac.uk/biostudies/arrayexpress/studies/E-MTAB-14665?query=E-MTAB-14665.

The source data of this paper are collected in the following database record: biostudies:S-SCDT-10_1038-S44321-025-00210-w.

# Peer review information

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

## Acknowledgements

We are grateful to Dr. Alex Buko, Human Metabolome Technologies for the tumor metabolome analyses, Dr. Simon Stritt, Novogene, for the RNA sequencing, and Dr. Lars Bräutigam at the zebrafish core facility at Karolinska Institute and the animal veterinaries for their technical support with animal studies. We especially thank Dr. Yuqing Hao for the lysosomal fractionation and Ellen Olerup (Project student) for the technical assistance. This work was supported by grants from the Swedish Research Council no: 2019-01535 (HVN) and 2021-01787 (EN), the Swedish Cancer Society CAN 2017/466 (EN), CAN 2017/1015 (EN) and 20 0979 PjF (HVN).

## Author contributions

**Xun Zhou**: Data curation; Formal analysis; Validation; Investigation; Visualization; Methodology; Writing—original draft. **Eva Berenger**: Data curation; Formal analysis; Validation; Investigation; Visualization; Methodology; Writing—review and editing. **Yong Shi**: Data curation; Formal analysis; Validation; Investigation; Visualization. **Vera Shirokova**: Data curation; Investigation; Methodology. **Elena Kochetkova**: Data curation; Formal analysis; Validation; Investigation; Visualization; Writing—review and editing. **Tina Becirovic**: Formal analysis; Visualization; Writing—review and editing. **Boxi Zhang**: Software; Formal analysis; Visualization. **Vitaliy O Kaminskyy**: Data curation; Formal analysis; Validation; Investigation; Writing—review and editing. **Yashar Esmaeilian**: Data curation; Formal analysis. **Kayoko Hosaka**: Data curation; Visualization; Methodology. **Cecilia Lindskog**: Resources; Methodology. **Per Hydbring**: Resources; Formal analysis. **Simon Ekman**: Resources; Formal analysis. **Yihai Cao**: Resources. **Maria Genander**: Resources; Validation; Methodology. **Marcin Iwanicki**: Resources; Writing—review and editing. **Erik Norberg**: Resources; Supervision; Funding acquisition; Visualization; Methodology; Project administration; Writing—review and editing. **Helin Vakifahmetoglu-Norberg**: Conceptualization; Resources; Formal analysis; Supervision; Funding acquisition; Validation; Investigation; Visualization; Methodology; Writing—original draft; Project administration; Writing—review and editing.

Source data underlying figure panels in this paper may have individual authorship assigned. Where available, figure panel/source data authorship is

listed in the following database record: biostudies:S-SCDT-10_1038-S44321-025-00210-w.

## Funding

## Disclosure and competing interests statement

The authors declare no competing interests.

# Expanded View Figures

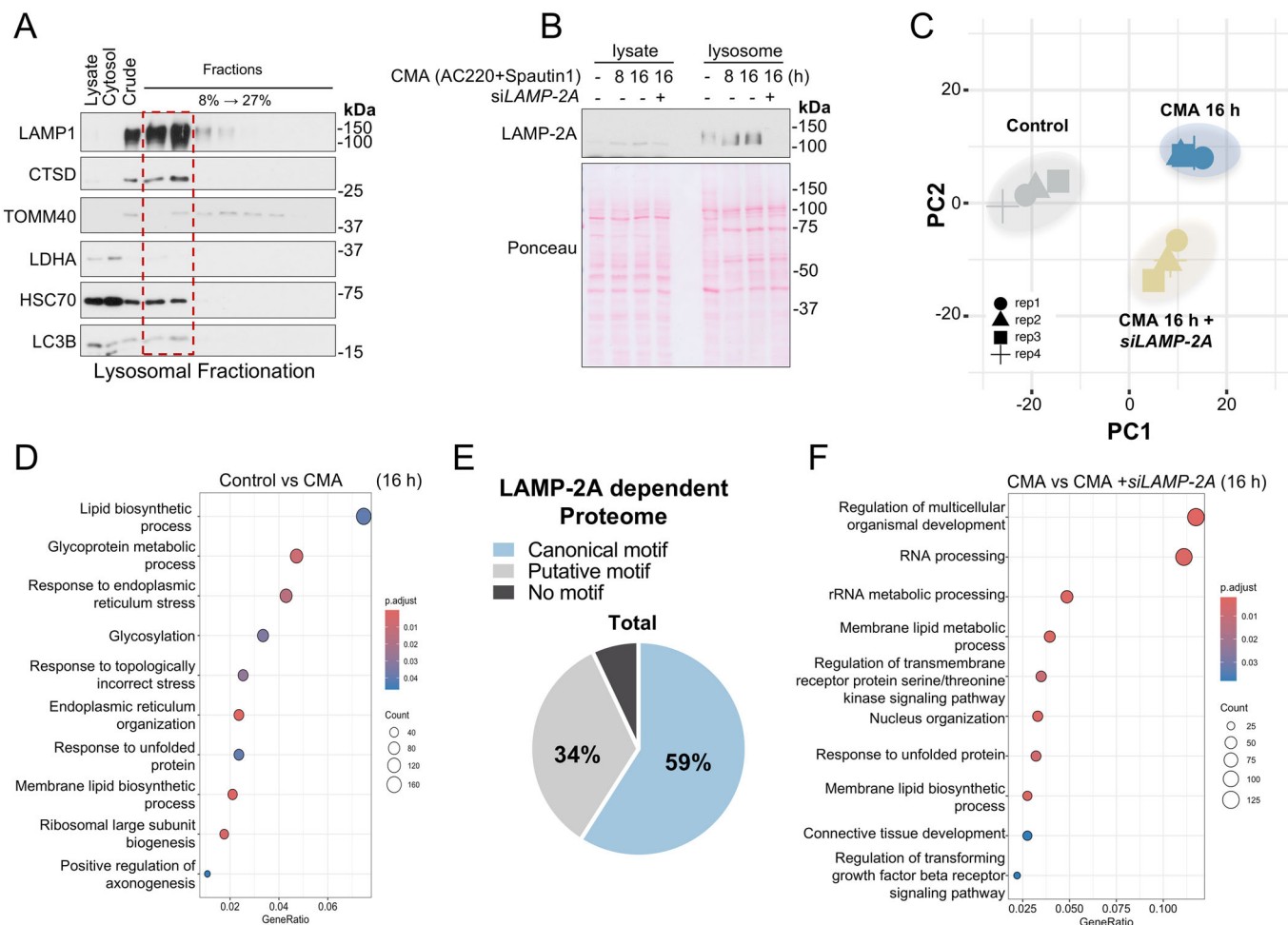

**Figure EV1. Lysosomal fractionation and quantitative proteomics of ES2 cells.**

(A) Cellular fractions isolated from ES2 cells by differential large-scale multi-layered density gradient centrifugations and analyzed by immunoblotting using antibodies against lysosomal membrane protein LAMP1, lysosomal matrix protein cathepsin D (CTSD), mitochondrial protein translocase of outer mitochondrial membrane 40 (TOMM40), cytosolic protein Lactate dehydrogenase A (LDHA), the Hsc70 chaperone protein and macroautophagy marker LC3B. The red square indicates the low percentage gradient fraction, enriched in lysosomes, and chosen for further analysis. (B) Cellular lysate and lysosomal fractions from control (DMSO), 8 and 16 h CMA activated (AC220+Spautin1) and 16 h CMA+si*LAMP-2A* conditions analyzed by immunoblotting using LAMP-2A antibody. Total protein loading is visualized by Ponceau S Red staining. (C) Score plot of principle component analysis (PCA) in 2D for control, 16 h CMA, and 16 h CMA+si*LAMP-2A* treated sample sets ($n_{exp}$ = 4). (D) Pathway enrichment analysis by Gene ontology (GO) of biological processes comparing control vs 16 h CMA treated sample sets (Over Representation Analysis). (E) Pie graphs of the experimentally validated LAMP-2A-dependent lysosome proteome showing the percentage of proteins with indicated types of KFERQ-like motifs. (F) GO analysis comparing 16 h CMA vs CMA+si*LAMP-2A* treated sample sets. Dot plots of top 10 enriched pathways are indicated as the ratio of the differentially expressed gene number to the total gene number for a certain annotation. The size and color of the dots represent the gene ratio and the range of adjusted *P* values, respectively (Over Representation Analysis). Source data are available online for this figure.

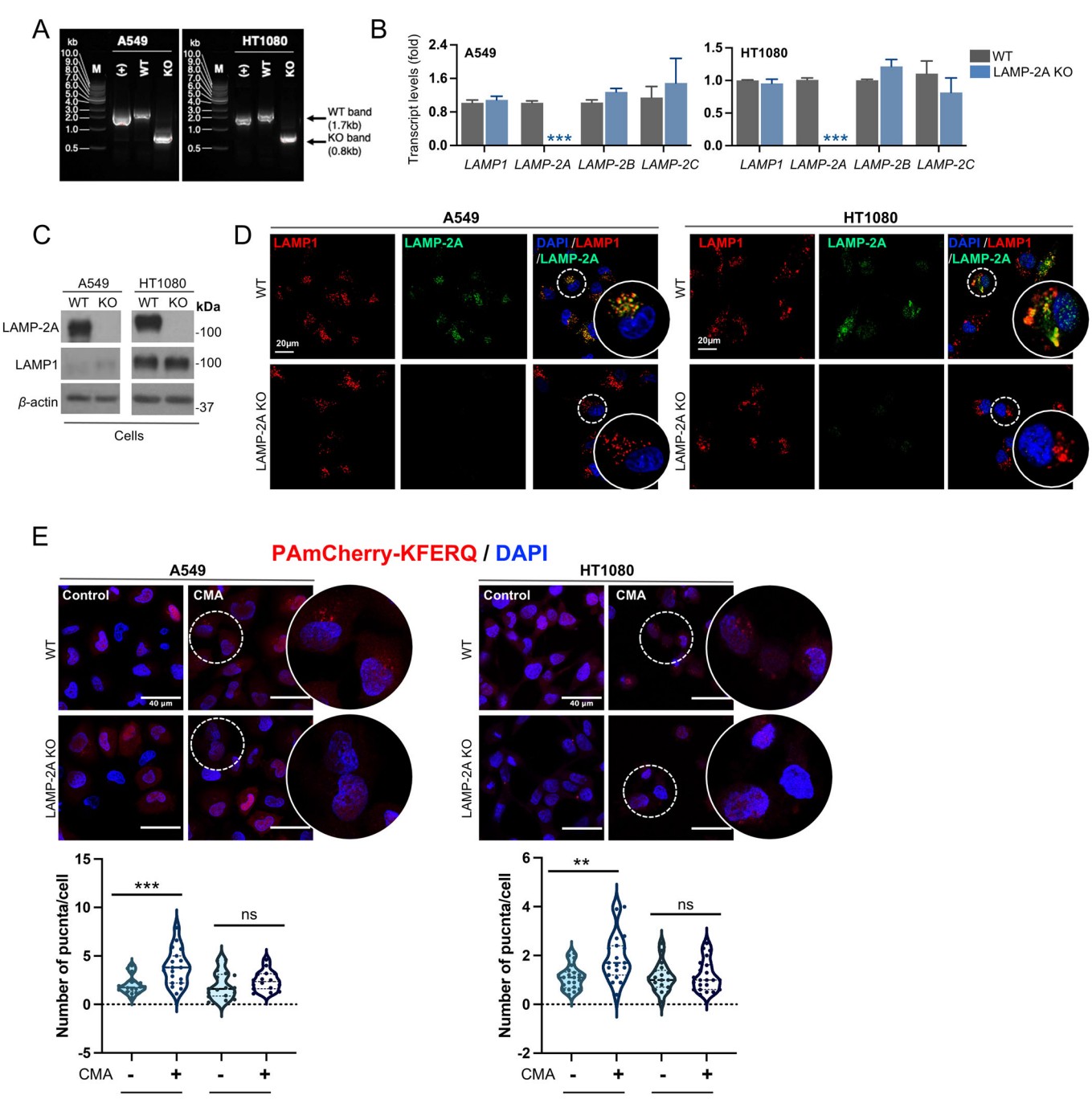

**Figure EV2. Generation and validation of isoform-specific knockout of *LAMP-2A*.**

(A) CRISPR/Cas9 genome editing and genotyping by agarose gel electrophoresis/PCR in regular (+), wild-type (WT), and *LAMP-2A* knockout (KO) A549 and HT1080 cells. Target amplicon in human *LAMP2* gene is detected by the fragment of ~800 bp. M: marker. (B) RT–qPCR detection of *LAMP1*, *LAMP-2A*, *-2B*, and *-2C* expression in WT and *LAMP-2A* KO A459 and HT1080 cell lines. Bars represent mean ± sd ($n_{exp}$ = 3). *P* values refer to ***$P$ < 0.001 (A549: $P_{LAMP-2A}$ = 0.0001; HT1080: $P_{LAMP-2A}$ < 0.0001 Student's *t*-test). (C) Immunoblot detection of LAMP-2A and LAMP1 in WT and *LAMP-2A* KO A549 and HT1080 cells. *β*-Actin: loading control. (D) Representative immunofluorescence confocal images of WT and *LAMP-2A* KO A549 and HT1080 cell lines stained with anti-LAMP-2A (green), anti-LAMP1 (red), and DAPI (blue) for nuclei highlighted with (circular insets) magnification. Scale bars: 20 μm. (E) CMA activity measured in WT and *LAMP-2A* KO A549 and HT1080 cells expressing PAmCherry-KFERQ CMA reporter. Cells were treated with AC220+Spautin1 to activate CMA for 16 h. Confocal images (upper panel) of merged red (PAmCherry-KFERQ) and blue (DAPI) for nuclei highlighted with (circular insets) magnification. Scale bars: 40 μm. Quantification (lower panel) of CMA activity as number of fluorescent puncta per cell. For each replicate ($n_{exp}$ = 3), at least four images were analyzed. Each dot in the graph represents the average number of puncta per cell. *P* values refer to ***$P$ < 0.001, ns: non-significant (A549: $P_{WT\ Ctrl\ vs\ CMA}$ = 0.0008, $P_{KO\ Ctrl\ vs\ CMA}$ = 0.28; HT1080: $P_{WT\ Ctrl\ vs\ CMA}$ = 0,0013, $P_{KO\ Ctrl\ vs\ CMA}$ = 0,75; Student's *t*-test). Source data are available online for this figure.

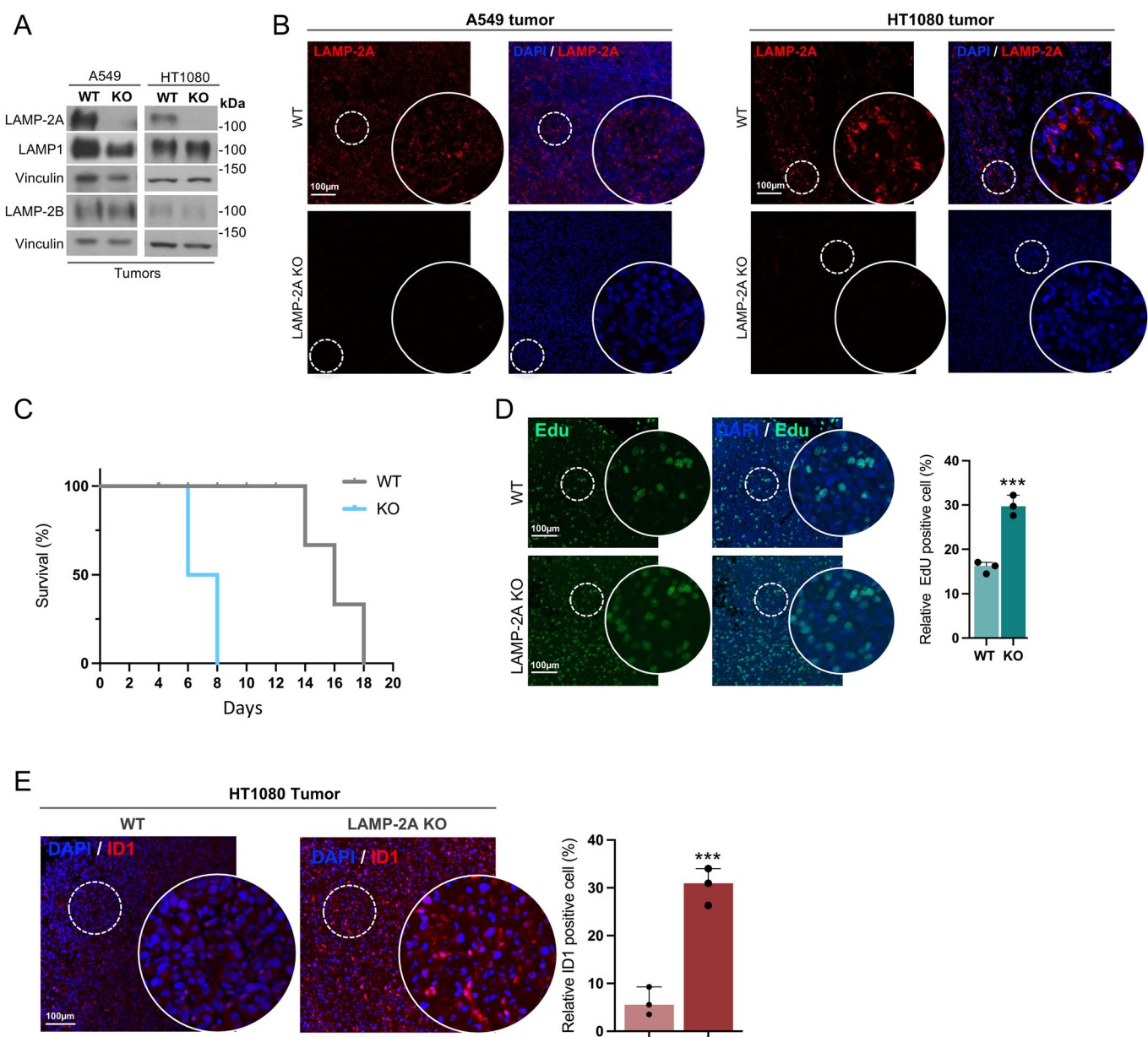

**Figure EV3.** The impact of *LAMP-2A* knockout in cancer cells and tumors.

(A) Immunoblot detection of LAMP-2A, LAMP-2B, and LAMP1 in WT and *LAMP-2A* KO A549 and HT1080 tumor lysates. Vinculin: loading control. (B) Immunofluorescence confocal images of WT and *LAMP-2A* KO A549 and HT1080 tumor sections stained with anti-LAMP-2A (red) and DAPI- (blue) for nuclei highlighted with (circular insets) magnification. Scale bars: 100 μm. (C) Kaplan–Meier survival curves for mice bearing WT or *LAMP-2A* KO HT1080 tumors. (D) Immunofluorescence confocal images (left panel) of EdU positive (green) and DAPI (nuclei, blue) labeled WT and *LAMP-2A* KO HT1080 tumor sections ($n_{exp}$ = 3). Scale bars: 100 μm. Quantification (right panel) of the percentage EdU+ cell ratio per 20× field (bar graph). Bars present mean ± sd. *P* values refer to ***$P < 0.001$ ($P = 0.0008$; Student's *t*-test). (E) Representative immunofluorescence confocal images (left panel) and quantification (right panel) of HT1080 WT and *LAMP-2A* KO tumor section stained with anti-ID1 (red) and DAPI (blue) for nuclei highlighted with (circular insets) magnification. Scale bar: 100 μm. Quantification of the ID1+ cell ratio per 20× field (bar graph) ($n = 3$). Error bars, ±SD. ***$P < 0.001$ ($P = 0.0010$; Student's *t*-test). Source data are available online for this figure.

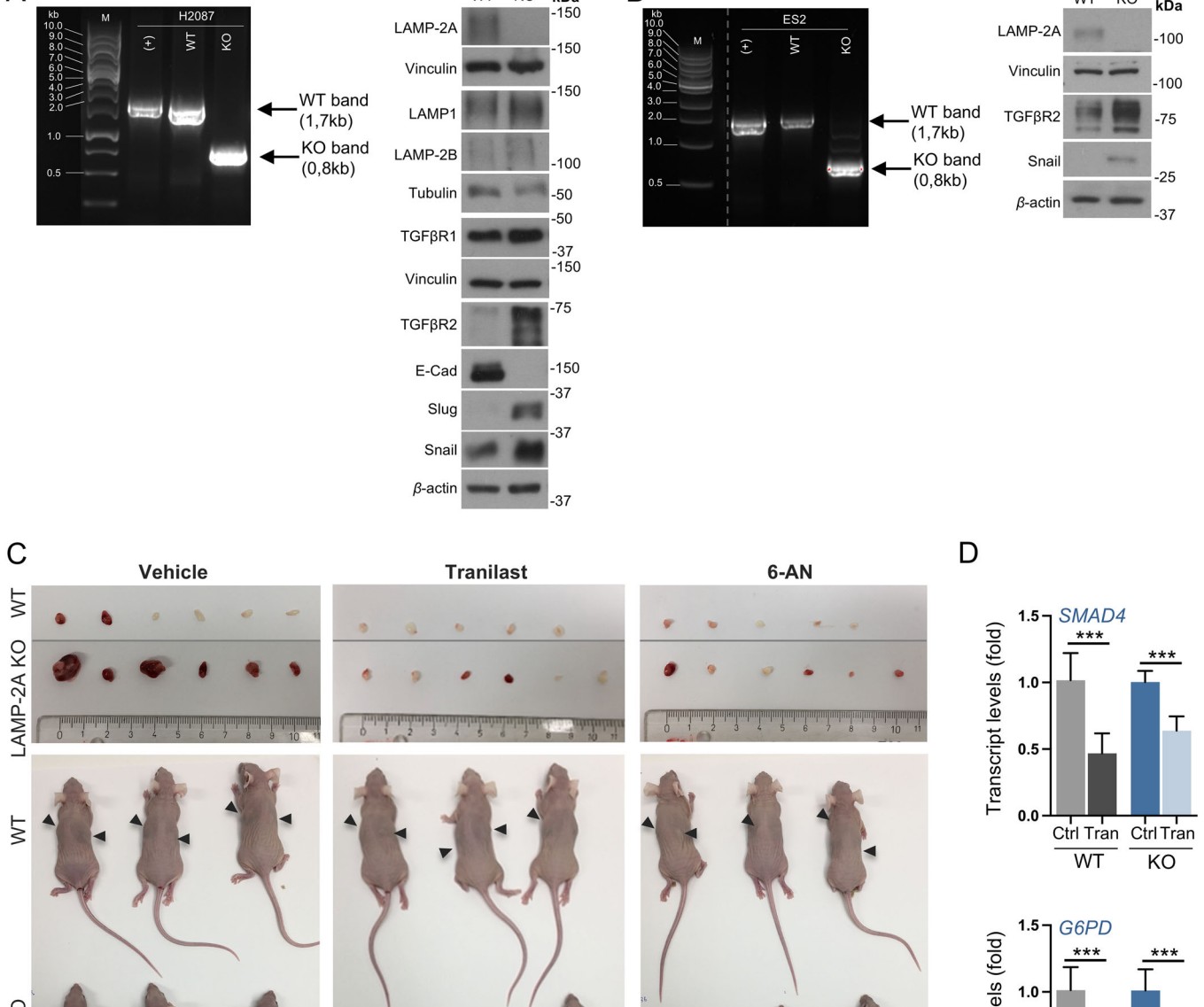

**Figure EV4. LAMP-2A-KO tumor regression in vivo.**

(A) (Left panel) CRISPR/Cas9 genome editing and genotyping by agarose gel electrophoresis/PCR in regular (+), WT, and *LAMP-2A* KO NCI-H2087 cells. Target amplicon in human *LAMP2* gene is detected by the fragment of ~800 bp. M: marker. (Right panel) Immunoblot detection of indicated LAMP proteins, TGFβ receptors, and EMT proteins in WT and *LAMP-2A* KO H2087 cells. Vinculin and tubulin: loading controls. (B) (Left panel) CRISPR/Cas9 genome editing and genotyping by agarose gel electrophoresis/PCR in regular (+), WT and *LAMP-2A* KO ES2 cells. Target amplicon in human *LAMP2* gene is detected by the fragment of ~800 bp. M: marker. (Right panel) Immunoblot detection of LAMP-2A, TGFβR2 and Snail in WT and *LAMP-2A* KO ES2 cells. β-Actin: loading control. (C) Representative images of WT and *LAMP-2A* KO HT1080 xenograft nude mice (two transplants per mice ($n_{mice} = 3$)) and harvested tumors ($n = 6$) after administration with Tranilast or 6-AN for nine days compared to vehicle (corn oil) treated mice/tumors. (D) RT–qPCR detection of *SMAD4* or *G6PD* in WT and *LAMP-2A* KO HT1080 tumors after administration with Tranilast (Tran) or 6-AN, respectively, compared to control (Ctrl) vehicle (corn oil) treatment. ($n_{mice} = 3$; Tran: $P_{WT} = 0.0003$, $P_{KO} < 0.0001$; 6-AN: $P_{WT} < 0.0001$, $P_{KO} < 0.0001$, Student's $t$-test). Bars present mean ± sd. $P$ values refer to ***$P < 0.001$. Source data are available online for this figure.

