## [Peer Review File · EMBO Molecular Medicine]

Chaperone-Mediated Autophagy regulates the Metastatic State of Mesenchymal Tumors

Xun Zhou, Eva Berenger, Yong Shi, Vera Shirokova, Elena Kochetkova, Tina Becirovic, Boxi Zhang, Vitaliy Kaminsky, Yashar Esmailian, Kayoko Hosaka, Cecilia Lindskog, Per Hydbring, Simon Ekman, Yihai Cao, Maria Genander, Marcin Iwanicki, Erik Norberg, and Helin Vakifahmetoglu Norberg

Corresponding authors: Helin Vakifahmetoglu Norberg (helin.norberg@ki.se) , Erik Norberg (Erik.Norberg@ki.se)

Review Timeline:

Submission Date:	4th Apr 24
Editorial Decision:	30th Apr 24
Revision Received:	23rd Dec 24
Editorial Decision:	22nd Jan 25
Revision Received:	27th Jan 25
Editorial Decision:	3rd Feb 25
Authors' Correspondence:	4th Feb 25
Revision Received:	5th Feb 25
Accepted:	17th Feb 25

Editor: Lise Roth

Transaction Report:

30th Apr 2024

Dear Dr. Vakifahmetoglu Norberg,

Thank you for submitting your work to EMBO Molecular Medicine. We have now heard back from the referees who agreed to evaluate your manuscript. As you will see below, the reviewers raise substantial concerns on your work, which unfortunately preclude its publication in EMM in its current form.

As you will see below, the reviewers find that the question addressed by the study is of potential interest, however they remain unconvinced that some of the major conclusions are sufficiently supported by the data.

If you feel you can satisfactorily address all points listed by the referees, you may wish to submit a revised version of your manuscript. Please attach a covering letter giving details of the way in which you have handled each of the points raised by the referees. A revised manuscript will once again be subject to review, and we cannot guarantee at this stage that the eventual outcome will be favorable.

We are expecting your revised manuscript within three to six months, if you anticipate any delay, please contact us.

We require:

- 1) A .docx formatted version of the manuscript text (including legends for main figures, EV figures and tables). Please make sure that the changes are highlighted to be clearly visible.
- 2) Individual production quality figure files as .eps, .tif, .jpg (one file per figure). For guidance, download the 'Figure Guide PDF' (<https://www.embopress.org/page/journal/17574684/authorguide#figureformat>).
- 3) At EMBO Press we ask authors to provide source data for the main figures. Our source data coordinator will contact you to discuss which figure panels we would need source data for and will also provide you with helpful tips on how to upload and organize the files.
- 4) A .docx formatted letter INCLUDING the reviewers' reports and your detailed point-by-point responses to their comments. As part of the EMBO Press transparent editorial process, the point-by-point response is part of the Review Process File (RPF), which will be published alongside your paper.
- 5) A complete author checklist, which you can download from our author guidelines (<https://www.embopress.org/page/journal/17574684/authorguide#submissionofrevisions>). Please insert information in the checklist that is also reflected in the manuscript. The completed author checklist will also be part of the RPF.
- 6) Please note that all corresponding authors are required to supply an ORCID ID for their name upon submission of a revised manuscript.
- 7) It is mandatory to include a 'Data Availability' section after the Materials and Methods. Before submitting your revision, primary datasets produced in this study need to be deposited in an appropriate public database, and the accession numbers and database listed under 'Data Availability'. Please remember to provide a reviewer password if the datasets are not yet public (see <https://www.embopress.org/page/journal/17574684/authorguide#dataavailability>). In case you have no data that requires deposition in a public database, please state so in this section. Note that the Data Availability Section is restricted to new primary data that are part of this study.
- 8) For data quantification: please specify the name of the statistical test used to generate error bars and P values, the number (n) of independent experiments (specify technical or biological replicates) underlying each data point and the test used to calculate p-values in each figure legend. The figure legends should contain a basic description of n, P and the test applied. Graphs must include a description of the bars and the error bars (s.d., s.e.m.). Please provide exact p values.
- 9) Our journal encourages inclusion of *data citations in the reference list* to directly cite datasets that were re-used and obtained from public databases. Data citations in the article text are distinct from normal bibliographical citations and should

directly link to the database records from which the data can be accessed. In the main text, data citations are formatted as follows: "Data ref: Smith et al, 2001" or "Data ref: NCBI Sequence Read Archive PRJNA342805, 2017". In the Reference list, data citations must be labeled with "[DATASET]". A data reference must provide the database name, accession number/identifiers and a resolvable link to the landing page from which the data can be accessed at the end of the reference. Further instructions are available at .

11) For more information: There is space at the end of each article to list relevant web links for further consultation by our readers. Could you identify some relevant ones and provide such information as well? Some examples are patient associations, relevant databases, OMIM/proteins/genes links, author's websites, etc...

12) Author contributions: CRedit has replaced the traditional author contributions section because it offers a systematic machine readable author contributions format that allows for more effective research assessment. Please remove the Authors Contributions from the manuscript and use the free text boxes beneath each contributing author's name in our system to add specific details on the author's contribution. More information is available in our guide to authors.

13) Disclosure statement and competing interests: We updated our journal's competing interests policy in January 2022 and request authors to consider both actual and perceived competing interests. Please review the policy <https://www.embopress.org/competing-interests> and update your competing interests if necessary.

14) Every published paper now includes a 'Synopsis' to further enhance discoverability. Synopses are displayed on the journal webpage and are freely accessible to all readers. They include a short stand first (maximum of 300 characters, including space) as well as 2-5 one-sentences bullet points that summarizes the paper. Please write the bullet points to summarize the key NEW findings. They should be designed to be complementary to the abstract - i.e. not repeat the same text. We encourage inclusion of key acronyms and quantitative information (maximum of 30 words / bullet point). Please use the passive voice. Please attach these in a separate file or send them by email, we will incorporate them accordingly.

15) As part of the EMBO Publications transparent editorial process initiative (see our Editorial at <http://embomolmed.embopress.org/content/2/9/329>), EMBO Molecular Medicine will publish online a Review Process File (RPF) to accompany accepted manuscripts.

In the event of acceptance, this file will be published in conjunction with your paper and will include the anonymous referee reports, your point-by-point response and all pertinent correspondence relating to the manuscript. Let us know whether you agree with the publication of the RPF and as here, if you want to remove or not any figures from it prior to publication.

I look forward to receiving your revised manuscript.

Yours sincerely,

Lise Roth

**** Reviewer's comments ****

Referee #1 (Comments on Novelty/Model System for Author):

The manuscript contains a huge amount of novel data. The results all point to a role of LAMP2A and CMA in maintaining the malignant phenotype of mesenchymal tumors.

Referee #1 (Remarks for Author):

The manuscript describes results supporting the conclusion that human cancer cell-specific knockout of LAMP2A, a limiting protein in chaperone-mediated autophagy (CMA), promotes aggressiveness of mesenchymal xenograft tumors. LAMP2A knockout in the tumor cells resulted in widespread tumor cell dissemination and invasion into the vasculature in zebrafish xenograft model. In human samples, metastatic lesions matched to primary tumors from the same patient showed loss of LAMP2A protein. The results with cell cultures and tumor xenografts showed that increased TGFbeta signaling decreased the levels of LAMP2A, while loss of LAMP2A increased TGFbeta signaling. Inhibition of TGFbeta signaling inhibited growth of LAMP2A-knockout tumors. TGFbetaR2, a receptor for TGFbeta, was shown to be a substrate of CMA.

The manuscript is relatively easy to read, and the majority of the results are convincing. The presentation and readability can be improved, as detailed below.

1. This issue remains unclear: How do the authors explain that there is a strong effect of LAMP2A knockout in promoting the aggressiveness of mesenchymal type tumor cells (which are already mesenchymal), but not epithelial type tumor cells (which need to undergo EMT to become aggressive), if they also propose that LAMP2A and CMA inhibit EMT? E.g. the abstract states 'tumor suppressive functions of CMA involving negative regulation of TGFβ-driven EMT'. Is EMT relevant to mesenchymal type tumor cells as well? Is LAMP2A/CMA loss relevant for the maintenance of the mesenchymal phenotype, rather than EMT? Please clarify this issue to the readers.
2. Are LAMP2A protein levels also affected by Tranilast, CMA activation, and TGFbetaR2 siRNA (Figure 5)?
3. The western blots in Figure 6 should be quantified and the differences should be tested for statistical significance.
4. Gene ontology analysis of the proteomics and RNA seq data (Figs. 1A, 3I) would be helpful for the readers.
5. Line 197: 'Several pro-angiogenic genes were significantly (adjusted-p value < 0.05, fold change > 2) upregulated ... (Fig. 3I)'. It remains unclear to the readers which of the genes in Fig. 3I are pro-angiogenic.
6. Please tell the readers in the results (line 113) and relevant Figure legends (starting from Figure 1) how the CMA levels were experimentally induced. The readers should not need to search for this information in the Methods section.
7. Line 239 onwards: 'we supplemented cells with galactose (to suppress glycolysis) over 5 days, which resulted in growth reduction of both WT and KO by >60% (Fig. 4H), compared to growth in normal media (Fig. 3A)'. It is difficult to compare the growth of the cells between two separate Figures. Please try to make this the 60% decrease visible in Figure 4H.
8. The Figure legends do not give enough information to understand what is shown. Examples: In Figure 1, does 'lysosomal targeted proteins' mean that the proteomics was done on isolated lysosomes? The color code of the heatmap is not explained (Fig. 1A). Does the heat map in Figure 3I refer to xenografts in zebrafish or mice? What kind of tumors were analyzed in Figure 4 (xenografts, human tumor samples)? In Figure 4, please indicate in the beginning of the Figure legend that KO means LAMP2A-knockout. In Figure 6I are the tumor samples from xenografts (mouse or zebrafish)? Figure 7A-B legend: are the results from xenografts? Please tell the readers in Results and Figure legend which cell line was used in Figure 6A-B and Figure 7A-G. In summary: Please give all the necessary details in each Figure legend.
9. Figure 5F legend: there are no upper and lower panels in Figure 5F. Because the annotations in the Figure tell which graph is for which cell type, this does not need to be explained in the legend.
10. The resolution is too low for reading the text in Figure 4A.

11. Figure 6A. Are the differences in the graph statistically significant (Puncta number/cell)? What do the black dots represent if the graph in panel A? They are all outside the bars, so cannot represent individual measurements.

12. Scalebar is mentioned in the legend of Figure 7, but it is not visible in the Figure 7G.

13. Please carefully check the English grammar and correct all mistakes and typos. Just a few examples: lines 120-122 (the sentence is not logical); lines 170-171 (HT1080 fibrosarcoma cells, which is a malignant cancer of mesenchymal origin -> HT1080 fibrosarcoma cells, derived from malignant cancer of mesenchymal origin); line 195 (genes expression changes-> gene expression changes); line 211 (that is essential cancer cell proliferation and tumor growth -> that is essential for cancer cell proliferation and tumor growth); line 213 (significant changes neither in the amino acid levels across the tumor sample groups or... -> significant changes neither in the amino acid levels across the tumor sample groups nor...); etc.

Referee #2 (Comments on Novelty/Model System for Author):

Only primary xenograft growth assays are performed in the mouse models; to draw robust conclusion regarding LAMP2A and CMA in metastasis in vivo, the experiments need to perform metastasis assays in the mouse models. This is major shortcoming of the paper.

Referee #2 (Remarks for Author):

This manuscript evaluates the role of CMA in cancer progression analyzing the effects of LAMP2A knockout in human cancer models exhibiting mesenchymal properties. The genetic loss of CMA results in global changes in the proteome of an ovarian cancer model marked by the increase in EMT-promoting proteins; in parallel, a large fraction of EMT mediators are notable for the presence of a KFERQ motif. Furthermore, in a NSCLC and fibrosarcoma cell lines, the authors demonstrate that loss of LAMP2A results in increased proliferation and xenograft growth. These tumors exhibit increased PPP flux as well as enhanced TGFB signaling. The authors identify TGF β 2 as a CMA target and uncover a reciprocal relationship between TGFB and LAMP2A; their results indicate that stimulating TGF β signaling dampens LAMP2A levels, while the loss of LAMP2A, and by inference CMA, augment TGF β signaling.

Furthermore, treatment of LAMP2KO tumors with the TGFB inhibitor Tranilast or the PPP inhibitor 6-AN reverses the enhancement in tumor growth and proliferation observed in the setting of LAMP2KO deficiency. Although the authors demonstrate effects of LAMP2A on xenograft tumor growth, the precise effects on metastatic potential remain unclear. Additional studies are needed to more rigorously support the conclusions that have been drawn.

1) The authors generate new LAMP2AKO cell lines in HT1080 and A549 but do not provide formal evidence of CMA blockade in these cells. This is an essential missing control. Accumulation of CMA substrates or evidence of CMA loss using the KFERQ reporter should be provided in these newly generated models.

2) The only evidence of a pro-metastatic phenotype is the increased dissemination of LAMP2AKO H1080 cells in a zebrafish transplant model. Further evidence for metastatic behavior the two cell types is needed in mouse models such as experimental metastasis assays. This is a critical missing experiment.

3) Similarly, the reversal assays using Tranilast and 6-AN focus on primary xenograft growth in Figs. 7 and S3. The functional importance of these pathways in metastasis associated with LAMP2A deficiency should be examined using experimental metastasis assays.

4) Tranilast has pleiotropic effects in addition to TGFB inhibition. The authors should provide formal evidence that the treatment is impacting TGFB signaling in these models. Similarly, evidence of PPP inhibition should be provide for the tumor models treated with 6AN.

5) Although EMT plays a role in certain steps of metastasis, such as dissemination, for most tumor types, the rate-limiting steps in metastasis are colonization and outgrowth at foreign tissue sites, which are frequently associated with a mesenchymal-to-epithelial transition (MET). Do the mesenchymal gene programs shown in Figure 2 represent a partial-EMT phenotype? The authors demonstrate low levels of CDH1 in the brain mets of these five patients, arguing against a partial EMT, but a wider gene epithelial program should be analyzed. Furthermore, what is the E-cadherin IHC status in the TMA tissues analyzed in Fig 2E and F?

6) For the TMAs in Fig. 2E and F, it is unclear if the tissue analyzed is primary tumors in patients who developed metastasis or actual tissue biopsied from metastatic sites. If it represents the primary site, then does the inverse relationship observed for LAMP2A and Vimentin also correlate with metastatic behavior or reduced survival, either overall survival or metastasis-free survival? Advanced grade, as measured in Fig 2F, is not a robust correlate for metastatic progression.

Referee #3 (Comments on Novelty/Model System for Author):

As indicated in the answer to the authors, several inconsistencies need to be corrected. While the NSCLC patient expression data, showing a correlation between the EMT and the CMA status and the link between CMA and TGFbeta are interesting, these data are not recapitulated by the most relevant in vivo model the authors used, the NSCLC cell line A549. Instead, authors switch around between cancer cell types and models (mice vs zebrafish), generating inconsistencies between models and patient data. Also, this study somehow contradicts the findings of Kon et al. (DOI: 10.1126/scitranslmed.3003182) who used a CMA knockdown approach in the same lung cancer cell line (A549) and observed decreased tumor burden, correlating with reduced metastasis and increased cell death. However, the only comment made by the authors about this previous paper is "most if not all investigations of CMA in human cancer have relied on KD studies". Authors should explain why there are such opposing effects. This point is not fully addressed. Also although the title and abstract allude to the metastatic potential of modulating CMA in cancer, the authors only show the effects of their genetic or pharmacological manipulations on primary tumors.

Referee #3 (Remarks for Author):

In this manuscript Zhou et al. investigate the effects of genetic loss of LAMP2A the effector of chaperone-mediated autophagy (CMA), on primary tumor growth.

The authors first provide evidence that LAMP2A expression is downregulated in brain metastasis of non-small cell lung cancer patients enriched in EMT markers compared with their matched primary tumors. They move on by showing that loss of LAMP2A increased tumor burden in fibrosarcoma tumors, possibly by a metabolic reprogramming in the PPP pathway, and further linked altered cancer cell's metabolism and reduced tumor growth to alterations in TGFbeta signaling induced by CMA KO.

While the NSCLC patient expression data, showing a correlation between the EMT and the CMA status and the link between CMA and TGFbeta are interesting, these data are not recapitulated by the most relevant in vivo model the authors used, the NSCLC cell line A549. Instead, authors switch around between cancer cell types and models (mice vs zebrafish), generating inconsistencies between models and patient data. Also, this study somehow contradicts the findings of Kon et al. (DOI: 10.1126/scitranslmed.3003182) who used a CMA knockdown approach in the same lung cancer cell line (A549) and observed decreased tumor burden, correlating with reduced metastasis and increased cell death. However, the only comment made by the authors about this previous paper is "most if not all investigations of CMA in human cancer have relied on KD studies". Authors should explain why there are such opposing effects. This point is not fully addressed. Also although the title and abstract allude to metastatic potential, the authors only show the effects of their genetic or pharmacological manipulations on primary tumors.

Specific comments:

- Authors do not provide blots/evidence on the purity of lysosomal isolation for the proteomics data shown in Figure 1a. This is essential to draw correct conclusions about the lysosomal cargo. Also please mention what type of CMA stimulus was applied. Furthermore, authors should characterize the activation status of CMA in their cells, in untreated/basal and following CMA stimulation (starvation?) by using available reporter systems for the selective measurements of CMA. Moreover, authors should further explore the status of macroautophagy in these cells, since both CMA and macroautophagy are lysosomal pathways of protein degradation stimulated by starvation.
 - Did the authors apply a CMA geneset analysis to the patient data (such as the reactome signature for CMA)? Did this follow the LAMP2a findings? This would strengthen the data of figure 2.
 - The statement "These cancer cells represent the first specific LAMP2A-KOs in human cells, allowing a great tool for experimental evaluation of CMA loss-of-function in cancer models" is not correct since A549 LAMP2A KO cells were already generated and used in previous studies from the Cuervo lab (see DOI: 10.1126/scitranslmed.3003182)
 - In Figure 3, it is unclear why 3 mice/conditions are considered for statistical analysis of tumor weight in WT and LAMP2a KO tumors. Furthermore, how do the authors explain their findings in figure 3E-F with those of Kon et al. STM 2011. In the latter case the knockdown of LAMP2A in human lung cancer cell lines, reduced tumor burden and metastasis. Authors should comment on this point. They should also repeat data with the A549 cells by expanding the number of mice to validate the effects of the LAMP2A in this model, as it looks like there is great variability in the tumor growth of the WT of A549 cells.
 - The apparent cancer type-specific effect of LAMP2A KO shown here, does not reconcile with the lung cancer patient data of Fig2. Why?
- Can the author expand the effects of LAMP2A KO in other human lung cancer cell lines? Could the chronic (KO) versus acute (shRNA) inhibition of CMA affect the biological outcome in these cells?
- While the zebrafish data is interesting the authors should tone down the statement of the abstract "...CMA resulted in widespread tumor cell dissemination and invasion into the vasculature, the earliest step of cancer metastasis" unless they provide evidence of reduced invasion in their xenograft models, or remove these data. Indeed the authors do not provide any measurement of metastatic dissemination they only provide data of primary tumor growth and the effects of LAMP2A KO thereof. Hence also the title is misleading.
 - The authors show in Fig1 that, among other EMT regulators, they picked up TFGBR11 motif. However, they do not investigate whether it is regulated by CMA until Figure 5/6. It is strange to jump from this in silico analysis to patient correlation, back to mouse models and mechanistic data. It would make more sense to move the patient correlation to the end and restructure the

mechanistic investigation to bring closer together the identification of TGFBR2 and the CMA-TGFBeta signaling co-regulation and its validation in vitro and in vivo.

Point by Point reply (Ref: EMM-2024-19764)

Corresponding author(s): Erik Norberg and Helin Norberg

General statement:

We thank the editor and reviewers, for their careful assessing of our manuscript, for pointing out its strengths, including novelty, and for providing constructive feedback and suggestion to improve the clarity of the manuscript. We believe that the proposed text changes and additional experimental data presented in response to the comments of all three reviewers have led to a significant clarification and further strengthening of our study

Referee #1

The manuscript contains a huge amount of novel data. The results all point to a role of LAMP2A and CMA in maintaining the malignant phenotype of mesenchymal tumors.

The manuscript describes results supporting the conclusion that human cancer cell-specific knockout of LAMP2A, a limiting protein in chaperone-mediated autophagy (CMA), promotes aggressiveness of mesenchymal xenograft tumors. LAMP2A knockout in the tumor cells resulted in widespread tumor cell dissemination and invasion into the vasculature in zebrafish xenograft model. In human samples, metastatic lesions matched to primary tumors from the same patient showed loss of LAMP2A protein. The results with cell cultures and tumor xenografts showed that increased TGFbeta signaling decreased the levels of LAMP2A, while loss of LAMP2A increased TGFbeta signaling. Inhibition of TGFbeta signaling inhibited growth of LAMP2A-knockout tumors. TGFbetaR2, a receptor for TGFbeta, was shown to be a substrate of CMA.

The manuscript is relatively easy to read, and the majority of the results are convincing. The presentation and readability can be improved, as detailed below.

Reply: We thank the reviewer for the valuable comments and suggestions, especially for acknowledging that our manuscript is bringing up novel important points. We have carefully addressed all concerns raised as below:

Comment 1. *This issue remains unclear: How do the authors explain that there is a strong effect of LAMP2A knockout in promoting the aggressiveness of mesenchymal type tumor cells (which are already mesenchymal), but not epithelial type tumor cells (which need to undergo EMT to become aggressive), if they also propose that LAMP2A and CMA inhibit EMT? E.g. the abstract states 'tumor suppressive functions of CMA involving negative regulation of TGFβ-driven EMT'. Is EMT relevant to mesenchymal type tumor cells as well? Is LAMP2A/CMA loss relevant for the maintenance of the mesenchymal phenotype, rather than EMT? Please clarify this issue to the readers.*

Reply: This is an excellent comment. Our data suggest that the CMA pathway target key components of the EMT cascade, and as also alluded by the reviewer, pharmacological activation of CMA halt EMT by degrading proteins, including mesenchymal marker and regulators of the TGFβ signaling. Therefore, in LAMP2A KO cells, the loss of CMA results in to augmented TGFβRII receptor, and mesenchymal protein accumulation, which contribute to further aggressiveness. At the same time, our data show transcriptional repression of LAMP2A upon EMT induction stimulated by TGFβ ligand. Our understanding of these data is that there is a significant weakening of the CMA function along or after EMT induction, and in cancer cells displaying mesenchymal traits. Thus, the mesenchymal state of cancer cells seems a critical determinant for CMA's function in cancer, regardless of the tumor cell origin, as we see similar effects in carcinomas after EMT induction. While the molecular details of how stimulation of TGFβ signaling suppresses the LAMP2A level needs further investigation, we did notice that several predicted Smad binding sites exist on the upstream promotor region of the human LAMP2 gene. These findings suggest that CMA function maybe actively

suppressed during TGF β -ligand induced EMT to contribute to a mesenchymal phenotype. Cancer cells may have found a way to co-regulate the stimulation of their aggressiveness through EMT signaling and at the same time to suppress CMA that counteracts EMT. In this way cancer cells retain metastatic properties to enhance mobility, invasion, while neutralizing the CMA pathway that prevents it.

Regarding the question whether EMT is relevant to mesenchymal type tumor cells, the precise role of EMT-related processes in tumors originating from mesenchymal tissues, such as soft-tissues sarcomas, incl. HT1080 cells is still paradoxal. This is mainly because these cancers are, by definition, mesenchymal *ab initio*. However accumulating evidence suggests that indeed sarcomas can undergo EMT-related processes. Sarcoma tumors are suggested to acquire features of more mesenchymal differentiated cells that could lead to highly aggressive clinical behavior because the entire tumor will take advantage of the EMT-related biological features of its cells (PMID: 28811330). For example, stimulation of uterine carcinosarcoma cells with TGF β can induce an EMT-like process, which associates with increased cellular migration (PMID: 25918253).

Whether LAMP-2A/CMA loss is more relevant for the maintenance of the mesenchymal phenotype, rather than EMT is also a very interesting question. When comparing WT and LAMP2A-KO A549 cells or tumors (which are epithelial and need to undergo EMT to become aggressive), we did not observe any changes in the TGFBR2 levels or any significant changes in the TGF β signaling, which suggest that the LAMP-2A loss per se is not sufficient to induce EMT. However, in epithelial carcinoma cells the induction of EMT by TGF β ligand is counteracted upon CMA activation, as shown in Figure 6F. Moreover, in ES2 cells that normally already display a mesenchymal phenotype in its WT state, the loss of LAMP-2A leads to expression of further mesenchymal markers, which argues for that LAMP-2A/CMA loss is more relevant for the maintenance of the mesenchymal phenotype, and not the actual induction of EMT.

Taken together based on our data, we suggest that CMA loss leads to accumulation of TGF β R2 and TFs involving in EMT, such as Snail and Slug and, therefore, either contribute to EMT process or to further aggressiveness of already mesenchymal tumors.

We have now added this discussion to the revised manuscript for further clarification of our results, and we have changed the word "potential" to "state" in the title.

Comment 2. *Are LAMP2A protein levels also affected by Tranilast, CMA activation, and TGFbetaR2 siRNA (Figure 5)?*

Reply: Yes, LAMP2A protein level is also upregulated in FUOV1 cells upon Tranilast and CMA activation treatment. This data is now presented in new figure 6E. LAMP2A protein level increase upon TGF β R2 siRNA in FUOV1 cells is further presented in new figure 6E, 6G.

Comment 3. *The western blots in Figure 6 should be quantified and the differences should be tested for statistical significance.*

Reply: All quantifications are now added for all western blots.

Comment 4. *Gene ontology analysis of the proteomics and RNA seq data (Figs. 1A, 3I) would be helpful for the readers.*

Reply: We are grateful for the reviewer's insightful comment. The GO analysis from the RNA seq data is now presented as new figure 4A, the top 10 most enriched pathways with adj p value <0.01 are shown. We have also added a heat map (please see new Figure 4B) to show upregulation of human genes related to angiogenesis in LAMP-2A knockout tumors. For proteomics, we present two GO analysis in the extended data 1E: one that shows top 10 most enriched pathways comparing control vs CMA 16h and the second one – CMA 16h vs CMA 16h+ siLAMP2A conditions.

Comment 5. Line 197: 'Several pro-angiogenic genes were significantly (adjusted-p value < 0.05, fold change > 2) upregulated ... (Fig. 3I)'. It remains unclear to the readers which of the genes in Fig. 3I pro-angiogenic.

Reply: We have now specified which ones are pro-angiogenic in the figure with a star * in new Figure 4B (previously figure 3I), based on the table below.

Genes	pro-angiogenic?	Ref
SH2D2A	Yes	Gordon EJ. The endothelial adaptor molecule TSAd is required for VEGF-induced angiogenic sprouting through junctional c-Src activation. Sci Signal. 2016 Jul 19;9(437):ra72.
ISM1	No	Rao N, Lee YF, Ge R. Novel endogenous angiogenesis inhibitors and their therapeutic potential. Acta Pharmacol Sin. 2015 Oct;36(10):1177-90.
SFRP1	Yes	Dufourcq P, Couffignal T, Ezan J, Barandon L, Moreau C, Daret D, Dupl�a C. FrzA, a secreted frizzled related protein, induced angiogenic response. Circulation. 2002 Dec 10;106(24):3097-103.
CCL24	Yes	Lim SJ. CCL24 Signaling in the Tumor Microenvironment. Adv Exp Med Biol. 2021;1302:91-98.
FGF6	yes	Ornitz DM, Itoh N. The Fibroblast Growth Factor signaling pathway. Wiley Interdiscip Rev Dev Biol. 2015 May-Jun;4(3):215-66.
ADTRP	yes	Lupu C, Patel MM, Lupu F. Insights into the Functional Role of ADTRP (Androgen-Dependent TFPI-Regulating Protein) in Health and Disease. Int J Mol Sci. 2021 Apr 24;22(9):4451.
WNT5A	yes	Chen T, Zhang F, Liu J, Huang Z, Zhang Y, Deng S, Liu Y, Wang J, Sun X. Dual role of WNT5A in promoting endothelial differentiation of glioma stem cells and angiogenesis of glioma derived endothelial cells. Oncogene. 2021 Aug;40(32):5081-5094.
IL1A	yes	Fahey E, Doyle SL. IL-1 Family Cytokine Regulation of Vascular Permeability and Angiogenesis. Front Immunol. 2019 Jun 25;10:1426.
THBS1	No (Metastasis)	Omatsu M. THBS1-producing tumor-infiltrating monocyte-like cells contribute to immunosuppression and metastasis in colorectal cancer. Nat Commun. 2023 Sep 25;14(1):5534.
ESM1	yes	Rocha SF. Esm1 modulates endothelial tip cell behavior and vascular permeability by enhancing VEGF bioavailability. Circ Res. 2014 Aug 29;115(6):581-90.
JCAD	yes	Hara T. Targeted Disruption of JCAD (Junctional Protein Associated With Coronary Artery Disease)/KIAA1462, a Coronary Artery Disease-Associated Gene Product, Inhibits Angiogenic Processes In Vitro and In Vivo. Arterioscler Thromb Vasc Biol. 2017 Sep;37(9):1667-1673.
COL22A1	yes	Liu H, Zeng Z, Sun P. Prognosis and immunoinfiltration analysis of angiogene-related genes in grade 4 diffuse gliomas. Aging (Albany NY). 2023 Sep 21;15(18):9842-9857.
FOXC2	yes	Wang T, Zheng L, Wang Q, Hu YW. Emerging roles and mechanisms of FOXC2 in cancer. Clin Chim Acta. 2018 Apr;479:84-93.
ADGRG1	no	Rao N, Lee YF, Ge R. Novel endogenous angiogenesis inhibitors and their therapeutic potential. Acta Pharmacol Sin. 2015 Oct;36(10):1177-90.

Table for reviewer only:

List of genes indicating their role in angiogenesis.

Comment 6. Please tell the readers in the results (line 113) and relevant Figure legends (starting from Figure 1) how the CMA levels were experimentally induced. The readers should not need to search for this information in Methods section.

Reply. The information on how CMA was activated is now added to all relevant figure legends.

Comment 7. Line 239 onwards: 'we supplemented cells with galactose (to suppress glycolysis) over 5 days, which resulted in growth reduction of both WT and KO by >60% (Fig. 4H), compared to growth in normal media (Fig. 3A).' It is difficult to compare the growth of the cells between two separate Figures. Please try to make this the 60% decrease visible in Figure 4H.

Reply: We have overlaid the growth graphs from Figure 3A into the graph of Figure 4H (now presented as Figure 5H) to make the 60% decrease it more visible.

Comment 8. The Figure legends do not give enough information to understand what is shown. Examples: In Figure 1, does 'lysosomal targeted proteins' mean that the proteomics was done on isolated lysosomes? The color code of the heatmap is not explained (Fig. 1A). Does the heat map in Figure 3I refer to xenografts in zebrafish or mice? What kind of tumors were analyzed in Figure 4 (xenografts, human tumor samples)? In Figure 4, please indicate in the beginning of the Figure legend that KO means LAMP2A-knockout. In Figure 6I are the tumors samples from xenografts (mouse or zebrafish)? Figure 7A-B legend: are the results from xenografts? Please tell the readers in Results and Figure legend which cell line was used in Figure 6A-B and Figure 7A-G. In summary: Please give all the necessary details in each Figure legend.

Reply: We apologies for not making these points clearer.

Figure 1: The proteomics was performed on isolated lysosome fractions from ES2 cells. Color code (min/max) for the heatmap is now presented for Figure 1A.

Figure 3I (now presented as 4B): The heatmap presents upregulated human genes related to angiogenesis from the RNA seq data performed on xenograft tumors from mice. Zebrafish xenograft was only used to present in vivo imaging to monitor invasion and dissemination of cancer cells implanted in the animal system.

Figure 4A (now presented as 5A): Wildtype and LAMP-2A KO xenograft tumors from mice are analyzed. We do not have human tumor samples with LAMP-2A KO.

We have now indicated that KO is LAMP2A-knockout in the figure legend.

Figure 7A-B (now presented as Figure 8A-B) legend: Results are from cell culture experiments.

Figure 6A-B and Figure 7A-G and for all other figure: All necessary information is now indicated both in the results and figure legends.

Comment 9. *Figure 5F legend: there are no upper and lower panels in Figure 5F. Because the annotations in the Figure tell which graph is for which cell type, this does not need to be explained in the legend.*

Reply: This has been clarified.

Comment 10. *The resolution is too low for reading the text in Figure 4A.*

Reply: Figure 4A (now presented as Figure 5A) has been corrected.

Comment 11. *Figure 6A. Are the differences in the graph statistically significant (Puncta number/cell)? What do the black dots represent if the graph in panel A? They are all outside the bars, so cannot represent individual measurements.*

Reply: In Figure 6A (now presented as Figure 7A), the number of puncta per cell was quantified (more than 50 cells), and the differences was analyzed by unpaired Student's t test. The difference between Ctrl and Tranilast groups and Ctrl and CMA groups are statistically significant $p < 0.0001$, which are now indicated in the figure.

The previous graph was created via GraphPad prism using Box and Whiskers plots. The data were present in the way of 10-90 percentiles. And in this way, the whiskers are drawn down to the 10th percentile and up to the 90th, points below and above the whiskers are drawn as individual points. Therefore, black dots above the bars represented the individual measurements, but only the points which are outside the range. However, to avoid any misunderstanding, we have now replotted this graph showing all individual values with indicated mean.

Comment 12. *Scalebar is mentioned in the legend of Figure 7, but it is not visible in the Figure 7G.*

Reply: Figure 7G (presented now as Figure 8G) has been corrected.

Comment 13. *Please carefully check the English grammar and correct all mistakes and typos. Just a few examples: lines 120-122 (the sentence is not logical); lines 170-171 (HT1080 fibrosarcoma cells, which is a malignant cancer of mesenchymal origin -> HT1080 fibrosarcoma cells, derived from malignant cancer of mesenchymal origin); line 195 (genes expression changes-> gene expression changes); line 211 (that is essential cancer cell proliferation and tumor growth -> that is essential for cancer cell proliferation and tumor growth); line 213 (significant changes neither in the amino acid levels across the tumor sample groups or... -> significant changes neither in the amino acid levels across the tumor sample groups nor...); etc.*

Reply: We thank the reviewer for identifying these errors. The suggested corrections have been implemented.

Referee #2

Only primary xenograft growth assays are performed in the mouse models; to draw robust conclusion regarding LAMP2A and CMA in metastasis *in vivo*, the experiments need to perform metastasis assays in the mouse models. This is major shortcoming of the paper.

This manuscript evaluates the role of CMA in cancer progression analyzing the effects of LAMP2A knockout in human cancer models exhibiting mesenchymal properties. The genetic loss of CMA results in global changes in the proteome of an ovarian cancer model marked by the increase in EMT-promoting proteins; in parallel, a large fraction of EMT mediators are notable for the presence of a KFERQ motif. Furthermore, in a NSCLC and fibrosarcoma cell lines, the authors demonstrate that loss of LAMP2A results in increased proliferation and xenograft growth. These tumors exhibit increased PPP flux as well as enhanced TGFB signaling. The authors identify TGF β R2 as a CMA target and uncover a reciprocal relationship between TGFB and LAMP2A; their results indicate that stimulating TGF β signaling dampens LAMP2A levels, while the loss of LAMP2A, and by inference CMA, augment TGF β signaling.

Furthermore, treatment of LAMP2KO tumors with the TGFB inhibitor Tranilast or the PPP inhibitor 6-AN reverses the enhancement in tumor growth and proliferation observed in the setting of LAMP2KO deficiency. Although the authors demonstrate effects of LAMP2A on xenograft tumor growth, the precise effects on metastatic potential remain unclear. Additional studies are needed to more rigorously support the conclusions that have been drawn.

Reply: We thank this reviewer for these valuable comments. We have taken all the suggestions, performed new experiments including metastasis assay in mouse models and carefully addressed all points raised by the reviewer as below:

Comment 1) The authors generate new LAMP2AKO cell lines in HT1080 and A549 but do not provide formal evidence of CMA blockade in these cells. This is an essential missing control. Accumulation of CMA substrates or evidence of CMA loss using the KFERQ reporter should be provided in these newly generated models.

Reply: We appreciate the reviewer's comment. For the newly generated A459 and HT1080 LAMP2A KO cell lines, we have added evidence of CMA loss using the KFERQ reporter, which are now presented in revised extended data 2F.

Comment 2) The only evidence of a pro-metastatic phenotype is the increased dissemination of LAMP2AKO H1080 cells in a zebrafish transplant model. Further evidence for metastatic behavior the two cell types is needed in mouse models such as experimental metastasis assays. This is a critical missing experiment.

Reply: By the recommendation of the reviewer, we wrote and applied for a completely new ethical permit to perform experimental metastasis assay using mouse models. In the experimental metastasis assay, tumor growth in the primary site is bypassed, and cells were delivered directly into the circulation for direct cancer cell seeding into distant organs through injection into the tail vein. To perform this experiment, we collaborated with Professor Yihai Cao, an internationally recognized in cancer metastasis and tumor angiogenesis research. These new data are presented as Figure 4D-E and provide strong experimental evidence that further strengthen our study and allow us to draw robust conclusion regarding LAMP-2A deficiency, thus CMA loss in metastasis *in vivo*.

Comment 3) Similarly, the reversal assays using Tranilast and 6-AN focus on primary xenograft growth in Figs. 7 and S3. The functional importance of these pathways in metastasis associated with LAMP2A deficiency should be examined using experimental metastasis assays.

Reply: We thank the reviewer for this comment. In the experimental metastasis assay, mice injected with WT HT1080 cells did not display a heavy metastatic burden compared to LAMP-2A KO mice. Since our data also demonstrate that administration of either tranilast or 6AN already display a significant impact on the growth of primary LAMP-2A KO HT1080 tumors *in*

vivo, prevents us from drawing any conclusions about its impact on LAMP-2A KO HT1080 metastatic tumors compared to WT cells. Based on the reviewer's request we performed the most critical experiment to prove that LAMP-2A KO has a significant impact on metastasis in the mouse model. Furthermore, we generated additional LAMP-2A KO in additional cancer cell lines to provide further evidence that LAMP-2A KO affects several genes associated with the metastatic potential of tumors besides its effect on TGF β signaling (please see new extended Figure 3). We hope that the reviewer agrees that this new data further support the main message from our study.

Comment 4) *Tranilast has pleiotropic effects in addition to TGFB inhibition. The authors should provide formal evidence that the treatment is impacting TGFB signaling in these models. Similarly, evidence of PPP inhibition should be provide for the tumor models treated with 6AN.*

Reply: We provide evidence on the inhibition of TGF β signaling or PPP in WT and KO HT1080 tumors by presenting qPCR data, which show a significant effect of Tranilast or 6AN treatment on *SMAD4* and *G6PD* genes, respectively. Please see these new data in the figure extended data 4D.

Comment 5) *Although EMT plays a role in certain steps of metastasis, such as dissemination, for most tumor types, the rate-limiting steps in metastasis are colonization and outgrowth at foreign tissue sites, which are frequently associated with a mesenchymal-to-epithelial transition (MET). Do the mesenchymal gene programs shown in Figure 2 represent a partial-EMT phenotype? The authors demonstrate low levels of CDH1 in the brain mets of these five patients, arguing against a partial EMT, but an wider gene epithelial program should be analyzed. Furthermore, what is the E-cadherin IHC status in the TMA tissues analyzed in Fig 2E and F?*

Reply: We appreciate the reviewer's insightful comment. The analysis on patient samples comprises a specific nanostring gene set (The nanostring panel is a pancancer immunology panel. <https://nanostring.com/products/ncounter-assays-panels/oncology/pancancer-immune-profiling/>). When the analysis was conducted, it was not possible to customize nanostring panels. Further, due to the nature of the clinical samples (retrospective FFPE samples), RNA-seq could not be conducted with sufficient quality output. Nanostring is more well-suited for retrospective samples since it measures endogenous transcripts directly by probe-binding and does not rely on amplification techniques. We chose this method because it still contained 16 EMT markers amenable for detection without amplification. Unfortunately, due to lack of additional clinical material of these matched samples, the analysis could not be repeated with a customized epithelial panel.

Further, in full agreement with the reviewer, a partial EMT is commonly classified as a state that is neither fully epithelial nor mesenchymal. However, multiple partial EMT states exist. Consequently, due to the dynamic nature of this process, revealing the actual EMT status and choosing suitable markers to detect partial or full EMT for tracing EMT stages in of cancer cells or tumors remains technically challenging. In fact, in 2020, Li et al. reported that continuous N-cadherin expression mapped the majority of pulmonary PyMT metastasis (PMID: 32668208). This was in sharp contrast to the results of Lüönd et al. that show complete absence of metastases labeled by N-cadherin (PMID: 34847378). Further, the data of Aban et al., (PMID: 33479502), demonstrated that downregulation of E-cadherin in pluripotent stem cells triggers partial EMT. Their data showed that decrease in E-cadherin causes an incomplete EMT where cells retain their undifferentiated state while expressing several characteristics of a mesenchymal-like phenotype.

Therefore, the interpretation of whether a low E-cad levels might argue for or against a partial or full EMT is difficult in our study. Here, E-cadherin (*CDH1*) represents one gene of the epithelial state and we believe that it does not allow us to draw any conclusions about the a partial EMT or not in the brain metastasis of patients.

Nonetheless, based on the request from the reviewer, we performed new IHC staining for E-cadherin in the TMA tissues analyzed in Fig 2E-F at the Human Protein Atlas. As seen by the staining below, E-Cad displayed very low levels across (except the first top left core). These data are in line with the impact LAMP-2A KO exerted in the newly generated H2087 cancer cell line, show in extended **Figure 4**. NCI-H2087 [H2087], which is a hyperdiploid cell line from lung, metastatic site, adenocarcinoma, held at ATCC. The LAMP2A-KO in H2087 cells led to significant increase in TGFBR2, Vimentin, Snail and Slug levels, concomitant with a marked decrease in E-Cad expression.

Figure for referee with unpublished data and its description has been removed upon request by the authors.

Comment 6) *For the TMAs in Fig. 2E and F, it is unclear if the tissue analyzed is primary tumors in patients who developed metastasis or actual tissue biopsied from metastatic sites. If it represents the primary site, then does the inverse relationship observed for LAMP2A and Vimentin also correlate with metastatic behavior or reduced survival, either overall survival or metastasis-free survival? Advanced grade, as measured in Fig 2F, is not a robust correlate for metastatic progression.*

Reply: TMAs used in the study are actual tissue resected from metastatic sites (purchased from Biocat US, as: a) MT2081 Multiple organ metastatic carcinoma tissue microarray, 19 organs/104 cases. b) MT801 Multiple organ metastatic carcinoma tissue microarray, 80 cases. Both TMA types are metastasis with pathological diagnosis, and they don't represent the primary sites. We have now included further clarification to improve our description in the material and methods part on TMAs.

We agree with the reviewer that tumor grade is not a robust measure for metastatic progression. Here the advanced grade is a correlate for cell differentiation, representing: Well differentiated (Low grade), Moderately differentiated (Intermediate grade) and Poorly differentiated (High grade). The grades for metastatic cancer samples have the same grading standards as primary cancer. Sometimes the metastatic tumors are higher grade than primary cancer and more malignant. Since we analyzed multi-organ metastatic tissues, we felt that it may be relevant to show the grading since it is important for some tumor types in planning the treatment and to predict the outcome including breast and prostate cancer as well as for sarcomas.

Referee #3

As indicated in the answer to the authors, several inconsistencies need to be corrected. While the NSCLC patient expression data, showing a correlation between the EMT and the CMA status and the link between CMA and TGFbeta are interesting, these data are not recapitulated by the most relevant in vivo model the authors used, the NSCLC cell line A549. Instead, authors switch around between cancer cell types and models (mice vs zebrafish), generating inconsistencies between models and patient data. Also, this study somehow contradicts the findings of Kon et al. (DOI: 10.1126/scitranslmed.3003182) who used a CMA knockdown approach in the same lung cancer cell line (A549) and observed decreased tumor burden, correlating with reduced metastasis and increased cell death. However, the only comment made by the authors about this previous paper is "most if not all investigations of CMA in human cancer have relied on KD studies". Authors should explain why there are such opposing effects. This point is not fully addressed. Also although the title and abstract allude to the metastatic potential of modulating CMA in cancer, the authors only show the effects of their genetic or pharmacological manipulations on primary tumors.

In this manuscript Zhou et al. investigate the effects of genetic loss of LAMP2A the effector of chaperone-mediated autophagy (CMA), on primary tumor growth. The authors first provide evidence that LAMP2A expression is downregulated in brain metastasis of non-small cell lung cancer patients enriched in EMT markers compared with their matched primary tumors. They move on by showing that loss of LAMP2A increased tumor burden in fibrosarcoma tumors, possibly by a metabolic reprogramming in the PPP pathway, and further linked altered cancer cell's metabolism and reduced tumor growth to alterations in TGFbeta signaling induced by CMA KO. While the NSCLC patient expression data, showing a correlation between the EMT and the CMA status and the link between CMA and TGFbeta are interesting, these data are not recapitulated by the most relevant in vivo model the authors used, the NSCLC cell line A549. Instead, authors switch around between cancer cell types and models (mice vs zebrafish), generating inconsistencies between models and patient data. Also, this study somehow contradicts the findings of Kon et al. (DOI: 10.1126/scitranslmed.3003182) who used a CMA knockdown approach in the same lung cancer cell line (A549) and observed decreased tumor burden, correlating with reduced metastasis and increased cell death. However, the only comment made by the authors about this previous paper is "most if not all investigations of CMA in human cancer have relied on KD studies". Authors should explain why there are such opposing effects. This point is not fully addressed. Also although the title and abstract allude to metastatic potential, the authors only show the effects of their genetic or pharmacological manipulations on primary tumors.

Reply: We are grateful to the reviewers' valuable suggestions, comments and acknowledging that our data are interesting. Please find below our detailed response to each point raised by the reviewer.

Comment 1: *Authors do not provide blots/evidence on the purity of lysosomal isolation for the proteomics data shown in Figure 1a. This is essential to draw correct conclusions about the lysosomal cargo. Also please mention what type of CMA stimulus was applied. Furthermore, authors should characterize the activation status of CMA in their cells, in untreated/basal and following CMA stimulation (starvation?) by using available reporter systems for the selective measurements of CMA. Moreover, authors should further explore the status of macroautophagy in these cells, since both CMA and macroautophagy are lysosomal pathways of protein degradation stimulated by starvation.*

Reply: Based on the request from the reviewer, we present a new figure "extended data 1" to show the lysosomal fractions from ES2 cells (extended data 1A). The analyzed fraction (marked in red) shows absence of LDHA, TOMM40 and LC3, demonstrating the purity of the lysosomal fractions and nonappearance of mitochondria and autophagosomes. Moreover, lysosomal fraction from control (DMSO) and CMA activated condition (8 h and 16 h AC220+Spautin-1 treatments), as well as 16 h CMA treatment with genetic silencing of LAMP2A conditions are shown (extended data 1B). LAMP2A levels are shown in these

fractions that were sent to Mass Spectrometry. These data clearly show that fractions used in the study were predominantly enriched with lysosomes. Further, we have in our previous publications from 2019 and 2021 (please see ref. 1, 2, below), demonstrated how multiplexed mass spectrometry analysis with quantitative proteomic comparison led to the identification of novel CMA substrates using isolated lysosome fraction sample sets from cancer cells.

¹ Hao Y, Kacal M, Ouchida AT, Zhang B, Norberg E, Vakifahmetoglu-Norberg H. Targetome analysis of chaperone-mediated autophagy in cancer cells. **Autophagy**. 2019;1-14.

²Kacal M, Zhang B, Hao Y, Norberg E, Vakifahmetoglu-Norberg H. Quantitative proteomic analysis of temporal lysosomal proteome and the impact of the KFERQ-like motif and LAMP2A in lysosomal targeting. **Autophagy**. 2021; 17(11):3865-3874

For CMA activation, all cells used in this study were treated with the small molecule receptor tyrosine kinase inhibitor AC220 (quizartinib), known to selectively reduce cellular glucose uptake without affecting glutamine uptake, simultaneously with the specific and potent autophagy inhibitor, Spautin-1 (ref 3), to eliminate the contribution from the MA and to activate the lysosomal proteolytic pathway of CMA (please see the ref. 3, 4 and 5 below). By using Spautin-1 we eliminate the contribution of macroautophagy (MA). Further, using genetic silencing of LAMP2A, we focused on LAMP2A-dependent lysosomal protein enrichment.

³ Liu J, Xia H, Kim M, Xu L, Li Y, Zhang L, et al. Beclin1 controls the levels of p53 by regulating the deubiquitination activity of USP10 and USP13. **Cell**. 2011;147(1):223-34.

⁴ Vakifahmetoglu-Norberg H, Kim M, Xia HG, Iwanicki MP, Ofengeim D, Coloff JL, Pan L, Ince TA, Kroemer G, Brugge JS *et al* (2013) Chaperone-mediated autophagy degrades mutant p53. *Genes Dev* 27: 1718-1730

⁵ Xia HG, Najafov A, Geng J, Galan-Acosta L, Han X, Guo Y, et al. Degradation of HK2 by chaperone-mediated autophagy promotes metabolic catastrophe and cell death. **J Cell Biol**. 2015;210(5):705-16.

This combination treatment to activate CMA was first described in 2015, “*Degradation of HK2 by chaperone-mediated autophagy promotes metabolic catastrophe and cell death. J Cell Biol*” (ref 5).

Figure for referee with unpublished data and its description has been removed upon request by the authors.

In 2019, a similar CMA reporter was created by Shu Leong Ho Lab (PubMed 30983487) and recently deposited at Addgene (pSIN-PAmCherry-KFERQ-NE (Plasmid #102365). We present data using this reporter upon CMA activation by the combination treatment of AC220

and Spautin-1 in both WT and KO LAMP-2A A549 and HT1080 cells in this study (extended data 2F). These data clearly show that CMA is activated in WT cells upon AC220 and Spautin-1 treatment and provide evidence of CMA loss in LAMP-2A KO cells by using a PAmCherry-KFERQ-CMA reporter. In addition, in Figure 7A, we present activation of CMA using the PAmCherry reporter in in OVPA8 cells.

Comment 2: *Did the authors apply a CMA geneset analysis to the patient data (such as the reactome signature for CMA)? Did this follow the LAMP2a findings? This would strengthen the data of figure 2.*

Reply: We appreciate the reviewers insightful comment. The analysis on patient samples comprises a specific nano string gene set (The nanostring panel is a pancancer immunology panel of close to 800 genes).

<https://nanostring.com/products/ncounter-assays-panels/oncology/pancancer-immune-profiling/>). When the analysis was conducted there were no possibility to customize nanostring panels for LAMP2A, since it's an isoform of LAMP2 gene. We choose nanostring analysis, over RNA seq, as it is more suitable for retrospective FFPE samples, as it measures endogenous transcripts directly by probe-binding.

Regarding a CMA gene set or reactome signature analysis mentioned by the reviewer, we agree that a CMA gene set analysis would be great to include in all data set in this study. We assume that the reviewer refers to the Human GeneSet:

https://www.gsea-msigdb.org/gsea/msigdb/cards/REACTOME_CHAPERONE_MEDIATED_AUTOPHAGY

Unfortunately, this gene set has several flaws (listed below) and is not so relevant to CMA and human cancer.

- The gene set includes LAMP2 and not LAMP-2A and is therefore not exclusive for CMA.
- Most genes included in this set are associated to CMA as substrates but not regulators of the CMA pathway.
- Some of these genes, such as PLIN2 is not a CMA substrate in human cells, but in mice and other genes, such as GFAP regulate the CMA post transcriptionally by its phosphorylation and not based on gene expression.
- The gene set mainly originates from publications describing CMA in the context of rodent fibroblasts, hepatocytes and liver tissues, or cultured fibroblasts in response to oxidative, or ER stress and in aging conditions of models of neurodegenerative diseases.

Thus, although the reviewer's suggestion is very relevant, based on currently available knowledge of the CMA pathway, it is not meaningful to profile these genes for expression in human cancer samples as it does not reflect a proper gene set for CMA's function.

Source Id	NCBI (Entrez) Gene Id	Gene Symbol	Gene Description
ENSG0000001626	1080	CFTR	CF transmembrane conductance regu...
ENSG0000005893	3920	LAMP2	lysosomal associated membrane pro...
ENSG00000026025	7431	VIM	vimentin [Source:HGNC Symbol;Acc:...
ENSG00000032742	8100	IFT88	intraflagellar transport 88 [Sour...
ENSG00000080824	3320	HSP90AA1	heat shock protein 90 alpha famil...
ENSG00000094631	10013	HDAC6	histone deacetylase 6 [Source:HGN...
ENSG00000096384	3326	HSP90AB1	heat shock protein 90 alpha famil...
ENSG00000105355	10226	PLIN3	perilipin 3 [Source:HGNC Symbol;A...
ENSG00000109971	3312	HSPA8	heat shock protein family A (Hsp7...
ENSG00000116288	11315	PARK7	Parkinsonism associated deglycase...
ENSG00000129538	6035	RNASE1	ribonuclease A family member 1, p...
ENSG00000131095	2670	GFAP	glial fibrillary acidic protein [...
ENSG00000143947	6233	RPS27A	ribosomal protein S27a [Source:HG...
ENSG00000147872	123	PLIN2	perilipin 2 [Source:HGNC Symbol;A...
ENSG00000150991	7316	UBC	ubiquitin C [Source:HGNC Symbol;A...
ENSG00000156508	1915	EEF1A1	eukaryotic translation elongation...
ENSG00000160299	5116	PCNT	pericentrin [Source:HGNC Symbol;A...
ENSG00000169379	200894	ARL13B	ADP ribosylation factor like GTPa...
ENSG00000170315	7314	UBB	ubiquitin B [Source:HGNC Symbol;A...
ENSG00000177143	1068	CETN1	centrin 1 [Source:HGNC Symbol;Acc...
ENSG00000221983	7311	UBA52	ubiquitin A-52 residue ribosomal ...
ENSG00000244734	3043	HBB	hemoglobin subunit beta [Source:H...

Table for reviewer only:

List of the 22 genes indicated in the CMA gene set:

https://www.gsea-msigdb.org/gsea/msigdb/cards/REACTOME_CHAPERONE_MEDIATED_AUTOPHAGY

Comment 3: *The statement "These cancer cells represent the first specific LAMP2A-KOs in human cells, allowing a great tool for experimental evaluation of CMA loss-of-function in cancer models" is not correct since A549 LAMP2A KO cells were already generated and used in previous studies from the Cuervo lab (seeDOI: 10.1126/scitranslmed.3003182)*

Reply: Here we want to clarify that the reviewers comment and refers to publication by Kon et al., (Sci Transl Med. 2011, seeDOI: 10.1126/scitranslmed.3003182). In this paper the authors used a knockdown approach and not a gene deletion, which is also stated by the reviewer in multiple of the comments as a CMA knockdown.

....."Also, this study somehow contradicts the findings of Kon et al. (DOI: 10.1126/scitranslmed.3003182) who used a **CMA knockdown approach** in the same lung cancer cell line (A549) and observed decreased tumor burden".....

Indeed, the cited publication by the reviewer does not contain any LAMP-2A KO cells, but instead knockdown of LAMP2A in the A549 cells, which was done using replication-deficient lentivirus-targeted short hairpin RNA (shRNA) against LAMP-2A.

Our lab is the first to develop a CRISPR-Cas9 mediated system to create isoform specific LAMP-2A KO human cell lines (PMID: 34972984). In 2022, we published the first method paper (Methods Mol Biol. 2022:2445:39-50. doi: 10.1007/978-1-0716-2071-7_3) on how to generate human isoform specific LAMP-2A KO cells without affecting the other LAM2 gene

isoforms (LAMP-2B and LAMP-2C) in human cancer cell lines, incl. A549 cells. We further validated LAMP-2A antibody specificity using the WT and KO A549 cancer cells, published in *Autophagy*. 2023 Sep;19(9):2575-2577. doi: 10.1080/15548627.2023.2213515.

Following our method paper, it was used to generate LAMP-2A KO by another lab, however, on human RPE cell line ARPE-19, which is a retinal pigment epithelia cell line (<https://www.ncbi.nlm.nih.gov/pmc/articles/PMC8956266/>), and not a cancer cell line.

Comment 4: *In Figure 3, it is unclear why 3 mice/conditions are considered for statistical analysis of tumor weight in WT and LAMP2a KO tumors. Furthermore, how do the authors explain their findings in figure 3E-F with those of Kon et al. STM 2011. In the latter case the knockdown of LAMP2A in human lung cancer cell lines, reduced tumor burden and metastasis. Authors should comment on this point. They should also repeat data with the A549 cells by expanding the number of mice to validate the effects of the LAMP2A in this model, as it looks like there is great variability in the tumor growth of the WT of A549 cells.*

Reply: The experiments were done with 6 mice/condition, where two WT and two LAMP-2A KO cells were injected per mouse. However, in the graphs, we presented the tumor weights from those exact 3 mice/condition shown in Figure 3. We have now added all data from 6 mice with all tumors as below to the graphs in the revised figure 3F.

Figure for referee with unpublished data and its description has been removed upon request by the authors.

The reviewer further states that our findings differ from those of Kon et al., in which the knockdown (KD) of LAMP-2A was investigated. There, the authors conclude that CMA KD led to growth disadvantage for A549 tumors measured in different mice. Our findings show no statistical difference between WT and KO LAMP2A A549 xenografts within the same mice. We have summarized the animal experiments from our study compared to Kon et al., below to make the differences clearer.

	Our findings	Kon et al.,
Mice used	BALB/c nude female	Nu/Nu athymic
Age of mice	7-8 week-old	5-6 week-old
Purchased from	Janvier	NCI-Frederick
Gender	female	male
LAMP2A	Knockout KO	Knockdown KD
Method	CRISPR-Cas9	shRNA
Cell injected	1 × 10 ⁶ cells	1 × 10 ⁶ cells
Flank	Right and Left	Right flank
Injection/mice	2 WT and 2 KO	1 WT or 1 KD

Moreover, we present a period of 14 days tumor growth for A549 cells. Looking at the graph Figure 5A in Kon et al., we see that at similar time points – up to 20 days showed no significant difference between WT and KD tumors.

Figure for referee with unpublished data and its description has been removed upon request by the authors.

Further, in Kon et al., the authors concluded that CMA knockdown led to growth disadvantage for A549 tumors, possibly due to reduced glycolytic capacity. They suggested that changes in mitochondrial oxidative phosphorylation in LAMP-2A KD contribute minimally to the overall energy balance of these cells. Our findings in LAMP2A-KO human fibrosarcoma cells indicate a stronger reliance on mitochondrial metabolism over glycolysis. This conclusion is further supported by our measurements of acetyl-CoA levels by mass spec. We believe the observed differences in energy dependence between may explain the variation in CMA activity after LAMP-2A KO and could be attributed to the distinct metabolic backgrounds of these cell types.

Of note, in a CMA study published in *Science* (PMID: 32703873), data both on LAMP2A KD and KO by CRISPR gene editing in mouse embryonic stem (ES) cells were investigated. In this study, it was demonstrated that the expression of previously known CMA substrates in non-stem cells, including several glycolytic enzymes, remained unchanged in embryonic stem (ES) cells upon LAMP2A KO compared to shown by knockdown. It is likely that genetic compensation may occur, and transcriptional adaptation can trigger changes in the genome, that may not be observed in (more acute) knockdown studies. However, to our understanding it would not be correct to compare shRNA-mediated KD data to CRISPR-mediated KO data, also considering that the animal experiments were performed with above mentioned differences.

Comment 5: *The apparent cancer type-specific effect of LAMP2A KO shown here, does not reconcile with the lung cancer patient data of Fig2. Why?*

Can the author expand the effects of LAMP2A KO in other human lung cancer cell lines? Could the chronic (KO) versus acute (shRNA) inhibition of CMA affect the biological outcome in these cells?

Reply: We thank the reviewer for this very relevant comment. Our data show that, unlike the epithelial cancer cell model (A549), LAMP2A-KO displays a more significant effect in the aggressiveness of the mesenchymal fibrosarcoma model (HT1080). We do not state that there is a cancer-type-specific effect of LAMP2A-KO due to differences of lung vs fibrosarcoma cell-type, but instead the mesenchymal state of cancer cells seems to be a critical determinant for CMA's function in cancer cells, regardless of the tumor cell origin, as we see similar effects in carcinomas after EMT induction. Therefore, there is no contradiction in the data presented with the lung cancer patient data of Figure 2.

Based on reviewers' request and we created an additional human lung cancer cell line with LAMP2A KO, NCI-H2087 [H2087], which is a hyperdiploid cell line from lung, metastatic

site, adenocarcinoma, held at ATCC. In contrast to A549, the LAMP2A-KO in H2087 cells led to significant increase in TGFBR2, Vimentin, Snail and Slug levels, concomitant with a marked decrease in E-Cad expression, which is in line with our data on fibrosarcoma model (HT1080). Therefore, our findings argues that the 'EMT status' is more of a determinant than the tissue of origin when it comes to the effect of CMA deficiency (LAMP-2A KO). These new data are now presented in the extended data 4A.

Comment 6: While the zebrafish data is interesting the authors should tone down the statement of the abstract "...CMA resulted in widespread tumor cell dissemination and invasion into the vasculature, the earliest step of cancer metastasis" unless they provide evidence of reduced invasion in their xenograft models, or remove these data. Indeed the authors do not provide any measurement of metastatic dissemination they only provide data of primary tumor growth and the effects of LAMP2A KO thereof. Hence also the title is misleading.

Reply: We thank the reviewer for this very insightful comment. We followed the recommendation, and we wrote and applied for a new ethical permit to perform experimental metastasis assay using mouse model. In the experimental metastasis assay, tumor growth in the primary site is bypassed, and cells were delivered directly into the circulation for direct seeding to distant organs through injection into the tail vein. These new data are presented as Fig 4, where the reviewer can appreciate a dramatic impact of CMA deficiency on metastatic tumor burden in lungs.

Comment 7: *The authors show in Fig1 that, among other EMT regulators, they picked up TGFBR2 motif. However, they do not investigate whether it is regulated by CMA until Figure 5/6. It is strange to jump from this in silico analysis to patient correlation, back to mouse models and mechanistic data. It would make more sense to move the patient correlation to the end and restructure the mechanistic investigation to bring closer together the identification of TGFBR2 and the CMA-TGFβ signaling co-regulation and its validation in vitro and in vivo.*

Reply: We have carefully re-checked all the EMT-related proteins with CMA motifs as presented in table 1 and those indicated in figure 1D. We realized that in table 1 and figure 1D we summarize EMT-related proteins with canonical CMA motif(s), while TGFBR2 contain two advanced CMA motifs (KTRKL and NVLRD) that should not have been presented in figure 1 at the first place. Instead, TGFβ1 should have been shown, as we do, in fact, discover TGFBR2 as a CMA-target protein by the *in vitro* data presented in figure 6/7 (previous 5/6). We have now clarified this in figure 1 and only state examples of EMT-related proteins with canonical CMA motifs, as indicated with a star ✨.

The proteomics and *in silico* analysis in figure 1 suggested a new function of CMA on regulating invasiveness of cancer cells and raised an exciting possibility to target pro-metastatic proteins by activating CMA. However, without assessment of the therapeutic impact of CMA, the treatment options and its role in human cancer remain speculative. Thus, before mechanistic dissection of the underlying causes, we set to understand the clinal importance of our findings both using patient data, but also by xenograft models of human LAMP2A knockout (KO) cancer cells, which have been lacking.

22nd Jan 2025

Dear Dr. Vakifahmetoglu Norberg,

Thank you for submitting your revised study, and please accept my apologies for the delay in getting back to you during this busy time of the year. We have now received the reports from the referees who evaluated your revised manuscript. As you will see from the reports below, they are overall satisfied with the revisions, and I will therefore be able to accept your manuscript once the following editorial issues are addressed:

1/ Referees' comments:

Please address the remaining concerns from referees #1 and #3. Please make sure the correct statistical tests are used throughout.

2/ Manuscript text:

- Please remove the blue font and only keep in track changes mode any new modification.

- Methods:

o Human samples: please clarify why no informed consent was required from patients or their family members. Please confirm that the experiments conformed to the principles set out in the WMA Declaration of Helsinki and the Department of Health and Human Services Belmont Report.

o Cells: please provide a statement on authentication.

o Statistical analysis: please provide a statement on exclusion/inclusion criteria, blinding and randomization.

- Data Availability: Thank you for depositing your datasets. Please note that they must be made public before acceptance of the manuscript. Please remove "All other data generated or analyzed during this study are included in this published article (and its supplementary information files)."

- Acknowledgements: Please note that the information provided in the manuscript and the submission system should match, and all funding bodies should be entered into our system, including project numbers.

- Please remove "Supplementary Material/Appendix/Expanded View Figures".

3/ Figures and Appendix:

- Figures EV1-4 should also be uploaded individual, high resolution figure files.

- For figures with images and zoomed sections, please add highlight boxes for all zoomed areas.

- Please make sure that all figures and figure panels are referenced in the text. Currently, callouts are missing for Appendix Table S1, Fig. 5D.

- Please remove "Source data are available online for this figure" in all figure legends.

- Please address the queries from our copy editors in the figure legends:

1. Please note that the exact p values are not provided in the legends of figures 1B, 2C, 3A, B; 3F, 4E, H, 5C, D, F, G, I; 6B-F; 7A, D; 8A-G. EV2B, E; EV3 D, E; EV4 D.

2. Please indicate the statistical test used for data analysis in the legends of figures 4A, EV1 D, F.

3. Please note that the box plots need to be defined in terms of minima, maxima, centre, bounds of box and whiskers, and percentile in the legends of figures 1B, 3F, 4H, 7D, 8A, F

4. Please note that information related to n is missing in the legends of figures 4E

5. Please note that for heatmap present in figures 1A, 2B, 4B a numbered scale bar is not provided. This needs to be rectified.

4/ Thank you for providing Source Data. Main figure source data should be uploaded as one file per figure; EV source data should be one file per figure, then zipped together. Carefully check the labeling of your EV Source Data.

5/ Checklist:

- Please fill in manuscript information (top left corner)

- Please fill in the subsections about blinding and inclusion/exclusion criteria in the Statistics section.

- Please fill in the right section of "Ethics", on informed consent and Helsinki declaration.

- Data Availability: please make sure that you indeed need to fill the second subsection, "human clinical and genomic datasets" as I am not sure it is the case here.

6/ Synopsis:

I slightly edited your synopsis text to fit our format. Please let me know if you agree with the changes, or amend as you see fit:

Metastatic progression is driven by the expression and stabilization of pro-metastatic proteins. This study reveals how Chaperone-Mediated Autophagy (CMA) deficiency stabilizes these proteins and highlights the therapeutic potential of CMA activation in mesenchymal cancers.

- Metastatic lesions displayed suppressed LAMP-2A expression compared to matched primary tumors from cancer patients.

- Mesenchymal tumor cell growth, dissemination and metastasis were promoted by the loss of LAMP-2A, in vitro and in vivo.

- CMA controlled TGF β signaling through TGF β R2 degradation, limiting epithelial-to-mesenchymal transition and migratory

potential of cancer cells.

- Reciprocally, TGF β signaling stimulation dampened LAMP-2A levels.
- CMA deficiency caused a metabolic shift promoting mesenchymal cancer cell proliferation, by enhancing pentose phosphate pathway (PPP) and nucleotide biosynthesis

Thank you for providing a nice synopsis picture. A small portion was cropped to serve as a thumbnail on our webpage (attached). Let me know if you agree with the selection, or kindly provide a alternative one (115px x 70 px).

7/ As part of the EMBO Publications transparent editorial process initiative (see our Editorial at <http://embomolmed.embopress.org/content/2/9/329>), EMBO Molecular Medicine will publish online a Review Process File (RPF) to accompany accepted manuscripts.

This file will be published in conjunction with your paper and will include the anonymous referee reports, your point-by-point response and all pertinent correspondence relating to the manuscript. Let us know whether you agree with the publication of the RPF and as here, if you want to remove or not any figures from it prior to publication.

I look forward to receiving your revised manuscript.

Yours sincerely,

Lise Roth

***** Reviewer's comments *****

Referee #1 (Comments on Novelty/Model System for Author):

The manuscript describes results supporting the conclusion that human cancer cell-specific knockout of LAMP2A, a limiting protein in chaperone-mediated autophagy (CMA), promotes aggressiveness of mesenchymal xenograft tumors. This is novel.

Referee #1 (Remarks for Author):

The revision has significantly improved the manuscript, considering both the data and presentation. Most of the points mentioned in my previous report have been addressed, except one.

Major

1. Note on my previous report and the Rebuttal letter:

"Comment 3. The western blots in Figure 6 should be quantified and the differences should be tested for statistical significance.

Reply: All quantifications are now added for all western blots."

The previous Figure 6 is now Figure 7. Even though quantifications have been added, statistical significance of the differences is still missing. Thus, the reader does not know whether the differences are from one single experiment or multiple independent experiments.

Minor

2. Line 197-98: 'allowing simultaneous comparison of multi sample detection of dynamic changes in protein abundance' The wording is confusing. Suggestion: allowing simultaneous comparison of dynamic changes in protein abundance in multiple samples

3. Line 217-218: 'comparing the LAMP-2A dependent proteome to those enriched upon treatment alone' It is unclear which samples are compared here - please be clear in defining the comparisons and treatments.

4. Lines 279-281: 'While AC220+Spautin1 treatment significantly elevated number of puncta per cell in wild-type (WT) cancer cells (measured by the PAmcherry- KFERQ reporter)' - Suggested wording: While AC220+Spautin1 treatment significantly elevated the number of CMA-indicative puncta per cell in wild-type (WT) cancer cells (measured by the PAmcherry-KFERQ reporter)

5. Lines 314-316: 'Mice bearing HT1080 LAMP-2A KO cells showed a heavier tumor burden in lung tissue with a significantly increased number of metastatic foci on the lung surface than their WT littermates' Should read: Mice bearing HT1080 LAMP-2A KO cells showed a heavier tumor burden in lung tissue with a significantly increased number of metastatic foci on the lung surface compared with mice bearing WT HT1080 cells. Same correction on line 320: WT littermates -> mice injected with WT cells

Referee #2 (Comments on Novelty/Model System for Author):

Addition of mouse metastasis assays strengthens the conclusions of the paper.

Referee #2 (Remarks for Author):

My concerns have been addressed.

Referee #3 (Comments on Novelty/Model System for Author):

The authors have extended and validated their data in zebrafish using experimental metastasis mouse models.

About the statistics: while previously not commented on, I now noticed that the authors used throughout the study only student t-test even when analyzing more than 2 data sets, for which ANOVA is more appropriate. Probably, this will not change the outcome of their results, but it drew my attention when boxing the technical quality.

Referee #3 (Remarks for Author):

The manuscript has been largely improved and the reviewer's comments have been addressed satisfactorily. The authors have performed the requested additional experimental metastasis experiments validating the data obtained in the zebrafish model on the protective role of CMA against invasion and metastasis.

Reviewer's comments

General response: We sincerely thank all the reviewer for the comprehensive and insightful assessment of our manuscript. We are gratified by the reviewer's response on our revised manuscript, additional experimental approach and the clarity of our presentation. Below is our point-by-point response associated to the new comments on the revised (R2) manuscript.

Referee #1 (Comments on Novelty/Model System for Author):

The manuscript describes results supporting the conclusion that human cancer cell-specific knockout of LAMP2A, a limiting protein in chaperone-mediated autophagy (CMA), promotes aggressiveness of mesenchymal xenograft tumors. This is novel.

Referee #1 (Remarks for Author):

The revision has significantly improved the manuscript, considering both the data and presentation. Most of the points mentioned in my previous report have been addressed, except one.

Response: We thank the reviewer for these comments.

Major

1. Note on my previous report and the Rebuttal letter:

"Comment 3. The western blots in Figure 6 should be quantified and the differences should be tested for statistical significance.

Reply: All quantifications are now added for all western blots."

The previous Figure 6 is now Figure 7. Even though quantifications have been added, statistical significance of the differences is still missing. Thus, the reader does not know whether the differences are from one single experiment or multiple independent experiments.

Response: For all the western blots shown in figure 7, we already provided densitometry data (excel files) from 3 independent blots as source data. In figure 7, we presented the quantifications as (fold n=3) average fold change. The statistical significance is presented as Appendix table S4.

Minor

2. Line 197-98: 'allowing simultaneous comparison of multi sample detection of dynamic changes in protein abundance' The wording is confusing. Suggestion: allowing simultaneous comparison of dynamic changes in protein abundance in multiple samples.

3. Line 217-218: 'comparing the LAMP-2A dependent proteome to those enriched upon treatment alone' It is unclear which samples are compared here - please be clear in defining the comparisons and treatments.

4. Lines 279-281: 'While AC220+Spautin1 treatment significantly elevated number of puncta per cell in wild-type (WT) cancer cells (measured by the PAmcherry- KFERQ reporter)' - Suggested wording: While AC220+Spautin1 treatment significantly elevated the number of CMA-indicative puncta per cell in wild-type (WT) cancer cells (measured by the PAmcherry-KFERQ reporter)

5. Lines 314-316: 'Mice bearing HT1080 LAMP-2A KO cells showed a heavier tumor burden in lung tissue with a significantly increased number of metastatic foci on the lung surface than their WT littermates' Should read: Mice bearing HT1080 LAMP-2A KO cells showed a heavier tumor burden in lung tissue with a significantly increased number of metastatic foci on the lung surface compared with mice bearing WT HT1080 cells. Same correction on line 320: WT littermates -> mice injected with WT cells.

Response: All the suggested reformulations have been implemented.

Referee #2 (Comments on Novelty/Model System for Author):

Addition of mouse metastasis assays strengthens the conclusions of the paper.

Referee #2 (Remarks for Author):

My concerns have been addressed.

Response: We thank the reviewer for the comment.

Referee #3 (Comments on Novelty/Model System for Author):

The authors have extended and validated their data in zebrafish using experimental metastasis mouse models.

About the statistics: while previously not commented on, I now noticed that the authors used throughout the study only student t-test even when analyzing more than 2 data sets, for which ANOVA is more appropriate. Probably, this will not change the outcome of their results, but it drowed my attention when boxing the technical quality.

Response: After revisiting all figures in detail, we only identified panel 6B, where either multiple t-tests or ANOVA could be applied to compare cell line groups (mesenchymal cell lines vs epithelial cell lines). We present ANOVA analysis for this panel and have added the corresponding *P*-value to the figure legend. However, as eluted by the reviewer, there was no difference in the outcome of both statistical tests. In all other analyses the difference between two conditions was compared, where Student's t-test was applied.

Referee #3 (Remarks for Author):

The manuscript has been largely improved and the reviewer's comments have been addressed satisfactorily. The authors have performed the requested additional experimental metastasis experiments validating the data obtained in the zebrafish model on the protective role of CMA against invasion and metastasis.

Response: We thank the reviewer for the comment.

3rd Feb 2025

Dear Dr. Vakifahmetoglu Norberg,

Thank you for submitting your revised files. Before I can accept your manuscript, please address the following remaining concerns:

1/ Statistical analyses:

Please carefully check the test employed in cases where 3 groups or more were tested (i.e. Fig. 1B, 5I, 6D, E, F and Fig. 8), and use ANOVA whenever appropriate.

2/ Informed consent:

As per our journal policy, informed consent must be obtained from all subjects. Please provide written documentation attesting that the ethical review board deemed that informed consent from patients or their families was not necessary in this case.

3/ Deposition of MS data:

The DOI for your article is: 10.15252/emmm.202419764

4/ Zommed in sections in figures:

Please note that a circle is not at the right place in Fig. EV2D.

5/ Checklist:

Please fill in the subsections about blinding and inclusion/exclusion criteria in the Statistics section.

I look forward to reading a new revised version of your manuscript as soon as possible.

Yours sincerely,

Lise Roth

Dear Dr. Roth

Thank you for your email,

I wasn't sure about how to provide the Patient sample ethics, so I am emailing those here to you directly. Dr, Simon Ekman (co author and the clinician) emailed the following below:

Should we keep the same statement as written in the manuscript as it is, or would you want us to change any, or please feel free to change it as you wish.

Thank you for all your help and assistance,

Best regards

Helin

Från: Simon Ekman <simon.ekman@ki.se>

Skickat: den 4 februari 2025 00:27

Till: Helin Norberg <helin.norberg@ki.se>; Per Hydbring <per.hydbring@ki.se>

Ämne: SV: EMM-2024-19764-V3 Decision Letter

All patients were deceased at the time of the study and thus we were not able to retrieve informed consent from the patients. It was considered by us and approved by the ethical review board that contacting the family members of the deceased would cause more harm than good by risking to evoke painful memories/traumas or rekindle grief. We therefore asked for a waiver from asking the patients' families which was approved by the ethical review board. As a written proof of this, please see two documents attached: one document containing the ethical application with English translation of the text (marked in orange colour) relevant for this study and where we motivate not to contact the families and one document with the approval from the ethical review board with English translation of key points.

The authors addressed the remaining editorial issues.

17th Feb 2025

Dear Dr. Vakifahmetoglu Norberg,

Thank you for bearing with the last editorial concerns, and for submitting your revised files. Please accept my apologies for the delay in getting back to you, which was caused by the absence of informed consent from the patients. Indeed, as our policy clearly states that informed consent MUST be obtained, I had to further consult with our head of publication and with one of our bioethic advisors.

I am glad to say that I have now heard back from this advisor, who is supportive of publication in light of the approval from the ethic review board.

I am thus pleased to inform you that your manuscript is now accepted for publication and being sent to our publisher to be included in the next available issue of EMBO Molecular Medicine.

With kind regards,

Lise Roth
